# Novel role of the synaptic scaffold protein Dlgap4 in ventricular surface integrity and neuronal migration during cortical development

Delfina M. Romero [1,2,3,15], Karine Poirier[4], Richard Belvindrah [1,2,3], Imane Moutkine[1,2,3], Anne Houllier[1,2,3], Anne-Gaëlle LeMoing[5], Florence Petit [6], Anne Boland [7], Stephan C. Collins[8], Mariano Soiza-Reilly [9], Binnaz Yalcin [8], Jamel Chelly [10], Jean-François Deleuze [7], Nadia Bahi-Buisson[11,12,13,14] & Fiona Francis [1,2,3✉]

Subcortical heterotopias are malformations associated with epilepsy and intellectual disability, characterized by the presence of ectopic neurons in the white matter. Mouse and human heterotopia mutations were identified in the microtubule-binding protein Echinoderm microtubule-associated protein-like 1, EML1. Further exploring pathological mechanisms, we identified a patient with an EML1-like phenotype and a novel genetic variation in *DLGAP4*. The protein belongs to a membrane-associated guanylate kinase family known to function in glutamate synapses. We showed that DLGAP4 is strongly expressed in the mouse ventricular zone (VZ) from early corticogenesis, and interacts with key VZ proteins including EML1. *In utero* electroporation of *Dlgap4* knockdown (KD) and overexpression constructs revealed a ventricular surface phenotype including changes in progenitor cell dynamics, morphology, proliferation and neuronal migration defects. The *Dlgap4* KD phenotype was rescued by wild-type but not mutant DLGAP4. Dlgap4 is required for the organization of radial glial cell adherens junction components and actin cytoskeleton dynamics at the apical domain, as well as during neuronal migration. Finally, *Dlgap4* heterozygous knockout (KO) mice also show developmental defects in the dorsal telencephalon. We hence identify a synapse-related scaffold protein with pleiotropic functions, influencing the integrity of the developing cerebral cortex.

[1] INSERM UMR-S 1270, F-75005 Paris, France. [2] Sorbonne University, F-75005 Paris, France. [3] Institut du Fer à Moulin, F-75005 Paris, France. [4] INSERM UMR-S 1163, Translational Genetics, Imagine Institute, Paris Cité University, Necker Enfants Malades University Hospital, Paris, France. [5] Service of Pediatric Neurology, CHU Amiens, Amiens, France. [6] Department of Clinical Genetics. Hôpital Jeanne de Flandre, CHU Lille, F-59000 Lille, France. [7] National Center of Human Genomics Research (CNRGH), François Jacob Institute of Biology, CEA, University Paris-Saclay, F-91057 Evry, France. [8] INSERM UMR-S 1231, University of Bourgogne Franche-Comté, 21000 Dijon, France. [9] Instituto de Fisiología, Biología Molecular y Neurociencias (IFIBYNE), CONICET, Universidad de Buenos Aires, Buenos Aires, Argentina. [10] IGBMC-CNRS UMR-S 7104, INSERM UMR-S 964, Strasbourg, France. [11] Laboratory of Genetics and Development of the Cerebral Cortex, INSERM UMR-S 1163, Imagine Institute, Paris, France. [12] Paris Cité University, Imagine Institute, Paris, France. [13] Pediatric Neurology APHP- Necker Enfants Malades University Hospital, Paris, France. [14] Centre of Reference, Rare Causes of Intellectual Deficiences, APHP- Necker Enfants Malades University Hospital, Paris, France. [15]Present address: Instituto de Biología Celular y Neurociencias "Prof. E. De Robertis" (IBCN), Facultad de Medicina, Universidad de Buenos Aires, CONICET, Buenos Aires, Argentina. ✉email: fiona.francis@inserm.fr

The appropriate establishment of neuronal circuits is critical for central nervous system (CNS) function. These processes require that newborn neurons are generated, migrate, differentiate and establish neuronal connections at the correct time and place during embryogenesis[1,2]. Perturbations of neuronal migration as well as defects in progenitor function are associated with cortical malformations, which are major causes of developmental disability and epilepsy[3–6].

Gray matter heterotopias are common cortical malformations that are typically associated with epilepsy and variable developmental delay in children and young adults. Affected patients can be divided into two groups based on clinical and imaging characteristics: subependymal and subcortical, including specifically band heterotopias (also called double cortex), with distinct underlying genetic disorders and clinical outcomes[7]. The most common type of heterotopic gray matter is the subependymal nodular type[8]. This cortical malformation is characterized by neural progenitor defects and the failure of neurons to migrate to their final positions in the cortex. Nodules of gray matter are located in the wall of the lateral ventricle, in a subependymal position[6,9,10]. Most cases of subependymal heterotopia are X-linked, showing mutations in *filamin A* (*FLNA*), coding for an actin-binding protein[9,11]. In contrast, much less is known about subcortical heterotopias, especially atypical cases, for which the definition is based on brain imaging studies. This cortical malformation is associated with abnormally positioned neurons in the white matter, suggesting neuronal migration defects, which may or may not be cell autonomous[6].

The main progenitor cells during corticogenesis in the rodent are apical radial glia (RGs)[2,12]. Their somata are situated in the proliferative ventricular zone (VZ), they have basal processes which extend towards the pial surface and apical processes which descend to the ventricular surface (VS). Apical processes contain polarity complexes, centrosomes, and the primary cilia[12]. As well as generating post-mitotic neurons, RGs give rise to other progenitor cells including intermediate progenitors localized in the subventricular zone (SVZ), and basal RGs (bRGs) mainly accumulating in an outer SVZ in gyrencephalic brains[13–18].

Mitotic spindle anomalies have been associated with a variety of cortical malformations[5], however, the possible actors and pathways involved remain still poorly understood. RGs also express epithelial to mesenchymal regulators involved in polarity transitions from a progenitor to a detached cell (e.g., intermediate progenitor, bRG, or a migrating neuron). This transition involves the formation/dissolution of adherens junctions (AJs), reorganization of the actin and microtubule cytoskeletons, and subsequent morphological changes[19,20].

We identified the first gene responsible for ribbon-like subcortical heterotopia in unrelated families with autosomal recessive *EML1* mutations[21]. Patients have a mixed form of subependymal and subcortical heterotopia, as well as macrocephaly, polymicrogyria (too many folds on the surface of the brain), and/or agenesis of the corpus callosum[22,23]. We demonstrated that Eml1 is expressed in RGs and its mutation leads to their mispositioning, subsequently affecting neuronal migration, and leading to a subtly altered VS[22–24]. However, its cytoskeletal roles and interacting protein partners in RGs still need to be further elucidated.

In this study we identify a new gene for subependymal and subcortical heterotopia, revealing a de novo *Disks large-associated protein 4* (*DLGAP*) frameshift variation in a patient with unilateral heterotopia and polymicrogyria, resembling the *EML1* phenotype. A second family with cortical malformations carrying *DLGAP4* in combination with *DLGAP1* gene variations was also identified. The DLGAP protein family (SAP90/PSD-95-associated proteins) is a group of scaffold proteins that interacts with DLGs of the membrane-associated guanylate kinase (MAGUK) superfamily[25].

Although well known for their function in synapses[26], several DLGAPs are also apparently expressed in the cortical VZ[27,28].

In this work, as well as identifying *DLGAP* mutations, we further show that DLGAP4 interacts with EML1. We demonstrate that Dlgap4 is strongly expressed in the VZ, and interacts with other VS proteins, including Dlg1 and β-catenin. Dlgap4 is also expressed in migrating neurons. Furthermore, a heterotopia-like malformation is induced in the mouse developing cortex after either *Dlgap4* KD or its overexpression. Moreover, a *Dlgap4* KO mouse shows neuroanatomical defects in the dorsal telencephalon. Our data show that Dlgap4 is important for VS integrity by influencing the F-actin cytoskeleton and progenitor cell behavior, as well as being important during neuronal migration. This study hence reveals important pleiotropic and non-synapse-related roles of Dlgap4/DLGAP4 during mouse and human corticogenesis.

## Results

***DLGAP4* mutations are associated with cortical malformations**. We carried out a study investigating the molecular basis of cortical malformations (150 patients) and particularly patients with rare, unexplained forms of subcortical heterotopia with or without subependymal heterotopia (10 patients). Trio-based exome sequencing identified a variation in *DLGAP4* (ENST00000373913) in the patient P616-3 (Supplementary Tables 1–3). Brain MRI showed the presence of extensive subependymal heterotopia at the left occipital junction, extending along the temporal horn of the left lateral ventricle, associated with occipital lobe polymicrogyria[6] (Fig. 1a, Supplementary information). P616-3 presented a de novo variant consisting of an insertion of a 7-nucleotide repeat (exon 12, NM_014902.5, c.2714_2715insCAGCTGG), affecting the last part of the conserved GH1 domain, a Guanylate Kinase Associated Protein (GKAP) homology domain 1 (Fig. 1b–d). In DLGAP4, the GH1 domain comprises the amino acids (aa) between the glutamic acid at aa 804 (Glu804) and the tryptophan Trp907[25]. The first of three consecutive CAGCTGG sequences starts at the asparagine Asn905 in P616-3, whereas this sequence is only duplicated in the normal human population. The insertion leads to a frameshift mutation changing the open reading frame and resulting in a different C-terminus after the Leu909 (Fig. 1c, e), including the loss of a poly proline-rich domain. The last 80 aa of the wild type (WT) protein are hence lost and replaced by 95 alternative aa. This mutation has a disease-causing predicted value of 1 in a Mutation Taster analysis[29]. In addition, *DLGAP4* has a pLI score of 0.992 in the Genome Aggregation Database (gnomAD), suggesting that the gene is highly intolerant to loss of function mutations.

In order to assess if the P616-3 mutation affects the overall predicted 3D structure and function, we took advantage of the I-TASSER iterative method[30]. As well as the abnormal C-terminus, this method predicts changes in the overall structure of the protein (Supplementary Fig. 1a, DLGAP4 mutant). Thus, the seven nucleotide insertion affecting the C-terminus of DLGAP4 in patient P616-3 is predicted to lead to multiple changes and to impact overall protein function[31].

Other cortical development gene mutations e.g., in DCX, have also been shown to give rise to subcortical heterotopia or reduced cortical folding problems such as pachygyria, depending on the patient[6]. Interestingly, trio-based exome sequencing revealed a second DLGAP family, P477, with affected monozygous twins having a distinct cortical malformation characterized by parieto-occipital pachygyria. They exhibited gene variations including a missense variant in *DLGAP4* inherited from the father and a second missense variant in *DLGAP1* from the mother (*DLGAP4* c.2893T>G, p.Ser965Ala, exon 13; *DLGAP1* c.1397A>G, p.Asp466Gly, exon 8) (Supplementary

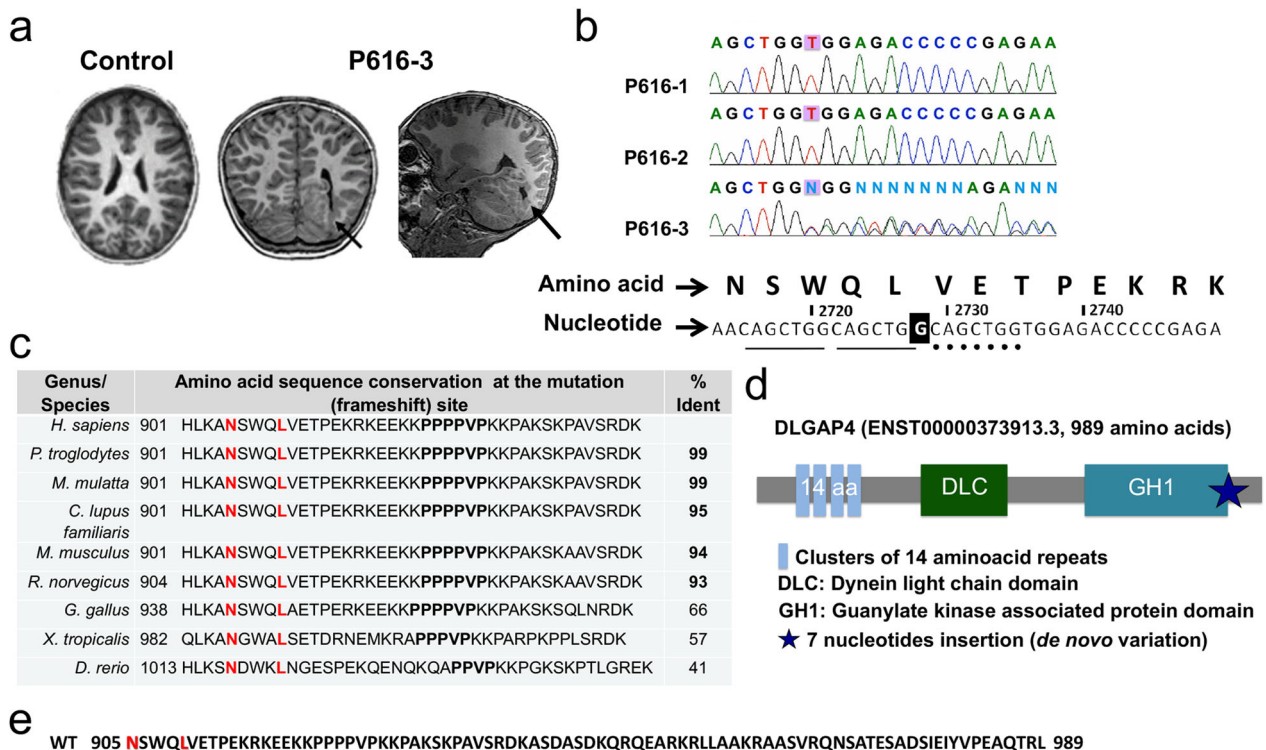

**Fig. 1 DLGAP4 mutation is associated with human cortical malformations. a** Representative T2-coronal brain MRIs of heterotopia patient (P616-3) were performed at the age of 10 years old, compared to an aged-matched control. The arrow indicates the region of the heterotopia in axial and sagittal sections. **b** Confirmation of a 7-nucleotide insertion (CAGCTGG, three repeats instead of two) using Sanger's sequencing method. The mutation is absent from the parents (P616-1 and 2) and identified as de novo in P616-3. **c** Conservation of the mutated region during evolution. The "% Ident" refers to the complete primary structure compared to the canonical human sequence. The region of the repeats starts at aminoacid (aa) N905, an extra repeat leads to aa changes from aa L909. **d** Structure and domains of DLGAP4 protein. A star indicates the region of the mutation. **e** Comparison of the WT and mutant C-terminal regions showing new aa from L909.

information and Supplementary Tables 1–3, Supplementary Fig. 1 b–e). The disease-causing predicted value for P477 *DLGAP4* and *DLGAP1* mutations is 0.99 in Mutation Taster analyses[31].

Given the severe cortical phenotypes, we queried DLGAP1 and DLGAP4 expression patterns across human cortical development. Existing RNA sequencing data from single human cells showed the expression of *DLGAP1* and *DLGAP4* in progenitor cells and neurons, being higher and more ubiquitous for the latter gene (Supplementary Fig. 1b)[32]. DLGAP1 is the closest in homology to DLGAP4, compared to other family members, and immunoprecipitation (IP) experiments showed that these proteins are present in the same protein complex (Supplementary Fig. 1c, d). These combined data led us to further assess the potential role of these proteins and notably DLGAP4 in cortical development.

**Dlgap4 is expressed in the mouse developing cortex**. We assessed *Dlgap4* expression by in situ hybridization (ISH) at different stages of mouse cortical development (from E13.5 to E18.5, Fig. 2a; Supplementary Fig. 2a, and data not shown). At every stage, *Dlgap4* was expressed in the VZ, where RGs are located, as well as in the cortical plate (CP), populated by neurons. *Dlgap4* was also detected in the SVZ, in the intermediate zone (IZ) populated by migrating neurons, as well as the ganglionic eminence. Consistent with ISH results, the single-cell transcriptomic atlas of the developing mouse neocortex at E14.5,

also showed the expression pattern of *Dlgap4* transcripts in the same cell types[33] (Supplementary Fig. 2b).

Immunohistochemistry (IHC) at E14.5 using a specific anti-Dlgap4 antibody (Fig. 2b–f, Supplementary Fig. 2c) also showed Dlgap4 protein expression with variable intensities in all regions of the cortical wall, including where presumed progenitor cells are located, in the VZ (apical RG) and SVZ (intermediate progenitors). The fluorescence was noticeably intense at the VS, and a strong expression was also observed in neurons in the CP. There was an apparent difference in expression of Dlgap4 and class III beta tubulin (TuJ1), the latter notably strongly expressed in axons in the IZ. TuJ1 and Dlgap4 however both showed expression in the cell bodies of migrating neurons in the SVZ[34] (Fig. 2d, e). Indeed, Dlgap4 labeling appeared present in the soma regions of each cell type, including VZ RG (Tbr2⁻) and SVZ Tbr2⁺ intermediate progenitors, as well as neurons in the developing mouse brain (Fig. 2c–f, Supplementary Fig. 2b). A punctate peri-nuclear pattern of Dlgap4 was observed in dividing and interphase Neuro2A cells, and in primary cultures of RGs, the latter identified by GFP expression driven by a Blbp RG-specific promoter (Supplementary Fig. 2d, e).

**DLGAP4 interacts with EML1 and DLG1**. Since *EML1* mutations give rise to a similar heterotopia phenotype[21], we tested for the possible interaction of DLGAP4 with EML1 by co-IPs (Supplementary Fig. 2f–i). Indeed, GFP-EML1 was identified in anti-Flag DLGAP4 IPs, and consistently, Flag-DLGAP4 was

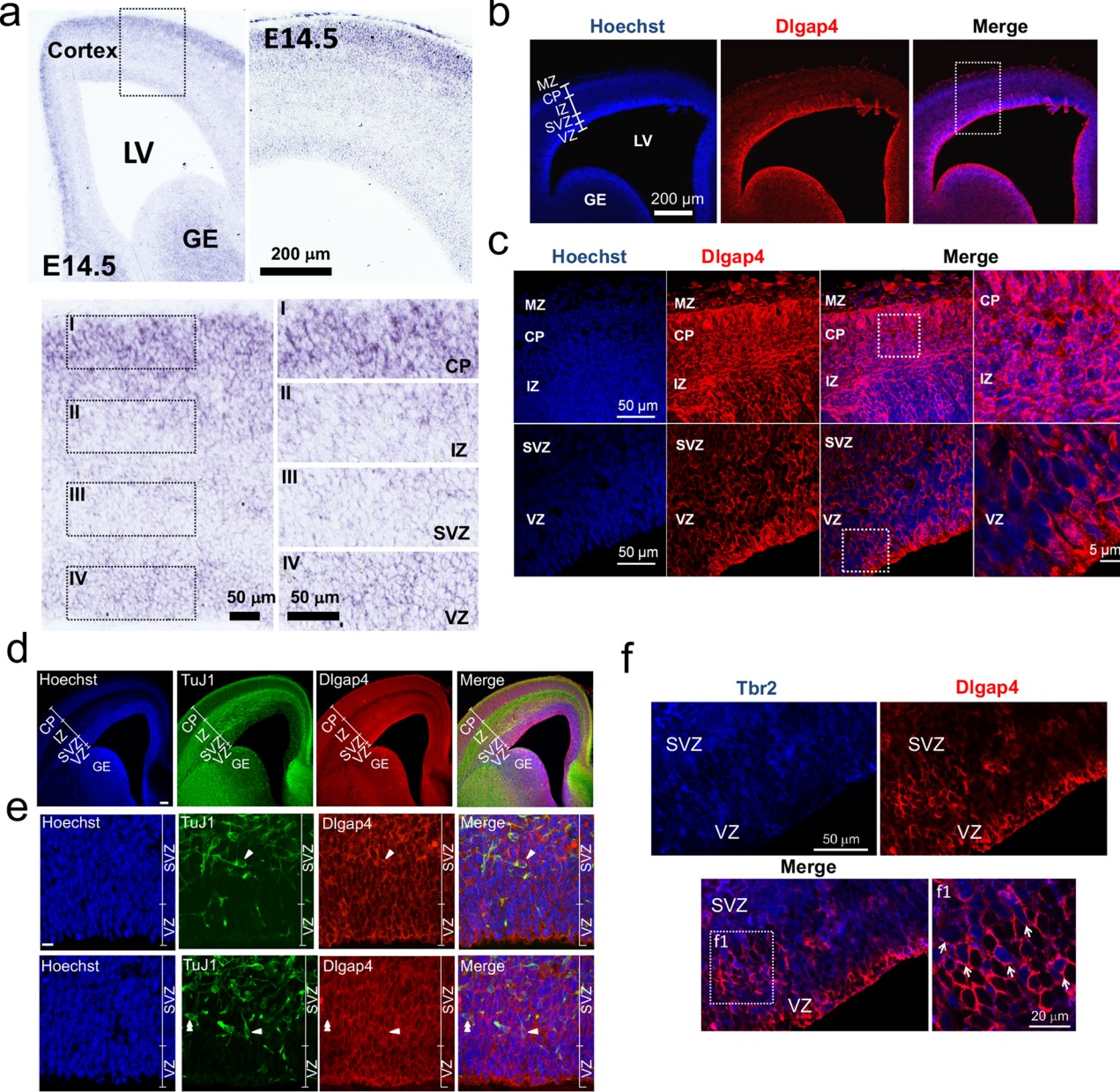

**Fig. 2 Dlgap4 is highly expressed in the developing cortex. a** In situ hybridization (ISH) at E14.5 shows mRNA expression of *Dlgap4* throughout the mouse cortical wall. Coronal sections are shown. Representative immunofluorescence images showing, Dlgap4 (red) and Hoechst staining (blue) coronal sections at E14.5, at lower (**b**) and higher (**c**) magnification. Images correspond to 10 μm stack projections. LV: lateral ventricle, VZ: ventricular zone, SVZ: sub-ventricular zone, IZ: intermediate zone, CP: cortical plate, MZ: marginal zone, GE: ganglionic eminence. **d** Co-labeling between Dlgap4 and class III beta tubulin (TuJ1) in E16.5 mouse brain (low magnification). **e** Immunolabelings of Dlgap4 (red), TuJ1 (green), and Hoechst staining (blue) show little overlap in expression in the axons, although high co-labeling in the cell bodies (higher magnification). Arrowheads indicate individual neurons (**f**) Co-labeling between Dlgap4 (red) and the intermediate progenitor marker Tbr2 (blue) shows Dlgap4 expression in the soma at E14.5 mouse brain. Z-stack 1 μm thickness and single confocal plane with higher magnification in far-right panel (**f1**). Arrows indicate individual cells with Tbr2 and Dlgap4 double labeling.

identified in anti-GFP EML1 IPs (Supplementary Fig. 2f, i). Then, WT and mutant DLGAP4 and EML1 proteins were compared. When GFP-DLGAP4 incorporated the mutation identified in patient P616-3, reduced amounts of Flag-EML1 were observed (~70%, $p = 0.062$) (Supplementary Fig. 2g, h). When Flag-EML1 incorporated the Thr243Ala patient missense mutation reported previously[21], the interaction with WT GFP-DLGAP4 was diminished by 94% ($p = 0.022$) (Supplementary Fig. 2h, j).

In addition, Dlgap4 interacts with Dlg1[35], a scaffold protein involved in cell polarity and planar cell divisions during development[19]. We confirmed this interaction in Neuro2A cells,

performing co-IPs in both directions and testing co-localization by immunofluorescence in E14.5 mouse brain (Supplementary Fig. 3a–c, e). Lowered amounts of GFP-DLG1 are found in mutant Flag-DLGAP4 co-IPs when compared to the WT protein (86%, $p = 0.004$) (Supplementary Fig. 3a–d). These experiments show that heterotopia mutations affect the interaction of DLGAP4 with key RG proteins expressed during cortical development.

**Dlgap4 knockdown reveals a cortical progenitor phenotype.** Since the role of Dlgap4/DLGAP4 during cortical development

was previously unknown, we explored this further by examining the effects of its downregulation using in utero electroporation (IUE) in the developing mouse cortex. A short hairpin (Sh) RNA targeting *Dlgap4* and a control (*Ctl*) sequence were designed and cloned in a pCAGMIR30 vector (Shmi*Dlgap4* and Shmi*Ctl* respectively). A reduction of 55% of *Dlgap4* transcripts was obtained after transfection of the Shmi*Dlgap4* vector in Neuro2A cells followed by RT-qPCR experiments ($p = 0.039$, Supplementary Fig. 4a). In addition, DLGAP4 protein immunoreactivity was reduced by 62.6% in Blbp-GFP$^+$ cells in the VZ, 48 h after IUE ($p < 0.0001$) (Supplementary Fig. 4b–d).

In these latter experiments, Shmi*Dlgap4* and Shmi*Ctl* constructs were individually co-electroporated with a Blbp-GFP reporter plasmid at E14.5, with sacrifices 1–2 days later, giving rise to an expression of GFP in RGs, with GFP remaining after RG differentiation into intermediate progenitors and neurons[36] (Fig. 3a). Nestin$^+$ RG fibers appeared occasionally disorganized in basal regions after *Dlgap4* KD (yellow arrows, Fig. 3a), and also close to the VS, with some horizontal and rearranged fibers (Supplementary Fig. 4e). Upon binning analysis (Fig. 3b), E15.5 Blbp-GFP$^+$ cells were found reduced in the apical-most bins (1–2) in the *Dlgap4* KD condition ($p < 0.003$). Conversely, the proportion of cells in bin 3 was increased ($p = 0.042$) and a similar tendency was also observed in the more superficial bins (Fig. 3b). No differences in the total number of electroporated Blbp-GFP$^+$ cells were observed between *Dlgap4* KD and *Ctl* brains (Fig. 3c). Strikingly, *Dlgap4* KD produced an irregular VS (71% of ShmiRNA-*Dlgap4* brains analyzed) and a proportion of Blbp-GFP$^+$ cells was found extruding into the ventricles (~50% of the same brains) resembling the situation described for human subependymal heterotopia (Fig. 3d). Staining for neuronal markers (e.g., TuJ1) also identified ectopic ventricular neurons at E15.5 (Supplementary Fig. 5a), suggesting abnormal migration/ accumulation of neurons at the VS. Notably, these TuJ1 cells were mostly non-GFP positive, indicating that their ectopic positioning resulted from non-cell-autonomous mechanisms induced by *Dlgap4* KD. In addition, 48 h after IUE, *Dlgap4* KD Blbp-GFP$^+$ cells still showed an abnormal distribution compared to control (see arrows, Supplementary Fig. 5b, c).

Apical RGs are integrated into the AJ belt mediating cell–cell adhesion, in which cadherin, cytoplasmic catenins, and F-actin are critical components[20]. Even in the absence of ventricular extruding ectopic GFP$^+$ cells, a decrease in VS expression of F-actin was observed in the electroporated regions ($p < 0.0001$) (Fig. 3e, f). Aberrant β-catenin and N-cadherin expression and localization were also evident (Fig. 3g, white arrows). This may indicate a VS weakening, potentially due to cell-to-cell adhesion defects contributing to a ventricular ectopic cell phenotype. We also found that Flag-DLGAP4 co-immunoprecipitated with GFP-β-catenin (Fig. 3h–j), and when DLGAP4 is mutated with the P616-3 variant, co-IP of β-catenin is reduced by 47% ($p = 0.045$) (Fig. 3h–j). These experiments further suggest a role for Dlgap4 at the VS. Hence, *Dlgap4* KD results in disruption of cortical development, impacting VS integrity and cell positioning.

### *Dlgap4* KD affects position and morphology of progenitors.
Next, we analyzed whether *Dlgap4* KD affects the behavior of Pax6$^+$ and Tbr2$^+$ cells (detecting RGs and intermediate progenitors, respectively) in the cortical wall at E15.5 (Fig. 4a–f). An overall decrease (25%) of electroporated Pax6$^+$ Blbp-GFP$^+$ cells ($p = 0.027$) and an increase (81%) of Tbr2$^+$ Blbp-GFP$^+$ cells ($p = 0.041$) were observed after *Dlgap4* KD (Fig. 4a–f). Additionally, we observed an altered distribution of Pax6$^+$ Blbp-GFP$^+$ cells (in bins 1–2, Fig. 4c), Tbr2$^+$ Blbp-GFP$^+$ cells (in bins 1–4), and total Tbr2$^+$ cells (in bins 2–3) within the cortical wall, with

proportionally more Tbr2$^+$ cells present in superficial bins (Fig. 4f, Supplementary Fig. 5i). Furthermore, the morphological analysis of progenitor cells in electroporated brains showed that Tbr2$^+$ Blbp-GFP$^+$ cells from the *Dlgap4* KD condition present a decreased number of cell processes (25%, $p = 0.034$), with a longer extension (41%, $p = 0.041$), accompanied by an enlarged somatic surface (27%, $p = 0.02$) (Fig. 4g–j).

The change in the proportions of progenitor cells prompted us to assess RG spindle orientations in the VZ. Preliminary quantitative analyses of Blbp-GFP$^+$ cell anaphase angle at the VS showed a decrease in the median value (73.09° *Ctl*, 48.03° Shmi*Dlgap4*, $p = 0.012$), consistent with an increased proportion of vertical and oblique divisions at the expense of horizontal divisions (Supplementary Fig. 6a–c). To further explore these findings, we studied the possible interaction of DLGAP4 with a protein involved in spindle orientation, LGN (coded by the GPSM2 gene), which has been implicated in Chudley-McCullough syndrome, including heterotopia and polymicrogyria[37]. We showed that GFP-LGN was present in Flag-DLGAP4 IPs and vice versa (Supplementary Fig. 6d–f). Mutant DLGAP4 showed a non-significant reduction of immunoprecipitated GFP-LGN (36%, $p = 0.21$) (Supplementary Fig. 6d, e, g). These results and our previous experiments support that DLGAP4 is present in the same protein complexes with either LGN or DLG1, both known for regulating spindle orientation and planar cell division[19,38,39].

### *Dlgap4* KD leads to proliferation and differentiation defects.
We further analyzed proliferating cells after *Dlgap4* KD, by performing injections of BrdU (incorporated in dividing cells) at E15.5, 24 h after IUE, with mice sacrificed 30 min and 24 h later. *Dlgap4* KD brains showed a decreased proportion of proliferating BrdU$^+$ Blbp-GFP$^+$ cells after a 30 min period (35%, $p = 0.0084$), accompanied by abnormal BrdU$^+$ Blbp-GFP$^+$ and total BrdU$^+$ cell distribution, less in bin 1 and tendencies for more in basal bins (Fig. 5a, b, Supplementary Fig. 5d, j). Cell proliferation and cell cycle exit were also assessed at E16.5. The overall proportion of proliferating Ki67$^+$ Blbp-GFP$^+$ cells was reduced in *Dlgap4* KD brains (27%, $p = 0.0073$) (Supplementary Fig. 5e–g). Moreover, with BrdU injection at E15.5, *Dlgap4* KD brains showed a decrease in the proliferation index (21.5%, Ki67$^+$ BrdU$^+$ Blbp-GFP$^+$/BrdU$^+$ Blbp-GFP$^+$, $p = 0.0005$) and a corresponding increase in the proportion of cells exiting the cell cycle (23.3%, Ki67$^-$ BrdU$^+$ Blbp-GFP$^+$/BrdU$^+$ Blbp-GFP$^+$, $p < 0.0001$) (Fig. 5c–e). Exiting cells are increased in proliferative bins (significant in bin 2, Supplementary Fig. 5k). These results may suggest a premature differentiation of cortical progenitors into neurons.

Since *Dlgap4* KD brains showed increased Ki67$^+$ Blbp-GFP$^+$ and Pax6$^+$ Blbp-GFP$^+$ cells in bin 1 (Fig. 4a, c, Supplementary Fig. 5e, f), we performed *en face* confocal imaging and phosphohistone 3 (PH3) labeling to characterize mitotic cells in this region. An increase in PH3$^+$ Blbp-GFP$^+$ mitotic cells by 76.5% ($p = 0.046$) was observed in ShmiRNA*Dlgap4* compared to control brains (Fig. 5f, g). In addition, non-cell-autonomous effects were suspected since many PH3$^+$ cells appeared GFP$^-$. Counting PH3$^+$ Blbp-GFP$^-$ cells (located within less than 10 μm from PH3$^+$ Blbp-GFP$^+$ cells) showed an 86.3% increase in the ShmiRNA*Dlgap4* condition compared to the control ($p = 0.02$) (Fig. 5f, h). Finally, *Dlgap4* KD brains showed an increased trend for total PH3$^+$ cells (GFP$^+$ or GFP$^-$, increased by 55.3% compared to control) (Fig. 5f, i). Since cell cycle exit is increased, these results may suggest defects in mitotic progression elicited by *Dlgap4* KD, or alternative perturbations of the VZ, altering progenitor behavior. *En face* F-actin labeling revealed a tendency for reduced mean intensity in *Dlgap4* KD brains (15%, $p = 0.39$, Supplementary Fig. 6h). The areas of the three brains analyzed did not show obvious VS fragmentation, thus further confirming that non-cell-autonomous effects can occur in the absence of damage.

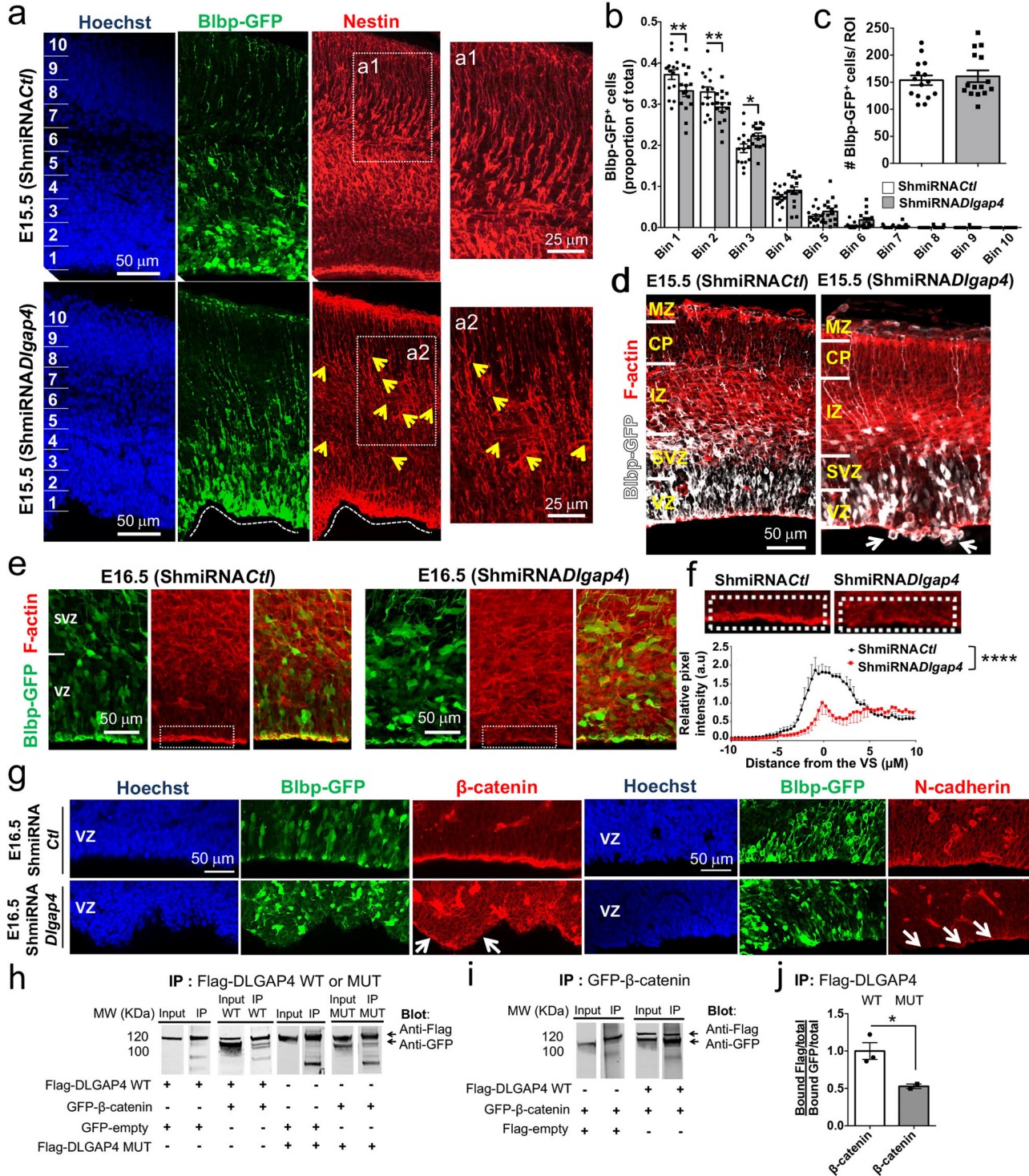

**WT but not mutant DLGAP4 prevents *Dlgap4* KD phenotypes**. The effect of human WT and mutant DLGAP4 overexpression (OE) was also tested using IUE, revealing VZ phenotypes similar to *Dlgap4* KD experiments (Supplementary Fig. 7a, c). TuJ1⁺ cells were also observed extruding into the ventricles (Supplementary Fig. 7a), often Blbp-GFP⁻, suggesting as well a non-cell-autonomous effect. In addition, rosette-like structures were present in the VZ potentially indicating changed adhesion between cells (Supplementary Fig. 7b). Blbp-GFP⁺ total cell counts were not changed 48 h after IUE (Supplementary Fig. 7c, d), although cell distribution was altered, increased in bins 1 and 2, and a clear trend for less cells was observed from bin 5 (Supplementary Fig. 7c–e). Moreover, active caspase-3

was increased in DLGAP4 OE conditions, also detected in GFP⁻ cells ($p = 0.045$) (Supplementary Fig. 7f, g). Indeed, active caspase-3 was also increased in *Dlgap4* KD brains ($p = 0.039$) (Supplementary Fig. 7h, i). Overall, these findings indicate that Dlgap4 expression levels seem to be critical to maintain VZ integrity.

Next, we sought to evaluate whether the *Dlgap4* KD phenotype could be mitigated by concomitant *DLGAP4* OE with the WT or mutant construct (resistant to the ShmiRNA). WT (75%, 9/12 brains) but not mutant (0%, 0/9 brains) DLGAP4 was able to rescue the continuity of the VS and the subependymal heterotopia-like phenotype (Fig. 6a). F-actin labeling showed a disrupted pattern in mutant conditions compared to control or

**Fig. 3 *Dlgap4* KD reveals a progenitor phenotype during mouse cortical development. a** Representative coronal section after IUE E14.5 for control (ShmiRNA*Ctl*, upper panel) and *Dlgap4* KD (ShmiRNA*Dlgap4*, lower panel) constructs one day later. Arrows show disorganized Nestin[+] RG fibers (red). Blbp-GFP[+] (green); IUE: in utero electroporation; *Ctl*: control; KD: knock down; RG: radial glia. **b** Blbp-GFP[+] cell distribution along the cortical wall divided into 10 bins. Two-way ANOVA: Interaction Bin × ShmiRNA condition: $F_{9, 280} = 4.79$, $p < 0.0001$; * $p = 0.042$, ** $p < 0.003$ by Sidak's multiple comparison test. **c** Quantification of total Blbp-GFP[+] cells at E15.5 (ShmiRNA*Ctl* and ShmiRNA*Dlgap4*, n = 15 embryos from at least 4 litters per condition). Means and individual values ± SEM are shown, two-sided unpaired *t*-test, $p = 0.60$. **d** A ventricular surface phenotype was observed in a large proportion of KD brains. Arrows show Blbp-GFP[+] (white) cells misplaced towards the lateral ventricle. F-actin (red); VZ: ventricular zone; SVZ: subventricular zone; IZ: intermediate zone; CP: cortical plate; MZ: marginal zone. **e** Representative images of F-actin staining in ShmiRNA*Ctl* and ShmiRNA*Dlgap4* KD brains, 48 h after IUE (n = 2 *Ctl* and n = 3 KD brains from 2 litters per condition). Images correspond to 10 μm stack projections. **f** Quantification of the relative pixel intensity of F-actin in the VZ per condition. Data represent the mean ± SEM (two-sided paired *t*-test, ****$p < 0.0001$). **g** Representative images showing β-catenin and N-cadherin (both in red) expression and localization patterns in ShmiRNA*Ctl* and ShmiRNA*Dlgap4* brains, 48 h after IUE. Arrows show altered patterns. **h, i** Co-immunoprecipitation (co-IP) analyses performed from Neuro2A cells co-transfected with tagged β-catenin and WT or MUT *DLGAP4* vectors. Total extracts were either loaded directly on the gel (Input) or subjected to IP with anti-Flag or anti-GFP antibodies. Representative blots are shown for all conditions. **j** Quantification data represent the relativized individual values, means ± SEM (n = 3 WT and n = 2 MUT DLGAP4 independent experiments per condition). Statistical analysis was performed using two-sided unpaired *t*-test with Welch's correction for unequal variances (*$p = 0.045$). WT: wild type; MUT: mutant. Images correspond to 10 μm stack projections.

WT OE (Fig. 6a). Overall numbers of Blbp-GFP[+] cells were found to be systematically lower when the KD was combined with the *DLGAP4* mutant OE condition (36% lower, $p = 0.033$) (Fig. 6b, c). Assessing Blbp-GFP[+] cell distribution, we found that only WT DLGAP4 appeared to normalize this aspect of the phenotype, largely restoring Blbp-GFP[+] cells to control levels (Fig. 6b, d). Thus, in bins 1 and 2 the decrease in Blbp-GFP[+] cells in the KD condition was restored to control levels. Moreover, Pax6[+] Blbp-GFP[+] cell counts were restored to control levels by the OE of the WT construct, although appeared excessively increased in the presence of the mutant protein ($p < 0.0001$) (Fig. 6e, f). Tbr2[+]Blbp-GFP[+] cells appeared restored to control levels by both the OE of WT and mutant proteins although to different extents (Fig. 6g, h). Concerning distribution, Pax6[+] Blbp-GFP[+] and Tbr2[+] Blbp-GFP[+] cells appeared restored in bin 1, and partially restored in bin 2, although the mutant led to a noticeable trend for increased cells in basal regions (e.g., from bin 4, Supplementary Fig. 8a, b). Finally, a 30 min BrdU pulse analyzed 24 h after IUE, showed that only the OE of the WT construct was able to correct the decreased proportion of BrdU[+] Blbp-GFP[+] cells (*Ctl* vs. OE *DLGAP4* MUT, $p < 0.001$) (Supplementary Fig. 8c, d), and also altered distribution of BrdU cells after KD (Supplementary Fig. 8e). Thus, mutant DLGAP4 incorporating an altered C-terminus showed differences from the WT protein, and was unable to completely prevent the observed KD cortical phenotypes, including the VS disruption, Blbp-GFP[+] cell distribution, and Pax6[+] Blbp-GFP[+] and BrdU[+] Blbp-GFP[+] cell counts and distribution. Tbr2[+] Blbp-GFP[+] cell numbers and distribution were partially restored with the mutant.

**DLGAP4 impacts actin cytoskeleton dynamics in vitro.** Since the actin cytoskeleton has been shown previously to be perturbed in different types of heterotopia[40–43], we further assessed the potential role of DLGAP4 on actin cytoskeleton dynamics, using the human Retina Epithelial Pigmented (RPE1) cell line, transfected with WT or mutant *DLGAP4*. RPE1 cells form extensive lamellipodia, a highly compact meshwork of actin filaments at the leading edge of the cell, together with a variable number of filopodia (Fig. 7a, GFP-empty control). Of note, no significant differences were found in the proportion of GFP[+] transfected cells in *DLGAP4* WT vs mutant conditions, nor in the expression levels of DLGAP4 protein (data not shown). Transfections with WT and mutant *DLGAP4* showed a decreased proportion of cells presenting lamellipodia morphology compared to control (by 26%, $p = 0.0029$ and 52%, $p < 0.0001$, respectively) (Fig. 7b). Increased filopodia elongation was observed similarly with both protein conditions (control: 3.4 μm ± 0.23 μm; WT DLGAP4:

6.71 μm ± 1.08 μm, $p = 0.036$; mutant DLGAP4: 6.61 μm ± 0.5 μm, $p = 0.041$) (Fig. 7c). Actin stress fibers are long bundles of filaments extending across the cell, making links to the extracellular matrix via integrins and focal adhesion complexes[44]. Organized stress fibers were not significantly affected (Fig. 7d). The changed lamellipodia and filopodia however further suggest that DLGAP4 may influence actin cytoskeleton dynamics, with potentially subtle differences between WT and mutant proteins. To further explore these findings, we performed co-IP experiments with the actin-nucleation-promoting factor cortactin, known to interact with the Dlgap4 interactor, Shank2[45]. Cortactin is an actin-binding protein promoting among other functions, lamellipodia persistence, actin polymerization, and cytoskeletal remodeling during the epithelial–mesenchymal transition. It interacts with AJ components allowing F-actin accumulation, and is an important factor during the breakdown and formation of AJs[46,47]. Cortactin is expressed in progenitors and neurons in the developing neocortex and hippocampus[32,33,48]. As expected, GFP-DLGAP4 was detected in anti-Flag cortactin IPs and vice versa (Fig. 7e–g). We found that the co-IP of Flag-cortactin with the GFP-DLGAP4 mutant was reduced by 54% ($p = 0.0018$) (Fig. 7f). These results together with the abnormalities of F-actin expression and AJ integrity at the VS in the *Dlgap4* KD condition (Fig. 3e–g, Fig. 6a), suggest a role of Dlgap4 in actin cytoskeleton organization and/or dynamics, most probably involving cortactin.

We also investigated how DLGAP4 may further influence cytoskeletal signaling. We assessed whether DLGAP4 may have a role in the downstream mTOR pathway, since it has been described in patients presenting cortical malformations[4,49]. Also, phosphorylated DLG1 impacts PI3K-AKT-mTOR signaling in both the PNS and CNS[50,51]. OE of mutant DLGAP4 in RPE1 cells, showed a mild increase of 20% in the endogenous levels of DLG1 (Fig. 7h, i). DLG1 is targeted by the p38 family of kinases which are known regulators of the actin cytoskeleton[51,52]. Endogenous p38α MAPK was significantly down-regulated after OE of both WT and mutant DLGAP4 (27% and 34%, $p < 0.0001$, respectively). The analysis of a downstream target of the p38 kinase, FLNA, also showed a trend to decrease with expression of WT (10%) and a 12% reduction with mutant DLGAP4 ($p = 0.025$). Furthermore, OE of either WT or mutant DLGAP4 induced a marked increase in Raptor expression (374%, $p = 0.0028$ and 405%, $p = 0.0016$, respectively), involved in mTOR signaling, known to influence the actin cytoskeleton[52]. Interestingly, only mutant DLGAP4 induced a significant decrease in Rictor expression (41%, $p = 0.042$) (Fig. 7h, i). We further tested for the presence of these latter proteins in the same complexes as DLGAP4 by co-transfection of tagged constructs. We found that HA-Raptor was present in the bound fraction with

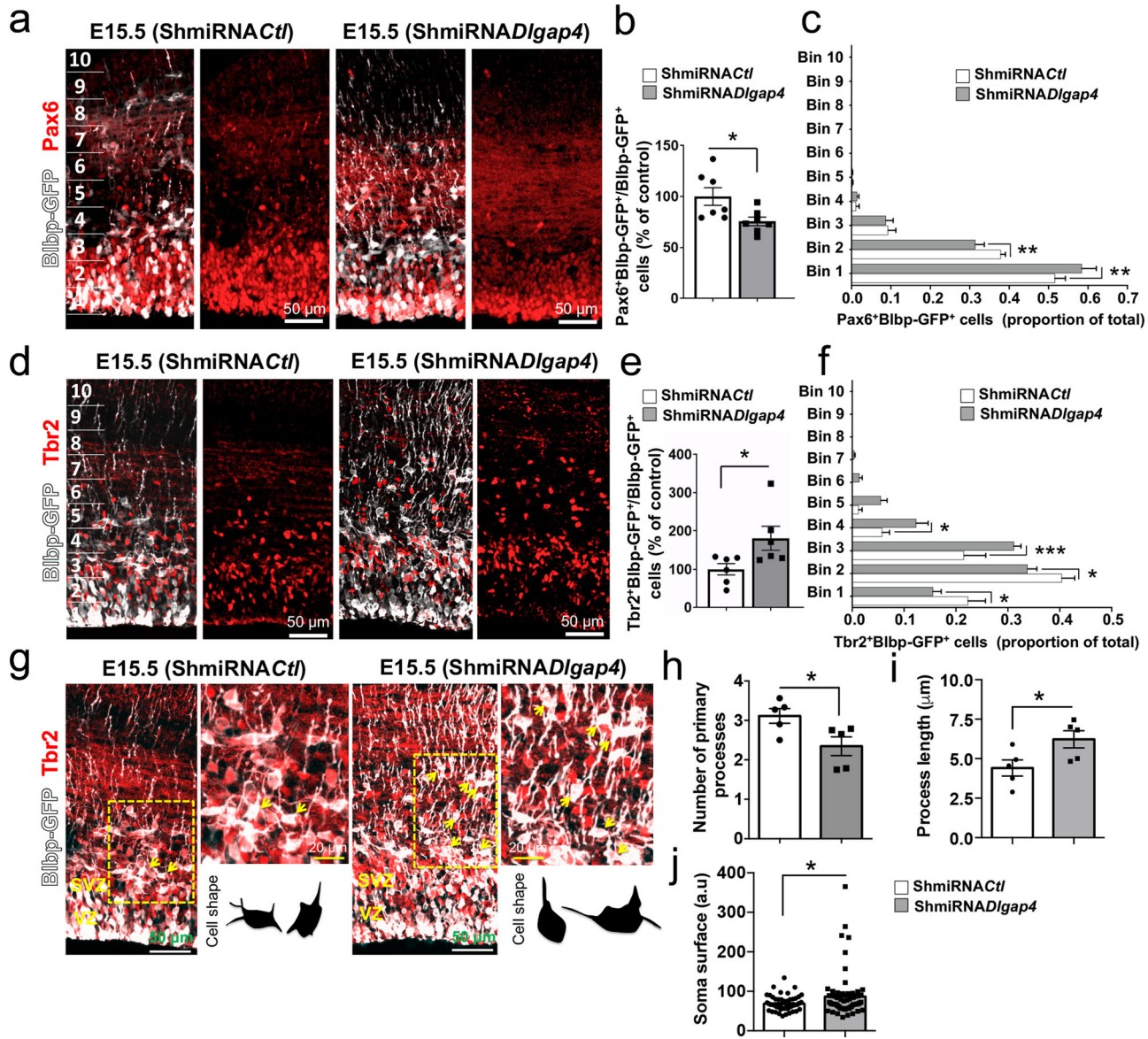

**Fig. 4 A change in progenitor cell dynamics and morphology is observed upon *Dlgap4* KD. a, b** Representative images and quantifications of Pax6+ (red) and Blbp-GFP+ (white) cells in E15.5 brains (*n* = 7 embryos from 3 litters per condition). Means and relativized individual values ± SEM are shown. *\*p* = 0.027 by two-sided unpaired *t* test. **c** Pax6+-Blbp-GFP+ cell distribution along the cortical wall divided into 10 bins. Two-way ANOVA: Interaction Bin × ShmiRNA condition: $F_{9,160}$ = 2.76, *p* = 0.0049; \*\**p* < 0.008 by Sidak's multiple comparison test. **d, e** Representative images and quantifications of Tbr2+ (red) and Blbp-GFP+ (white) cells in E15.5 brains (*n* = 6 embryos from 3 litters per condition). Means and relativized individual values ± SEM are shown. Two-sided unpaired *t*-test, \* *p* = 0.041. **f** Quantification of Tbr2+Blbp-GFP cells along the cortical wall. Analysis was performed by two-way ANOVA followed by Sidak's multiple comparisons test. Interaction Bin x ShmiRNA, $F_{9, 270}$ = 5.13, *p* < 0.0001; \* *p* < 0.04, \*\*\* *p* = 0.0003. **g** Analysis of cellular shape and morphology of Tbr2+Blbp-GFP+ cells in ShmiRNA*Ctl* and ShmiRNA*Dlgap4* KD conditions. Arrows indicate examples of double-labeled Tbr2+Blbp-GFP+ cells in IUE experiments. **h** The number of primary processes, **i** process length, and **j** soma surface, were analyzed in Tbr2+Blbp-GFP+ cells in both ShmiRNA*Ctl* and ShmiRNA*Dlgap4* KD conditions (*n* = 57 ShmiRNA*Ctl* and *n* = 68 ShmiRNA*Dlgap4* KD cells from *n* = 5 embryos per condition). Means and individual values ± SEM are shown. \**p* < 0.05 by two-sided unpaired *t*-test. Images correspond to 10 μm stack projections.

immunoprecipitated WT Flag-DLGAP4 (and vice versa), and bound HA-Raptor showed a reduced trend (31%) specifically with mutant Flag-DLGAP4 (Fig. 7j–l). Rictor however, did not co-IP with either WT or mutant DLGAP4 (data not shown). Altogether, OE of DLGAP4 constructs induces changes in p38 and mTOR pathway expression in vitro, potentially related to actin cytoskeleton modifications engaged in the DLGAP4-dependent phenotypes observed in vivo.

### *Dlgap4* KD and OE leads to anomalies in neuronal migration.
Neurons born after E14.5 are destined for the superficial layers of

the cortex, we hence stained for Cux1, a marker of cortical superficial layers II/III. Fewer cells in the *Dlgap4* KD condition were labeled for Cux1, and the thickness of the Cux1 layers in the CP was also significantly reduced (20%, *p* = 0.018), as was the cell density (32%, *p* = 0.0066) (Fig. 8a-c). Blbp-GFP+ labeling indicated a probable migration defect since many mutant cells were still observed distributed along the cortical wall at P0 (Fig. 8d). In addition, proportions of GFP+ cells were also ectopically located at P8 forming abnormal clusters of cells in both *Dlgap4* KD and *DLGAP4* mutant OE (Supplementary Fig. 9). The identity of these cells was confirmed by co-immunolabelings either with

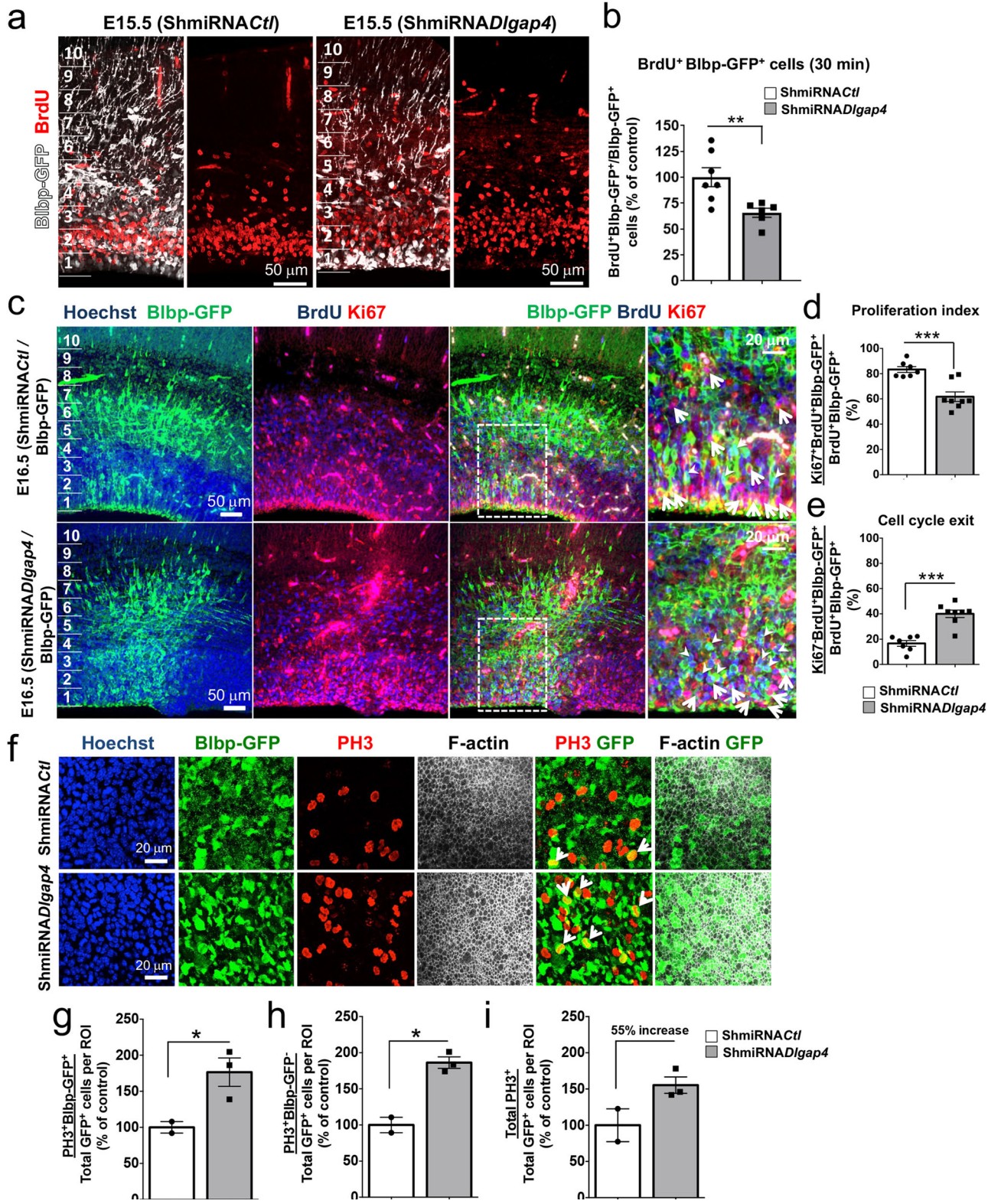

Cux1 or Ctip2 (marker for cortical deep layers), showing the presence of many Ctip2[+] GFP[+] cells (arrowheads) in the heterotopia-like region (see Supplementary Fig. 9). The Cux1[+] cell layer also appeared to be reduced in thickness at P8 in mutant conditions, likely due to a combination of proliferation, apoptosis, and migration defects (Supplementary Fig. 9).

To more specifically study neuronal defects, we used a Dcx promoter-GFP reporter plasmid co-electroporated with a similarly

neuron-specific overexpressed WT or mutant *DLGAP4* (pDcx-*DLGAP4*-WT or –MUT -Flag-ires-GFP) construct (Fig. 8e). Four days after electroporation, a delay in neuronal migration was found in both DLGAP4 WT and MUT OE conditions along the cortical wall ($p < 0.0001$) (Fig. 8f, g). An increased proportion of Dcx-GFP[+] cells was found in bins 1–2 and less cells reached superficial bins 8–9 in both conditions compared to control (Fig. 8f, g). Cux1[+]Dcx-GFP[+] cells were identified showing that these neurons could adopt

**Fig. 5 Cortical *Dlgap4* KD leads to reduced proliferation, increased cell cycle exit, and increased proportion of PH3+ cells at the ventricular surface.**
**a**, **b** Representative images and quantifications of BrdU+ (red) and Blbp-GFP+ (white) cells after a 30 min pulse of BrdU at E15.5 ($n = 7$ ShmiRNA*Ctl* and $n = 6$ ShmiRNA*Dlgap4* KD embryos from 3 litters per condition). Mean and relativized individual values ± SEM are shown. **$p = 0.0084$ by two-sided unpaired *t*-test. **c** Representative images showing BrdU (blue), Ki67 (red) and Blbp-GFP (green) immunostainings in ShmiRNA*Ctl* and ShmiRNA*Dlgap4* KD conditions. Arrows: triple labeled Ki67+BrdU+Blbp-GFP+ cells, arrowheads: double-labeled BrdU+Blbp-GFP+ cells. **d**, **e** Quantification of the proliferation index and cell cycle exit, 48 h after IUE. Proliferation index was evaluated as the ratio of Ki67+BrdU+ and BrdU+Blbp-GFP+ immunolabelings. Cell cycle exit was analyzed as the ratio of Ki67−BrdU+ and BrdU+Blbp-GFP+ immunolabelings ($n = 7$ ShmiRNA*Ctl* and $n = 8$ ShmiRNA*Dlgap4* embryos from 3 litters per condition). Quantification data represent the individual values expressed as raw %, means ± SEM, ***$p < 0.0005$ by two-sided unpaired *t*-test. **f** Representative *en face* confocal imaging of ShmiRNA*Ctl* and ShmiRNA*Dlgap4* electroporated brains (E16.5, 48 h after IUE). Arrows indicate PH3+ (red) and Blbp-GFP+ (green) double-labeled cells. **g** Quantification of PH3+Blbp-GFP+/Total Blbp-GFP+ per ROI, **h** PH3+Blbp-GFP−/Total Blbp-GFP+ per ROI localized <10 μm distance from a PH3+Blbp-GFP+ cell and **i** total PH3+/Total Blbp-GFP+ cells per ROI expressed as % of control. ROI: 246.27 μm × 246.27 μm. Data are represented as relativized individual values, mean ± SEM. Two-sided unpaired *t* test with Welch's correction was performed (**g**–**i**), *$p < 0.05$; n.s: $p = 0.11$. No obvious ventricular surface damage was observed. ROI: Region Of Interest; *n.s*: not significant.

an appropriate identity. Total Dcx-GFP+ Cux1+/GFP+ cells did not change among conditions ($p > 0.05$) (Fig. 8h). However, similar results of retarded migration were found by analyzing Dcx-GFP+/Cux1+ cells along the cortical wall ($p < 0.0001$) (Fig. 8i, j). Overall, specific OE of *DLGAP4* constructs in neurons using the Dcx promoter revealed perturbed migration, while the total number of Cux1+ cells seems to be reduced only when Dlgap4 expression is also impaired (mutated or KD *Dlgap4*) in progenitors.

***Dlgap4* heterozygous KO mice show neuroanatomical defects.**
To further study the role of Dlgap4 during brain development, we obtained mouse mutants from the International Mouse Phenotyping Consortium produced using the knockout-first allele method[53] (see Supplementary information, Supplementary Fig. 10a). The expected number of WT and heterozygous mice was observed, but no homozygous animals, suggesting that the double dosage of the mutant allele of *Dlgap4* is not viable. Using a recently developed robust approach for the assessment of 40 brain parameters across 22 distinct brain regions[54,55], we analyzed neuroanatomical defects in adult *Dlgap4* heterozygous mice (Supplementary Fig. 10b, c, Supplementary information). A number of brain structures showed significantly decreased size in heterozygotes when compared to WTs (Supplementary Fig. 10b). The total brain area was reduced by 22% ($p = 0.0004$), with especially decreased size of the hippocampus (29%, $p = 0.00005$) (Supplementary Fig. 10d), the corpus callosum (24%, $p = 0.0009$), the anterior commissure (29%, $p = 0.00006$), the thalamus (15%, $p = 0.018$) and the neocortex (10%, $p = 0.0004$) compared to WTs. In order to determine which cortical layers were most affected, the cortex was divided into four bins corresponding to cortical layers I, II–IV, V, and VI (Supplementary Fig. 10e). We found that bins 2–4 were reduced in size when compared to WT, including layers II–IV (bin 2) 28%, $p = 0.03$ and layer VI (bin 4), 16%, $p = 0.05$ (Supplementary Fig. 10f). Finally, cell counts across the entire brain were decreased by 20% ($p = 0.03$) (Supplementary Fig. 10g). Despite the absence of heterotopia-like phenotypes in this chronic heterozygous KO model, at least at the lateral +0.60 mm level studied in unilateral sagittal brain sections, these results strongly suggest that Dlgap4 plays a role in determining the size of several telencephalic structures.

## Discussion

Identifying *DLGAP4* mutations associated with cortical malformations, we link this well-known synapse-related gene to functions in neuronal progenitors and migrating neurons during cortical development. Searching to explain certain pathogenic mechanisms associated with subependymal and subcortical heterotopia, we provide evidence that Dlgap4 interacts with proteins mediating VS integrity, influences the F-actin cytoskeleton, as

well as progenitor dynamics and proliferation. Indeed, we observed a higher proportion of Pax6+Blbp-GFP+ and Ki67+Blbp-GFP+ cells accumulating at the VS, although these are overall reduced in *Dlgap4* KD brains. Since an *en face* view of PH3+Blbp-GFP+ cells showed increased numbers in *Dlgap4* KD brains, this may suggest defects in mitotic progression, including lengthened mitosis. These results could be in agreement with a higher cell differentiation[56,57], although we do not exclude that RGs have difficulties reaching basal positions of the VZ during G1. Moreover, G1 lengthening has been associated with the transition from stem cell-like apical progenitors to fate-restricted intermediate progenitors[58]. Thus, our combined results of increased cell numbers in apical regions may be coherent with aberrant fates.

We further show a cell-autonomous role of Dlgap4 in migrating neurons. The latter could be due to multipolar to bipolar transition defects or slowed neuronal migration[59]. In our study, we decided to focus instead on the exact features of the RG phenotype, affecting proliferation and the substrate for radial neuronal migration, since this appeared less well known. Furthermore, homozygous KO mice die during embryonic development and heterozygous mutants show reduced brain size, also suggesting a key role of *Dlgap4* in progenitor cells during neurodevelopment. Reconstructing molecular mechanisms, we show that DLGAP4 is found in protein complexes with key RG polarity (e.g., DLG1, β-catenin) and spindle orientation (e.g., LGN, EML1) proteins, other scaffold family members (DLGAP1), as well as with actin cytoskeleton-regulating partners (e.g., cortactin). Importantly, DLGAP4 mutation perturbs these interactions. In addition, we identify a potential impact on cytoskeletal associated p38 kinase and mTOR pathways, which could similarly contribute to these mechanisms. These combined data support the unsuspected role of DLGAP4 in the VZ, as well as in migrating neurons, highlighting similarities with FLNA pathways[11], and further implicating actin cytoskeletal defects in this form of heterotopia.

DLGAP4 belongs to the SAPAP family, which has been described as one of the first components to reach neuronal synapses, involved in the re-localization of PSD-95 from the cytoplasm to the plasma membrane[25,26,35]. Our study suggests scaffolding roles in cortical progenitors and migrating neurons. As well as identifying a C-terminal heterotopia mutation (family P616), we also identified one further cortical malformation family (P477) showing complex gene variations involving both *DLGAP4* and *DLGAP1*. Concerning *DLGAP4*, the P477 twin siblings notably show a C-terminal (p.Ser965Ala) variation, as well as exhibiting an occipital malformation, similar to patient P616-3. The DLGAP4 protein sequence is highly conserved in the regions of the C-terminal disruptions identified here[25], and no similar variations affecting these residues have been detected in the

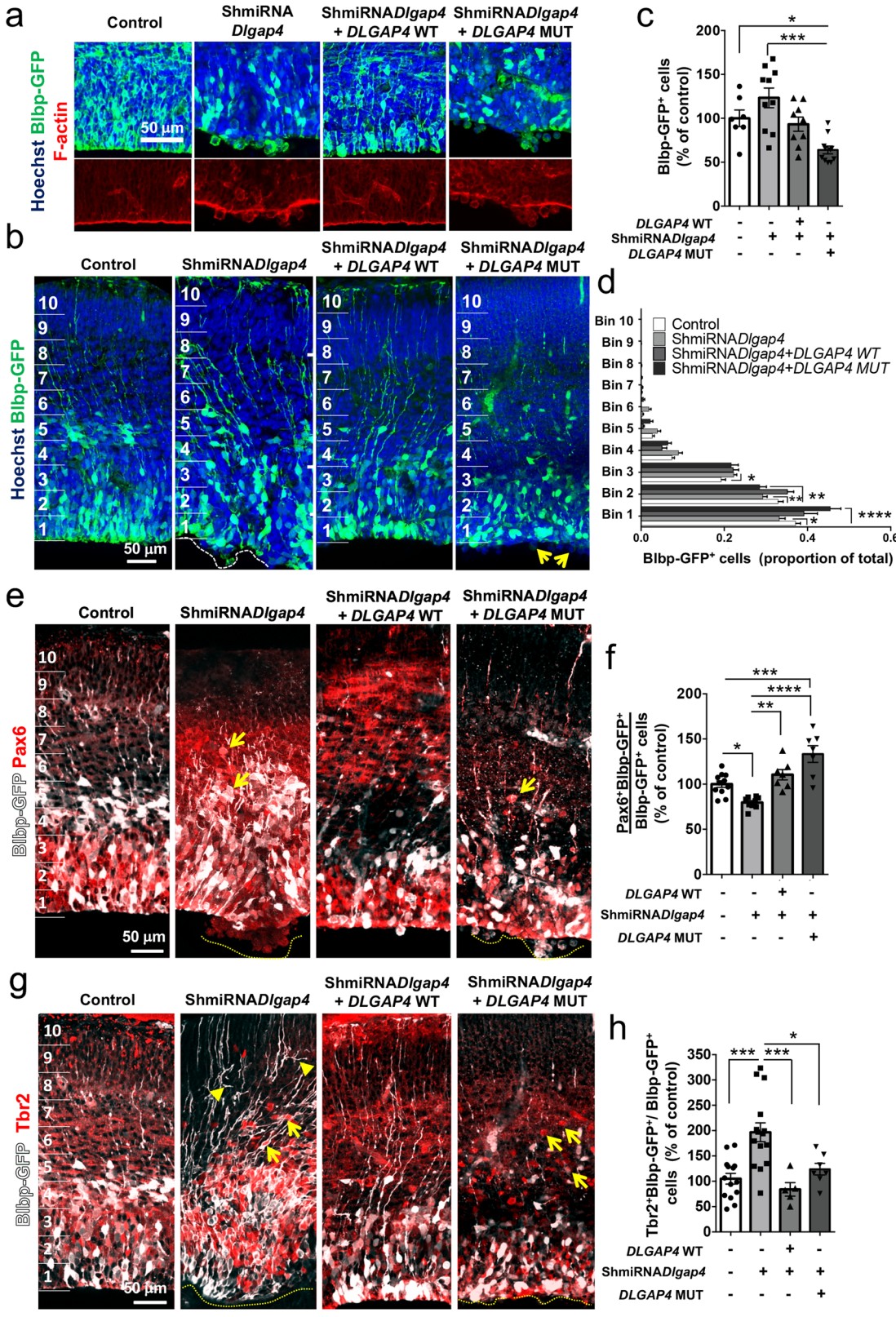

normal population[60]. It is thus likely that variations in this particular region of DLGAP4 give rise to the cortical malformations observed. Increased levels of DLGAP4 have previously been associated with early-onset cerebellar ataxia[61]. We show here that experimental OE in the developing mouse dorsal cortex can also produce a phenotype. Although it is also known that some proteins present gradient expression patterns in the neocortex[62], our

data in mouse suggest that Dlgap4 is largely ubiquitous. We cannot rule out gradients in human brain. We also speculate that different areas of the developing brain may have variable susceptibilities to perturbation, related to levels of mechanical tension and proliferation/migration characteristics. Of the mouse heterotopia models known, most seem to give rise to heterotopia in caudo-medial regions above the hippocampus[40,63–65], although

**Fig. 6 Restoration of normal corticogenesis after OE of WT and mutant *DLGAP4* concomitantly with *Dlgap4* KD. a** Representative ventricular surface images showing F-actin (red) staining 24 h after electroporation. Mutant *DLGAP4* does not prevent the ventricular phenotype. Blbp-GFP: green; Hoechst: blue. **b** Representative images of OE experiments in *Dlgap4* KD mice showing Blbp-GFP$^+$ cell distribution and the ventricular surface phenotype (dashed lines and arrows). OE: overexpression. **c** Quantification of Blbp-GFP$^+$ cells among conditions ($n = 7$ control, $n = 10$ ShmiRNA*Dlgap4*, $n = 9$ OE WT *DLGAP4* and $n = 11$ OE MUT *DLGAP4*, from 3 litters per condition). Quantification data represent the relativized individual values, mean ± SEM. One-way ANOVA: $F_{3,33} = 9.35$, $p = 0.0001$; *$p = 0.033$, ***$p < 0.0001$ by post hoc Tukey's test. **d** Blbp-GFP$^+$ cell distribution is restored only by OE of *DLGAP4* WT in bins 1 and 2. Two-way ANOVA: Interaction Bin x Experimental condition: $F_{27,450} = 5.67$, $p < 0.0001$; *$p = 0.0049$, **$p < 0.009$, ***$p < 0.001$ by post hoc Sidak's test. **e** Representative images of OE experiments in *Dlgap4* KD mice showing Blbp-GFP (white) and Pax6 (red) labeling. **f** Quantification of double Pax6$^+$Blbp-GFP$^+$ cells among conditions ($n = 10$ control, $n = 7$ ShmiRNA*Dlgap4*, $n = 7$ OE WT *DLGAP4* and $n = 7$ OE MUT *DLGAP4*, from at least 3 litters per condition). Mean and relativized individual values ± SEM are shown. One-way ANOVA: $F_{3,29} = 17.77$, $p < 0.0001$; *$p = 0.014$, **$p = 0.0015$, ***$p = 0.0009$, ****$p < 0.0001$ with post hoc Tukey's test. **g** Representative images of OE experiments in *Dlgap4* KD mice showing Blbp-GFP and Tbr2 labeling. **h** Quantification of double Tbr2$^+$ (red) and Blbp-GFP$^+$ (white) cells among conditions ($n = 14$ control, $n = 15$ ShmiRNA*Dlgap4*, $n = 5$ OE WT *DLGAP4* and $n = 7$ OE MUT *DLGAP4*, from at least 3 litters per condition). Mean and relativized individual values ± SEM are shown. One-way ANOVA: $F_{3,37} = 10.06$, $p < 0.0001$; *$p = 0.022$, ***$p < 0.001$ with post hoc Tukey's test. In **e** and **g**, arrows show basal Pax6$^+$ or Tbr2$^+$ cells; arrowheads show abnormal basal RG processes labeled with Blbp-GFP$^+$.

exact reasons for this are to our knowledge currently unknown. All in all though, it is clear that appropriate levels of DLGAP4 are required for correct brain development.

Subependymal nodular heterotopias have been described in patients with chromosomal rearrangements, as well as associated with intragenic gene mutations (e.g., in *FLNA*, *ARFGEF2*, *DCHS1*, *FAT4*, *ERMARD*, and *NEDD4L*). Certain animal models have also revealed a disrupted VS[5,6], although most of the genes involved do not show mutations in human patients, with *FLNA* being a notable exception. Indeed, *FlnA*[11], *aPKCλ*[65], *Cdc42*[66], *RhoA*[40], *Mekk4*[67], *α-E-catenin*[41], and *Llgl1* (*Lethal Giant Larvae Homolog 1*)[68] conditional (cKO) or constitutive KO mouse models, show VS defects. Focusing on this phenotype, we have shown that Dlgap4, also an intracellular protein important for VS integrity, is present in the same protein complexes as β-catenin and the actin binding protein cortactin. β-catenin is known to regulate AJs and to interact with α-catenin and the actin cytoskeleton[20] and cortactin is important to stabilize the cadherin/catenin complex[69]. Thus, these proteins are critical for apical neuroepithelial integrity. AJs are localized basolateral to the apical endfoot of RGs, and are connected to the F-actin belt, allowing cell-to-cell anchoring, which helps form and maintain the VS. With *Dlgap4* mutation we showed F-actin downregulation, as well as abnormalities in the expression and distribution of β-catenin, but also the AJ protein N-cadherin. Murine models, including *α-E-catenin*[41], *Llgl1*[68,70], *N-cadherin*[71], and *RhoA* cKO mice[40] also pinpoint AJs as critical to maintain VS integrity[6,40,41,68,70,71]. Thus, AJ abnormalities go together with actin cytoskeleton disruption, coherent with *Dlgap4* mutant phenotypes in the mouse (KD and OE).

A further link to adhesion is suggested by the presence of rosette-like structures in DLGAP4 OE brains. These structures are also observed in *Llgl1*[70], *N-cadherin*[71], and *RhoA* cKO telencephali[40] as well as being described in other human brain pathologies[72,73]. It is clear that intracellular molecules (e.g., scaffold and signaling proteins) are required to regulate adhesion proteins and help to anchor them to the actin cytoskeleton. Related to this, cortactin is an actin crosslinking protein[69,74] as is FlnA, involved in cytoskeleton remodeling and connection to the plasma membrane[11,75]. The functional similarities between Dlgap4 and these proteins may suggest that this scaffold protein directly functions as a mediator for actin cytoskeleton remodeling. Indeed, our data suggest a role for DLGAP4 regulating actin dynamics in RPE1 cells. Importantly, compromised F-actin integrity, leading to a subependymal heterotopia-like phenotype at the VS, was restored by DLGAP4 WT protein and not by mutant DLGAP4 (P616-3 mutation), potentially suggesting a role for the C- terminus in these mechanisms.

In non-neuronal cells, we show that DLGAP4 OE in vitro leads to the downregulation of endogenous p38, a downstream target of Mekk4 and a regulator of FlnA[67]. When Mekk4 is disrupted in the mouse, leading to a VS phenotype, FlnA phosphorylation is altered[67]. p38 kinase activity is also altered after Mekk4 inactivation and this was shown to contribute to actin cytoskeletal breakdown[76,77]. These data suggest that correct levels of Dlgap4 are not only important for the cortactin function on actin remodeling, but also for pathways involving Mekk4 and p38, regulating FlnA, and thus the stability of the actin cytoskeleton at the VS, and potentially also in migrating neurons.

A number of investigations have demonstrated the essential role of AJ components in RG proliferation, as well as in maintaining their morphology, polarity, and localization, important for correct cortical development[12,19,41,70,71]. Alterations in RG morphology were found in *Dlgap4* KD brains. Moreover, Tbr2$^+$Blbp-GFP$^+$ cells present an abnormal morphology, showing an increased soma size and a decreased number of primary processes. We found that the OE of either the WT or mutant constructs was able to restore control Tbr2$^+$Blbp-GFP$^+$ cell values. These results may suggest that the N-terminal domain of the protein (WT and mutant) is involved in the molecular mechanisms leading to cell differentiation. In addition, DLGAP4 interactors LGN and DLG1, are both involved in cell polarity and spindle orientation in neuronal progenitors[19,39,78,79] and EML1 has also been linked to these phenomena[21]. Together, these results may also suggest a scaffold role for Dlgap4 in the maintenance of spindle orientations, as well as apical surface integrity, which when perturbed may help explain the disruptions to highly regulated progenitor dynamics, as well as the non-cell-autonomous effects identified by *en face* imaging.

Finally, the increased Raptor (subunit of mTORC1 pathway) and decreased Rictor (subunit of mTORC2 pathway) protein levels observed when mutant *DLGAP4* was transfected in RPE1 cells, indicates an imbalance of these mTOR pathways. Raptor, but not Rictor, was immunoprecipitated with both WT and mutant DLGAP4. Since the mTORC2 pathway plays a role in actin remodeling[80], it could be quite relevant that mutant DLGAP4 decreases Rictor levels. Recently, a human mutation in *RHEB*, a RAS family of small GTPases and a direct activator of mTORC1, showed severe focal cortical lesions, resembling subependymal nodular heterotopia[81]. In addition, persistent activation of mTORC1 induced neuronal misplacement in the cerebral cortex and functional changes in axonal connectivity leading to increased excitability and generalized seizures[81]. Moreover, the DLGAP4 interactor DLG1 also plays a role as a signaling molecule in this pathway, as shown in other CNS processes[50,51]. Changes in mTOR signaling have previously been associated with

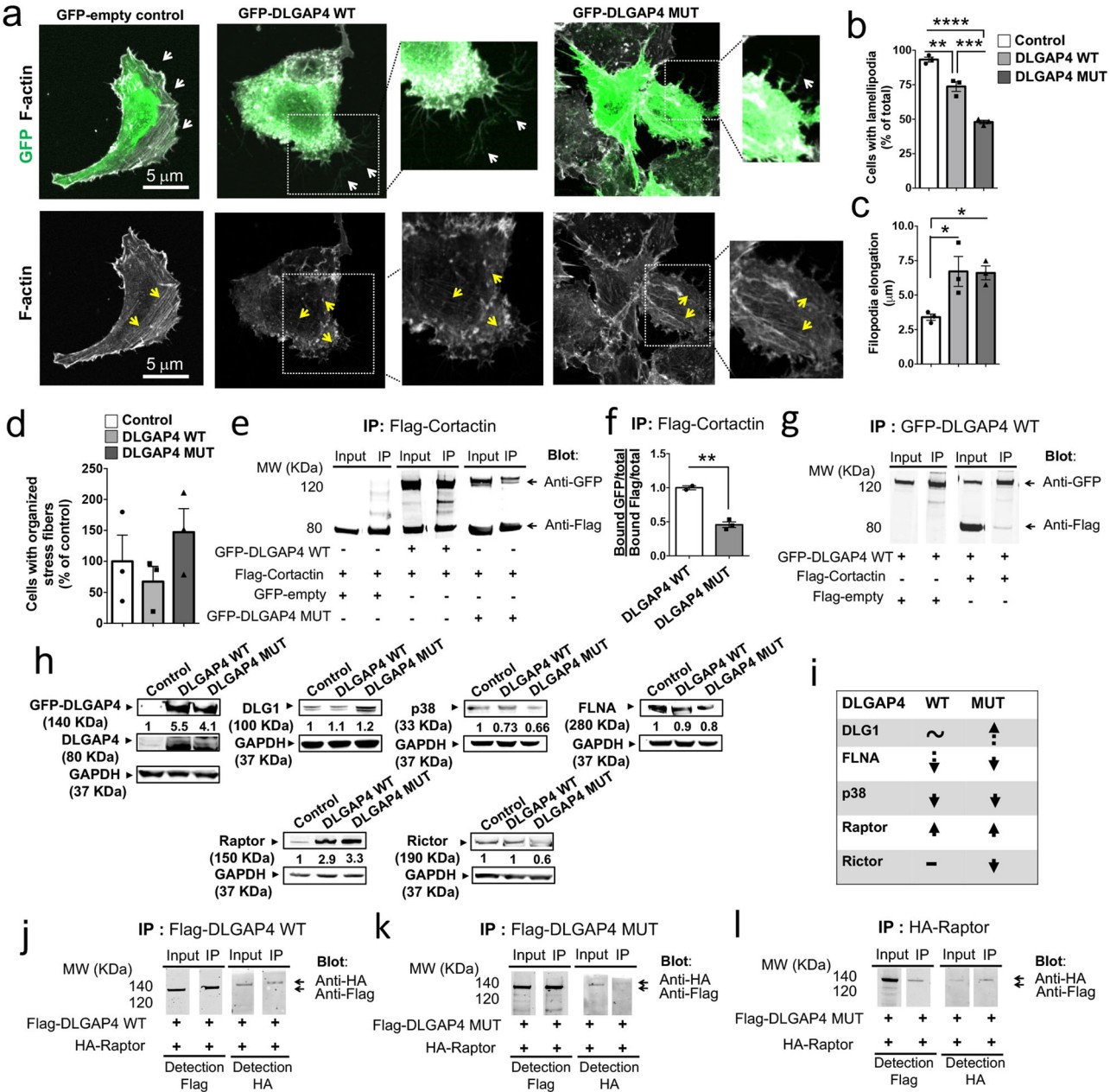

**Fig. 7 *DLGAP4* OE impacts actin cytoskeleton dynamics in vitro and may influence the downstream mTOR pathway. a** Representative RPE1 cell images showing reduced lamellipodia morphology and increased filopodia elongation (white arrows) after WT or mutant *DLGAP4* transfections. Yellow arrows: F-actin. **b** Quantifications of cell percentages with lamellipodia morphology, **c** filopodia elongation (μm) and **d** organized stress fibers. Quantification data (**b–d**) represents the individual values, mean ± SEM (~100 cells per condition, $n = 3$ independent experiments). One-way ANOVA with post hoc Tukey's test was performed. **b** $F_{2,6} = 88.8$, $p < 0.0001$; **$p = 0.0029$, ***$p = 0.0007$, ****$p < 0.0001$. (**c**) $F_{2,6} = 7.21$, $p = 0.025$; Ctl vs WT DLGAP4: *$p = 0.036$ and Ctl vs MUT DLGAP4: *$p = 0.041$. **d** $F_{2,6} = 1.25$, $p = 0.35$. **e** IP performed from Neuro2A cells co-transfected with Flag-cortactin and GFP-*DLGAP4* (WT or MUT). **f** Quantification data represent the relativized individual values, means ± SEM ($n = 2$ WT; $n = 3$ MUT), ** $p = 0.0018$, two-sided unpaired t-test with Welch's correction. **g** Flag-cortactin was found in GFP-DLGAP4 IPs. **h** Western blot assays from RPE1 cell extracts transfected with WT or MUT GFP-DLGAP4. Endogenous levels of DLG1, p38, FLNA, Raptor and Rictor were detected and normalized to GAPDH. **i** Table summarizing results. Dashed arrows: trend. One-way ANOVA with post hoc Tukey and at least three independent experiments were performed. DLGAP4 ($n = 5$): $F_{2,12} = 6.51$, $p = 0.012$; control vs WT, $p = 0.011$, control vs MUT, $p = 0.079$. DLG1 ($n = 3$): $F_{2,6} = 0.7$, $p = 0.53$. p38 ($n = 3$): $F_{2,6} = 147.3$, $p < 0.0001$; control vs WT and control vs MUT, $p < 0.0001$. FLNA ($n = 4$): $F_{2,9} = 5.25$, $p = 0.031$; control vs WT, $p = 0.26$; control vs MUT, $p = 0.025$. Raptor ($n = 3$): $F_{2,6} = 25.18$, $p = 0.0012$; control vs WT, $p = 0.0028$, control vs MUT, $p = 0.0016$. Rictor ($n = 3$): $F_{2,6} = 6.49$, $p = 0.032$; control vs WT, $p = 0.98$, control vs MUT, $p = 0.042$. **j–l** HA-Raptor and WT or MUT Flag-DLGAP4 co-IP in Neuro2A cells ($n = 2$).

diverse pathological cortical phenotypes[82,83], including sub-ependymal heterotopia and neuronal abnormalities[42,84]. We hence suggest that DLGAP4 is involved in similar mechanisms, although these results are still preliminary and further experiments, including in vivo confirmation, would help corroborate this hypothesis.

DLGAP4, as other SAPAPs, was shown to link glutamate receptors with other receptors, signaling molecules and the

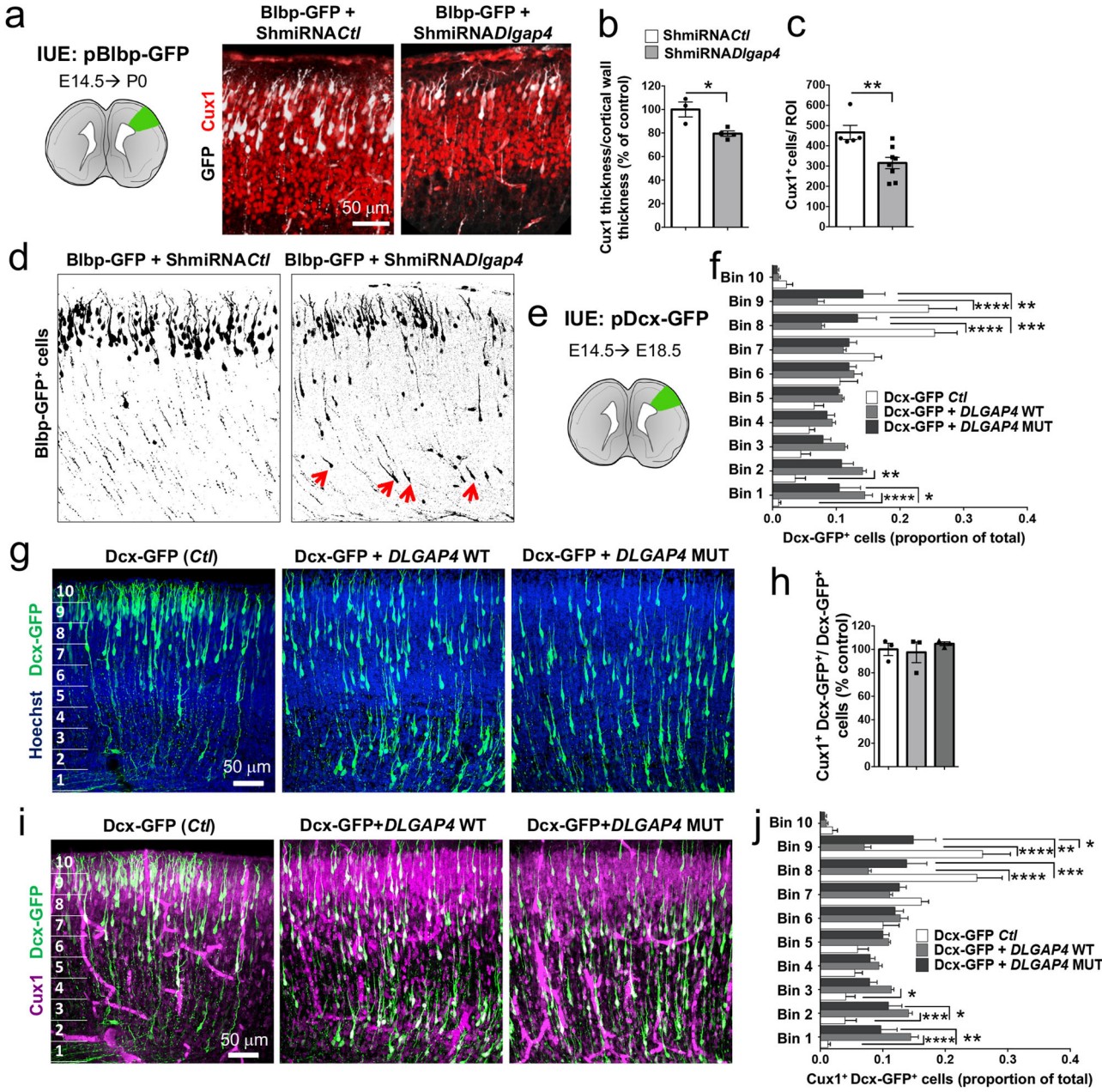

**Fig. 8 *Dlgap4* KD or OE leads to slowed neuronal migration. a** Left: Schematic representation of the electroporated ROI at P0. Right: Representative Cux1 (red) labeling and Blbp-GFP (white) in Ctl and *Dlgap4* KD brains at P0. **b** Cux1 thickness relativized to cortical wall thickness ($n = 3$ Ctl, $n = 4$ *Dlgap4* KD brains from 2 litters per condition). Quantification data represent relativized individual values, mean ± SEM (two-sided unpaired *t*-test, *$p = 0.018$). **c** Total Cux1$^+$ cells per ROI ($n = 5$ Ctl, $n = 8$ *Dlgap4* KD brains from 2 litters per condition). ROI: 135 μm height × 370 μm width. Quantification data represent individual values, mean ± SEM (two-sided unpaired *t*-test, **$p = 0.0066$). **d** GFP$^+$ cells are still migrating below the CP in the KD at P0 (e.g., red arrows). **e** Schematic representation of the electroporated ROI at E18.5. A Dcx-GFP reporter promoter vector was co-electroporated with a pDcx-*DLGAP4* WT-Flag-ires-GFP or pDcx-*DLGAP4* MUT-Flag-ires-GFP construct. **f** Quantification of Dcx-GFP$^+$ cells along the cortical wall divided in 10 bins. Two-way ANOVA followed by Sidak's test was performed ($n = 3$ brains from 2 litters per condition). Relativized mean values ± SEM are shown. Two-way ANOVA, Interaction Bin x OE condition, $F_{9, 60} = 16.11$, $p < 0.0001$; *$p = 0.014$, **$p < 0.005$, ***$p = 0.0004$, ****$p < 0.0001$. **g** Representative images of OE experiments targeting specific neuronal migration after IUE showing Dcx-GFP$^+$ (green) cell distribution and Hoechst staining. **h** Total Cux1$^+$ Dcx-GFP$^+$/ Dcx-GFP$^+$ cells did not change along conditions. Relativized mean and individual values ± SEM are shown. One-way ANOVA followed by Tukey's test, $F_{2, 6} = 0.39$, $p = 0.70$. **i** Representative images of OE experiments showing Cux1$^+$ (magenta) and Dcx-GFP$^+$ (green) cell distribution. **j** Quantification of Cux1$^+$ Dcx-GFP$^+$/Dcx-GFP$^+$ cells along the cortical wall divided in 10 bins. Two-way ANOVA followed by Sidak's test was performed ($n = 3$ from 2 litters per condition). Relativized mean values ± SEM are shown. Two-way ANOVA, Interaction Bin × OE condition, $F_{9, 60} = 16.63$, $p < 0.0001$; *$p < 0.05$, **$p < 0.005$, ***$p < 0.001$, ****$p < 0.0001$.

cytoskeleton at the postsynaptic membrane[25,26]. Previous KO mouse models for SAPAP family members have mostly focused on synapse architecture and function, with mutants revealing autism-like phenotypes and cognitive impairment[85]. C-terminal mutant DLGAP4 shows altered binding to multiple structural and signaling proteins, suggesting it may be a key scaffolding point in RG morphogenesis and neuronal migration. We hence emphasize that 'synaptic' genes and associated protein complexes can be involved in wider mechanisms than previously thought, and may be considered as candidates for a larger spectrum of neurological disorders. We provide here evidence of a crucial role of DLGAP4 maintaining the progenitor pool, regulating neurogenesis, as well as influencing migration. This work hence enlarges the pathological spectrum for a single gene, and helps to identify possible molecular pathways in RGs involved in this phenotype, adding important mechanistic insights to the understanding of pathogenic mechanisms associated with overlapping phenotypes of subependymal and subcortical heterotopia.

## Methods

**Ethics**. We confirm that our research complies with all relevant ethical regulations. Animal research in Paris was specifically approved by the local ethical committee (Charles Darwin committee Paris, France), for the French Ministry of Higher Education, Research and Innovation (MESRI), and for the Dlgap4 knockout in accordance with UK Home Office regulations, UK Animals (Scientific Procedures) Act of 1986, approved by the UK Home Office and reviewed by the Wellcome Sanger Institute Animal Welfare and Ethical Review Body.

Genetic research and whole exome sequencing of patient DNAs were conducted with approval by the Imagine Institute, Paris, following guidelines provided by French APHP-Délégation Interrégionale à la Recherche Clinique and the French Ministry of Health.

**Animals (Mus musculus)**. Research was conducted according to national and international guidelines (EU directive 2010/63), with protocols followed by local ethical committees (e.g., Charles Darwin committee Paris, France, French MESRI 00984.02). Timed-pregnant wild-type SWR/J mice provided by Janvier Labs (https://www.janvier-labs.com/) were used for in utero electroporation experiments as well as for primary progenitor cultures and in situ hybridization. Both males and females were used for all experiments. For each in utero electroporation experiment, we routinely use 54 animals (6 pregnant females and 6 litters of 8 embryos or pups). Ten further embryos were used for expression studies. For the KO model, further information are provided in Supplementary Information. For staging of embryos, the day of vaginal plug was considered E0.5. Mice were housed with a standard 12 h light/dark schedule (lights on at 07:00 a.m.).

**Plasmids**. Full-length human *DLGAP4* (transcript variant 1, RefSeq NM_014902.4, GenScript) was subcloned by restriction enzyme digest into the p3XFlag-CMV10 vector (Sigma-Aldrich) and into the pEGFP-C3 vector (Clontech) with the tag N-terminal to *DLGAP4*. *DLGAP4* site-directed mutagenesis was performed using a QuikChange Lightning kit (Agilent) following manufacturer's guidelines. In addition, using the mouse pEGFP-C1-GKAP vector, Dlgap1 Asp773Gly (equivalent to Asp466Gly mutation in human *DLGAP1*) was obtained using Site-Directed Mutagenesis kit (Agilent) following manufacturer's guidelines. Sequences were verified via Sanger sequencing (Genewiz). Full-length *Eml1* WT or expressing a patient mutation (family P135, mutation T243A) were cloned into the p3XFlag-CMV10 or pEGFP-C3[21]. Full-length chicken pEGFP-C1-DLG1 (RefSeq XM_422701) and mouse pEGFP-C1-LGN (RefSeq NM_029522.2) constructs were generously provided by X. Morin (IBENS-ENS, Paris). The pBlbp-EGFP construct containing 1.6 Kb Blbp promoter region was obtained from the N. Heintz laboratory (Rockefeller University, New York). The mouse pEGFP-β-catenin was kindly provided by Dr. C. Perret (Institut Cochin, Paris). *DLGAP4* PCR products were subcloned into pDcx-IRES-GFP downstream of the Dcx promoter using an In-Fusion kit (Thermoscientific) following the manufacturer's protocol. Positive clones were sequence verified using Sanger sequencing. The human c-myc-CMV-Rictor (#11367), HA-CMV-Raptor (#8513), mouse pcDNA3-Flag-cortactin (#74476) were purchased from Addgene. Following gene nomenclature guidelines (https://www.genenames.org/about/guidelines/), we use upper case for the human version of proteins and lower case for the murine version throughout the manuscript.

**Neuro2A cell culture, transfection, and co-immunoprecipitations**. Neuro2A cells (ATCC® CCL-131™) were maintained in Dulbecco's Modified Eagle Medium (DMEM-low glucose) (31885-023, Thermo Fisher) supplemented with 10% heat-inactivated fetal bovine serum (FBS), 100 U/mL penicillin and 100 µg/mL streptomycin (Gibco). Cells were cultured at 37 °C in a humidified atmosphere of 5%

$CO_2$–95% air, and the medium was renewed three times a week. For all experiments, Neuro2A cells were dissociated with 0.025% trypsin–EDTA (Gibco), diluted in DMEM, and $1 \times 10^6$ cells plated into 100 mm Petri dishes. After 24 h in culture, cells reaching ~70% confluence were transfected in fresh DMEM containing 10% FBS.

For co-immunoprecipitation (co-IP) studies, Neuro2A cells were transfected using polyethylenimine with 3 µg DNA of each vector p3XFlag-CMV10- or pEGFP-C3-DLGAP4 WT or mutant; p3XFlag-CMV10- or pEGFP-C3-EML1; CMV-3xFlag-EML1_T243A (mutant); pEGFP-C1-LGN; pEGFP-C1-DLG1; pEGFP-β-catenin; pcDNA3-Flag-cortactin; c-myc-CMV-Rictor; HA-CMV-Raptor; pEGFP-C1-GKAP, also including all control "empty" vectors. After 48 h, cells were washed with ice-cold PBS and lysed by rotation for 20 min at 4 °C in RIPA buffer (50 mM Tris/HCl, pH 8, 100 mM NaCl, 1 mM EDTA, 0.1% SDS, 1% Nonidet P-40, 0.5% sodium deoxicholate) and protease inhibitor mix 1× (Complete Protease Inhibitor Cocktail Tablets EDTA-Free, Roche). After centrifugation at 15,000 g for 15 min, supernatants of cell lysates were incubated overnight at 4 °C with indicated antibodies (Flag, GFP, or c-myc) bound to protein G-Sepharose (Sigma Aldrich). Immunoprecipitated samples were collected by centrifugation and extensively washed with RIPA buffer. Immunoprecipitated proteins were eluted with NuPAGE buffer (Invitrogen) and analyzed by 4–12.5% SDS-PAGE following western blot standard procedures using specific antibodies (see below). The bound protein was normalized to total levels of the protein and to the amount of immunoprecipitated bait protein. Co-IP experiments were performed in both directions to confirm protein interactions. At least two or three independent experiments were performed per condition.

**Western blot**. Denatured samples were prepared in 2× NuPAGE LDS Buffer (Thermo Fisher) heated for 10 min at 70 °C. Protein samples corresponded to the total fraction (input), unbound (not shown) and bound to the immunoprecipitated protein (IP: DLGAP4 WT, DLGAP4 mutant, EML1, DLG1, LGN, Raptor, Rictor, β-catenin, cortactin, Dlgap1). Ponceau dye staining was performed to verify protein transfer to the membrane. Non-specific binding sites were detected incubating the membranes with the secondary antibody, incubated in the dark for 45 min, followed by extensive washes with TBST (Tris 100 mM, NaCl 150 mM, pH 7.5, ethanol 10%). The primary antibodies were incubated O/N at 4 °C in agitation. The following primary antibodies were used: mouse anti-GFP (G6539, Sigma-Aldrich, 1:000), anti-Flag (F1804, Sigma Aldrich, 1:500), anti-c-Myc (clone 9E10, Roche, 1:400); rabbit anti-GFP (A6455, Invitrogen, 1:500), anti-Flag (F7425, Sigma-Aldrich, 1:300), rat anti-HA (3F10, Roche, 1:300), chicken anti-GAPDH (AB2302, Millipore, 1:2000). The membranes were extensively washed with TBST, followed by incubation with the secondary antibody: DyLight anti-mouse 680 or 800, anti-rabbit 680 or 800, anti-rat 680, and anti-chicken 680 (Thermo Fisher and Rockland, 1:10,000). The membranes were scanned in the Odyssey (Licor) infrared scanner. Images were analyzed using ImageJ. Results are expressed as the proportion of the co- immunoprecipitated proteins related to the total protein.

**Retina epithelial pigmented cells (hTERT RPE-1) cell culture and transfection**. hTERT *RPE-1* (ATCC® CRL-4000™) (RPE1) cells were cultured in DMEM/F12 medium (30-2006, ThermoFisher), complemented with 10% heat-inactivated FBS, 100 U/mL Penicillin and 100 µg/mL Streptomycin (Gibco) at 37 °C and 5% $CO_2$. RPE1 cells were dissociated with 0.25% (w/v) Trypsin, 0.53 mM EDTA, PBS 1× solution. For cell morphology studies, 30.000 cells were plated in previously coated 14 mm glass coverslips (Poly-L-Lysine 1×, Sigma Aldrich, 10 min RT, followed by five washes with sterile water). Cells were transfected 24 h later, with a ratio of 1:4 DNA (0.5 µg) and lipofectamine 2000 (Invitrogen). 24 h after transfection, the cells were fixed with 4% paraformaldehyde (PFA) dissolved in 0.1 M phosphate buffer (PBS), pH 7.4 for 20 min. at RT and immunostainings were performed (see below).

For western blot studies designed to study the mTOR signaling pathway or p38, RPE1 cells were transfected with 3 µg DNA of different constructs (pEGFP-C3-DLGAP4WT, pEGFP-C3-DLGAP4 mutant and pEGFP-C3-empty as control). After 48 h, cells were washed twice with ice-cold PBS and lysed by rotation for 20 min at 4 °C in RIPA buffer. Total proteins were quantified with the Pierce™ BCA Protein Assay Kit (Thermo Scientific) using a microplate reader (Mithras LB940, Berthold Technologies) at 580 nm. Western blots were performed as previously explained. Endogenous protein levels were detected by the following primary antibodies, applied O/N at 4 °C: mouse anti-GFP (G6539, Sigma-Aldrich, 1:1000), anti-DLG1 (clone S64-15, Abnova, 1:1000), rabbit anti-DLGAP4 (Ab73285, Abcam, 1:500), anti-Raptor (ab5454, Abcam, 1:2000), anti-Rictor (A300-459A, Bethyl, 1:2000), anti-FlnA (A301-135A, Bethyl, 1:2000), anti-p38 (SPC-172, StressMarq Bioscience, 1:1000). Immunoreactive bands were detected employing an infrared fluorescence system (Odyssey LiCor), similar as described in the western blot section. Quantitative changes in protein levels were evaluated with ImageJ software (NIH).

**Neuronal progenitor primary cultures**. Neuronal progenitor primary cultures were generated from E12.5 cortices maintained in a B27/N2 medium[86] containing a mixture (1:4) of Neurobasal/B27 medium without vitamin A and DMEM/F12 with Glutamax, supplemented with N2, 0.1 mM nonessential amino acids, 1 mM sodium pyruvate, 500 µg/ml BSA, 0.1 mM 2-mercaptoethanol and l00 µg/ml

Primocin (Invivogen). Cells were placed on 14 mm poly-L-lysine (Sigma Aldrich) and laminin (Gibco) treated coverslips. After 24 h in culture, cells were transfected with Lipofectamine 2000 (Thermo Fisher) and 0.5 µg DNA pBLBP-GFP. Cells were fixed in 4% PFA for 20 min at RT, 24 h after transfections. Immunocytochemistry (ICC) was performed according to standard procedures (see below).

**Immunocytochemistry**. The same protocol was performed either for Neuro2A, RPE1 and neuronal progenitor cell cultures ICC. For endogenous Dlgap4 detection in Neuro2A cells, $3 \times 10^4$ cells were plated in 14 mm coverslips and cultured for 48 h in DMEM containing 10% FBS. Then cells were fixed for 20 min. at RT with 4% w/v PFA in PBS, followed by extensive washing with PBS. Then, 15 min washes in PBST 0.2% (0.2% Triton X-100 in PBS) and incubation with blocking solution (3% BSA, 0.2% Triton X-100 in PBS) was performed for 1 h at RT. Next primary antibodies were applied for 2 h at RT or O/N at 4 °C: mouse anti-GFP (G6539, Sigma-Aldrich, 1:500), rabbit anti-DLGAP4 (Ab73285, Abcam, 1:500). Secondary antibodies were incubated combined with Hoechst (1:1000, Thermo Fisher) for 45 min at RT in the dark: anti-mouse Alexa 488 and anti-rabbit Alexa 568 (ThermoFisher, 1:1000). Finally, the cells were extensively washed in 0.2% PBST and the cell-containing coverslips were mounted with Fluoromount G (Southern Biotechnology). Images were acquired with a TCS Leica SP5-II confocal microscope, and quantifications performed with ImageJ.

***En face* immunohistochemistry and imaging**. *En face* IHC was performed as described in[23]. Briefly, positive electroporated mouse E16.5 brains (IUE at E14.5) from ShmiRNACtl and ShmiRNA*Dlgap4* were fixed in 4% w/v PFA (Sigma-Aldrich, France). Cortical explants were dissected, incubated 15 min at RT in PBST 1% (PBS 1X containing 1% Triton X-100 v/v and 0.02% sodium azide) and then incubated for 2 h at RT in blocking solution (PBS 1X, 0.3% Triton X-100 v/v, 0.02% sodium azide, 3% w/v Bovine Serum Albumin). F-actin immunodetection was performed to delineate cell boundaries[24], combined with the mitosis marker PH3 and GFP to identify the electroporated cells. Primary antibodies rabbit anti-PH3 (Millipore, 06-570, 1:400) and chicken anti-GFP (AB16901, Millipore, 1:500) were applied O/N at RT. Explants were extensive washed in blocking solution and incubated O/N at RT with secondary antibodies anti-mouse Alexa 488 and anti-rabbit Alexa 568 (1:1000, ThermoFisher) and Hoechst (1:1000, ThermoFisher). After extensive washes in blocking solution and PBS, the explants were incubated with Alexa Fluor 633 Phalloidin (1:100, Life Technologies) in PBS for 1 h at RT. After five washes in PBST 1% and PBS 1X, the explants were mounted with Fluoromount G (Invitrogen) with the VS toward the coverslip, in order to obtain the en face view of the ventricular side of the cortex. Fluorescently stained sections were imaged with a confocal TCS Leica SP5-II microscope.

**In situ hybridization**. Mouse digoxigenin-labeled riboprobes were generated by in vitro transcription of a fragment amplified from *Dlgap4* (Probe: NM_146128.3, Allen Brain Atlas). To generate the probe, cDNA was reverse-transcribed from E14.5 mouse brain total RNA, and an 852 nucleotide (nt) fragment amplified (see primers in Supplementary Table 4). Digoxigenin (DIG) probes were synthesized with a labeling kit according to the manufacturer's instructions (Roche Diagnostics). The protocol followed was adapted from[21]. Post fixed in 4% PFA, frozen cryostat sections (E14.5, E17.5) were rinsed in PBS for 35 min, tissue sections were hybridized at 70 °C overnight with the DIG-labeled probes diluted 1/1000 in hybridization buffer (50% deionised formamide, 10% dextran sulfate, 1 mg/ml tRNA, 1× Denhardt's solution). The next day, sections were sequentially washed twice preheated at 65 °C, 1X saline sodium citrate, 50% formamide, 0.1% Tween20, pH 7.5 for 45 min. After cooling down the samples for 1 h, several washes in PBS were performed. For immunological detection of DIG-labeled hybrids, sections were blocked for 30 min (10% fetal bovine serum, 0.1% Triton in PBS) and then incubated overnight at 4 °C in the same solution containing sheep anti-DIG-alkaline phosphatase-conjugated Fab fragments (Roche Diagnostics) diluted 1/2000. The following day, sections were extensively washed in PBS 0.1% triton and then in NTMT buffer (100 mm NaCl, 100 mm Tris-HCl, pH 9.5, 50 mm MgCl₂, 0.1% Tween 20). The alkaline phosphatase chromogen reaction was performed in NTMT buffer containing 100 mg/ml nitro blue tetrazolium (Roche Diagnostics) and 50 mg/ml 5-bromo-4-chloro-3-indolyl phosphate (Roche) at 37 °C for 2–3 h and stopped with PBS. Sections were mounted on glass slides, dried and cover-slipped with Vectamount (Vector Laboratories). Images were obtained using the EVOS XL system (Thermo Fisher).

**Immunohistochemistry**. Mouse embryo brains (E14.5–E18.5) were fixed by immersion overnight at 4 °C in 4% PFA in PBS. Brains were placed in a solution of 75 mg/ml agarose and 150 mg/ml sucrose in phosphate saline buffer (Dulbecco's PBS, Gibco Invitrogen) and coronal sections (70 µm) were prepared using a vibrating blade microtome (Thermo Scientific Microm HM 650 V). Immunodetection was performed using standard protocols. Commercial primary antibodies were from the following sources: mouse anti-GFP (G6539, Sigma-Aldrich, 1:500), anti-Nestin (ab6142, Abcam 1:600), anti-Ki-67 (#556003, BD Biosciences, 1:200), anti-β-catenin (C19220, BD Biosciences, 1:500), anti-N-cadherin (#610920, clone 32, BD Biosciences, 1:500), anti-DLG1 (clone S64-15, Abnova, 1:500), anti-Cux1 (sc13024, Santa Cruz Biotechnology, 1:1000), Class III β3 tubulin, mouse clone

TuJ1 (MMS-435P, Covance,1:10000); rat anti-BrdU (clone BU1/75 (ICR1), BioRad, 1:1000), anti-Ctip2 (clone 25B6, ab18465, Abcam, 1:500); rabbit anti-DLGAP4 (ab73285, Abcam, 1:500), anti-GFP (A6455, Invitrogen, 1:500), anti–caspase-3 (#559565, BD, 1:250), anti-Pax6 (#901301, BioLegend, 1:300), anti-Tbr2 (ab23345, Abcam, 1:300), anti-PH3 (#06-570, Millipore, 1:400); chicken anti-GFP (AB16901, Millipore, 1:500), anti-Tbr2 (AB15894, Millipore, 1:200). Brain slices were extensively washed in blocking solution and incubated for 2 h at RT with secondary antibodies anti-mouse or anti-chicken Alexa 488, anti-rabbit Alexa 568, and anti-rabbit Alexa 633 (1:1000, ThermoFisher) and Hoechst (1:1000, ThermoFisher). Fluorescently labeled sections were imaged with confocal microscopes (Olympus Fluoview FV10i or Leica SP5) equipped with 10×, 20×, and 40× oil Plan-Neofluar and 63× oil Plan-Apochromat objectives. Negative controls were assessed in the absence of primary antibodies. Each individual image was taken every 0.3–0.5 µm, and a stack projection 10 µm thick was use for all the immunohistochemistry analyses. To obtain the whole *z*-stack data set, we used the "Z-projection" mode from ImageJ.

**Peptide competition assay**. Briefly, two consecutive 70 µm brain slices from E14.5 embryos were used for each experiment. Rabbit anti-DLGAP4 antibody (1 µg/ml, Ab73285, Abcam) was diluted at a concentration of 1:500 in blocking solution (PBS, 10% NGS, 0.1% triton X-100). The solution was then divided into two tubes. In the first tube, labeled "blocked", respectively 5× or 100× excess of DLGAP4 blocking peptide (0.2 mg/ml, Ab219578, Abcam) were added. In the second tube, labeled "control", an equivalent amount of buffer was added. Both tubes were then incubated, with agitation, overnight at 4 °C. The immunofluorescence procedure was performed as described in the manuscript. The labeling that disappears when using the "blocked" antibody corresponds to the specific labeling of the DLGAP4 primary antibody. The images were acquired using an Olympus FV10i confocal microscope.

***Dlgap4* RNA interference constructs and validation of shRNAs by RT-qPCR**. Total RNA samples were extracted from Neuro2A cell cultures transfected with 5 µg DNA (pCAGMIR30-ShRNA*Dlgap4*) (Broad Institute, Supplementary Table 4) or a scrambled version (pCAGMIR30-ShRNA *Ctl.*) designed with the siRNA Wizard Software (Invivogen). The ShRNA for *Dlgap4* and the scrambled sequences were cloned in the pCAGMIR30 vector[87]. Cultures were performed in quadruplicate. Real-time qPCRs were performed using the SYBR green method on total, RQ1 DNase-treated RNA samples, following MIQE guidelines[88]. First-strand cDNA was synthesized using 50 ng/µl of total RNA, oligo (dT) and the Superscript III Reverse Transcriptase kit (Invitrogen). Gene-specific primers were designed using Primer Blast (NIH) from NM_146128.6 sequence (see Supplementary Table 4). The cyclophilin B (Ppib, peptidylpropyl isomerase B) gene was used for normalization. Amplicon sizes were between 54 and 79 bp. Standard curves were generated from assays made with serial dilutions of cDNA to calculate PCR efficiencies (90% < efficiency < 105%, with $r^2 \geq 0.998$). Threshold cycles (Ct) were transformed into quantity values using the formula $(1 + efficiency)-Ct$. Differential expression analyses were performed with Student *t*-tests. Only means of quadruplicates with a coefficient of variation of less than 10% were analyzed. Interplate variation was below 8%.

**In utero electroporation (IUE)**. For IUE, timed-pregnant (E14.5) Swiss WT mice were anesthetized with isoflurane (4% during induction and 2–2.5% during surgery) and embryos exposed within the intact uterine wall after sectioning the abdomen. Embryos were placed in Dulbecco's PBS 1× (GIBCO), electrodes (System CUY650P5 NepaGene Co) were placed around the embryo head at a 45° angle and plasmids electroporated by discharging a 4000-µF capacitor charged to 35 V (five electric pulses of 50 ms with 500-ms intervals) with a CUY21 NepaGene electroporator. Embryos were electroporated with different combinations of the expression vectors: pCAGMIR30-ShRNA*Dlgap4* or pCAGMIR30-ShRNAScr*Dlgap4* as a control, jointly with pBlbp-GFP (1 µl, at 1 µg/µl each vector, in sterile endo-free water containing 20% w/v fast green). For rescue experiments, the expression vectors pCAGMIR30-ShRNA*Dlgap4*, p3XFlag-CMV10-*DLGAP4*WT, a Sh-resistant construct, and pBlbp-GFP were injected into the ventricular region of embryonic brains. For overexpression experiments, the expression vectors p3XFlag-CMV10-*DLGAP4*WT or mutant or p3XFlag-CMV10-empty (control) were each co-electroporated with 1:1 pBlbp-GFP, after injection in the lateral ventricles. In Dcx-*DLGAP4* overexpression experiments, pDcx-*DLGAP4*WT-Flag-ires-GFP or pDcx-*DLGAP4*MUT-Flag-ires-GFP were each co-electroporated with 1:1 pDcx-GFP[89]. The embryos were placed back in the abdominal cavity and development was allowed to continue until E15.5, E16.5, E18.5, P0, or P8. Pregnant mice were sacrificed and the embryos were collected in cold dissecting medium (MEM, Gibco, with 15 mM glucose and 10 mM Tris, pH 7.4) and brains were fixed in PFA 4%.

**Cell counting and quantifications**. After immunohistochemistry experiments, z stacks were acquired for each coronal section in a multitrack mode, avoiding crosstalk artifacts for the fluorochromes. All z stacks and image processing were performed with ImageJ 1.51 software (NIH). In general, image stacks contained between 25–40 confocal planes. Counting was performed by assigning cells in 10 bins across the cortical wall (VZ, SVZ, IZ, CP) and relativizing the number of each

bin to the total count. In % analyses, average control (ShmiRNA*Ctl* or in rescue experiments, ShmiRNA*Ctl* + Flag-empty) values were defined as 100% and the percentage of each individual value was established as relative to the average number. Each experimental condition was relativized to the control. In proliferation experiments, raw data expressed as % was established by counting the number of Ki67$^+$BrdU$^+$Blbp-GFP$^+$ or Ki67$^-$BrdU$^+$Blbp-GFP$^+$ cells and dividing them by the total number of BrdU$^+$Blbp-GFP$^+$. Labeled cells were counted in a Region Of Interest (ROI), a 100 or 200-μm-wide and 10-μm-deep stripe across the cortical wall divided by 10 bins. For shRNA and rescue experiments, Blbp-GFP$^+$ cells were counted in an ROI in sections of 70 μm, distinguishing cells restricted to or outside the VZ. In all experiments, the total electroporated cells counted in each ROI ranged between 75 and 250 cells. Only the brains with <75 electroporated cells were excluded from data analysis. To assay spindle orientations, cortical sections were analyzed to determine anaphase-shaped nuclei lining the apical membrane after either N-cadherin immunolabeling or cell labeling with 1:200 Phalloidin in PBS (Molecular Probes, Alexa Fluor™ 633), in the VZ of electroporated ShRNA*Ctl* or ShRNA*Dlgap4* brains at E15.5. Anaphase nucleus angles were measured with the Image J software "angle tool". To assay cell morphology, Tbr2$^+$ Blbp-GFP$^+$ soma surface were analyzed using the Image J software.

**BrdU injections and assessment of proliferation**. Timed-pregnant mice were injected intraperitoneally with BrdU (BrdU, 99% Bioultra, Sigma-Aldrich, 50 μg per gram body weight in 0.15 M phosphate buffer, 0.9% w/v NaCl, pH 7.4) at E15.5, 24 h after in utero electroporation experiments. Then, mice were sacrificed after 30 min or 24 h and their brains processed for immunohistochemistry.

**Patients**. One child from Family P616 (Fig. 1a) and two siblings from Family P477 were initially referred for assessment of developmental delay and/or neurological impairment, and were investigated in a clinical diagnostic setting. All patient information was de-identified. Written informed consent for Next-Generation-Sequencing was obtained from the patients' parents. Genetic research was conducted with approval by the Imagine Institute, Paris. The study design and conduct complied with all relevant regulations regarding the use of human study participants and was conducted in accordance with the criteria set by the Declaration of Helsinki. Both males and females were included in this study for diagnostic purposes, and there was no participant compensation. Children ranged from 0 to 18 years.

**Analysis of human DLGAP4 in EML1-like cortical malformation patients**. Following standard protocols, patient DNA or blood samples were obtained according to the guidelines of the Imagine Institute, Paris. Prior to exome analysis, for each patient (10 atypical heterotopia and 150 other cortical malformation patients), DNA was analyzed by CGH-microarray using the NimbleGen chip 720,000-probes (720 K) array (Roche-NimbleGen, Madison, WI) to exclude potential pathogenic CNVs. Metabolic screening was performed to exclude metabolic disorders such as peroxisomal syndromes.

Whole exome sequencing of peripheral blood DNA from the proband and both parents was performed using the Agilent SureSelect Human All Exon Kits v5, and sequence was generated on a HISeq2500 machine (Illumina). Sequences were aligned to hg19 by using BWA v.0.6.1, and single nucleotide variants (SNVs) and indels were called by using GATK v.1.3. Annotation of variants was performed with GATK Unified Genotyper. All calls with read coverage of ≤2× or a Phred-scaled SNP quality score of ≤20 were removed from consideration. The annotation process was based on the latest release of the Ensembl database. Variants were annotated and analyzed using the Polyweb software interface designed by the Bioinformatics platform of University Paris Descartes and Imagine Institute (see also Supplementary Tables 1–3).

Filters used for variant screening were as following: (i) all variants previously observed (in dbSNP138 and/or in in-house projects) were excluded; (ii) only variants leading to abnormal protein sequence (splicing, non-synonymous, frameshift, stop) were retained; (iii) we considered the PolyPhen-2 and SIFT prediction status as informative but not restrictive. Because all patients were sporadic cases from unrelated parents, the following models of inheritance in the variant screening were considered: autosomal recessive (in particular compound heterozygous but without excluding homozygous variants), X-linked, and de novo SNVs.

Genomic DNA amplifications were performed using standard procedures (primers in Supplementary Table 4), and PCR products were analyzed by direct sequencing using an ABI3700 DNA analyzer (Applied Biosystems, Foster City, CA).

A family with a de novo variant in *DLGAP4* gene was identified (family P616; c.2714_2715insCAGCTGG, N905Qfs, exon 12, see Supplementary information for clinical details). In an additional family (P477), heterozygous amino acid (aa) changes were observed in twins, carrying a missense mutation in *DLGAP4* inherited from the father and a missense mutation in *DLGAP1* inherited from the mother (Family P477; *DLGAP4* c.2893T>G, p.Ser965Ala, exon 13; *DLGAP1* c.1397A>G, p.Asp466Gly, exon 8; see Supplementary information for clinical details and Supplementary Table 1). Genomic DNA amplifications were performed for *DLGAP4* using standard procedures, and PCR products were analyzed by direct sequencing using an ABI3700 DNA analyzer (Applied Biosystems, Foster City, CA). See Supplementary information comparing clinical information of both P616 and P477. Further information are available upon request.

**En face imaging analysis**. Confocal images were acquired with a 0.17 μm z stack depth and the analyzed images had a total depth of 1 μm. At least three randomly-chosen ROIs (246.27 × 246.27 μm) were analyzed per embryo (n = 2–3 embryos per condition). Images were analyzed using ImageJ and the software CellProfiler[24].

**Protein modeling**. Models were built using the I-TASSER (Iterative Threading ASSEmbly Refinement), a hierarchical method for protein structure and function prediction[30]. Two predicted models containing patient variant (P616: c.2714_2715insCAGCTGG, N905Qfs, exon 12) or WT residues were plotted on the solved protein structures for human DLGAP4.

**Statistical analysis**. All data are shown as mean ± SEM. Statistical analysis was performed using Graph Pad Prism 6.0. Comparisons of means in two groups were made using an unpaired Student *t*-test. Normality and homogeneity of variances were tested. Two-tailed analyses were carried out with a significance level established at $p < 0.05$. When data did not comply with the homogeneity of variances, *t*-test with Welch's correction was performed or Kruskal-Wallis, followed by Dunn's multiple comparisons test. For pBlbp-GFP electroporations and cell count quantifications, data were subjected to *t*-test or analysis of variance (ANOVA). Significant main effects were analyzed further by post hoc comparisons of means using Tukey's multiple comparisons test. For cell distribution (or bin) analysis we performed Two-way ANOVA and Tukey's multiple comparisons tests. Pearson's correlation coefficient was calculated from colocalization of pixels images using the JACoP plugin and ImageJ. Pregnant mice/embryos were randomly assigned to the different DNA constructs. When possible, data were collected and analyzed in a blind manner to the experimenter.

**Reporting summary**. Further information on research design is available in the Nature Research Reporting Summary linked to this article.

## Data availability

Whole exome sequencing data are available from the authors upon reasonable request. The patient/parent consent does not cover the deposition of the full genetic data in a public database for privacy reasons; however, data is available for academic researchers upon request for academic research purposes and will be provided likely within one month of the request by email (corresponding author: fiona.francis@inserm.fr). The following databases and in silico softwares were used in the study: Human Gene Mutation Databases (http://www.hgmd.cf.ac.uk/ac/introduction.php?lang=english), NHLBI Exome Sequencing Project (ESP), Exome Variant Server (https://evs.gs.washington.edu/EVS/), the single Nucleotide Polymorphism database (http://ftp.ncbi.nih.gov/snp/), genome aggregation database (gnomAD, https://gnomad.broadinstitute.org), 1000 genomes (https://www.internationalgenome.org/), ExAC Browser (http://exac.broadinstitute.org), PolyQuery (https://polyweb.fr/polyweb/index.html), Polyphen-2 (http://genetics.bwh.harvard.edu/pph2/), Mutation Taster (http://www.mutationtaster.org/), Sorting Intolerant from Tolerant (SIFT, https://sift.bii.a-star.edu.sg/). The genetic variant data generated in this study have been deposited in the dbSNP database under accession codes: rs761228547, rs1371852290, rs752804789, rs138812925, rs1663305011, rs750585274, rs72648943, rs72648278, rs747654057, rs55945684, rs112794616, rs200795874. The original pictures of co-immunoprecipitation and WB membranes and all immunofluorescence quantifications are available in the "Romero_et_al_ncomms-source-data.xlsx" file, within the Source Data File.

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

## Acknowledgements

We thank Dr. X. Morin (LGN and DLG1 constructs), Dr. R. Bayliss (EML1 constructs), Dr. F. Watrin (shRNA empty vector, pCAGMIR30), and Dr. S. Etienne-Manneville (Dlgap1 construct). We thank Dr. A. Baffet (RPE1 cells). We are grateful to S. Haddad, C. Chentouf, members of the Yalcin lab and Dr. M. Penisson for experimental help, and to Dr. A. Uzquiano and Dr. K. Chinnappa for feedback on the manuscript. We thank members of the lab for comments and discussions. We thank the IFM animal experimentation facility and cellular and tissue imaging platforms at the Institut du Fer à Moulin, supported also by the Région Ile de France and the FRC Rotary.

D.M.R. and F.F. are associated with the BioPsy Labex project and the Ecole des Neurosciences de Paris Ile-de-France (ENP) network. Our salaries and lab were supported by Inserm, Centre national de la recherche scientifique (CNRS), Sorbonne University. F.F.'s group obtained the following funding French Agence National de la Recherche (ANR-13-BSV4-0008-01; ANR-16-CE16-0011-03, to F.F. and N.B.B.), Fondation Bettencourt Schueller (F.F.), the European Union (EU- HEALTH-2013, DESIRE, No 60253, to F.F. and N.B.B.), the JTC 2015 Neurodevelopmental Disorders affiliated with the ANR (for NEURON8-Full- 815-006 STEM-MCD, to F.F., N.B.), the Fondation Maladies Rares/Phenomin (project IR4995, F.F.), the European Cooperation on Science and Technology (COST Action CA16118, including NBB and F.F.). This project was also further supported by the French ANR under the frame of E-Rare-3, the ERA-Net for Research on Rare Diseases (ERARE18-049) and the Fondation pour la recherche médicale (FRM, Equipe FRM 2020 awarded to F.F.).

## Author contributions

D.M.R. conceived, designed, and performed experiments, analyzed data and wrote the manuscript. K.P. contributed to the identification of the mutated gene P616 and performed experiments. R.B. helped in I.U.E. experiments and performed some of the IHC and en face confocal studies. I.M. contributed with reagents and molecular cloning of constructs. A.H performed some of the IP and WB experiments. A-G.L.M. and F.P. are the patient clinicians and provided clinical information. A.B. and J-F.D. performed exome-sequencing experiments. M.S-R. helped in I.U.E. experiments and performed some of the IHC confocal studies. B.Y and S.C.C. obtained and characterized the *Dlgap4* KO mouse model and performed experiments. J.C. contributed to discussion. N.B.-B. contributed to discussion related to patient clinical information and MRI interpretations. F.F. conceived and designed experiments, helped with data analyses and interpretation, and wrote the manuscript.

## Competing interests

The authors declare no competing interests.
