## [Peer Review File · Nature Communications]

Novel role of the synaptic scaffold protein Dlgap4 in ventricular surface integrity and neuronal migration during cortical developmentReviewers' Comments:

Reviewer #1 (Remarks to the Author):

In this manuscript, the authors identified the DLGAP4 mutations associated with cortical malformation including heterotopia and showed that Dlgap4 interacts with EML1-- the microtubule-binding protein which has been shown related to human heterotopia-- and other VZ proteins. The authors further performed knock-down (KD) and overexpression studies in mouse brain to show DLGAP4 function in neural progenitor cells and their daughter cells.

The reviewer thinks that this manuscript includes the potentially interesting points, including involvement in the human heterotopia, but also feels that Dlgap4 function on the neural progenitor cells is not clarified enough to understand the mechanism of heterotopia generation because of their experimental design and interpretation of the results.

Major points:

(1) The in utero electroporation induces the transgenes into the cells near the apical surface at the electroporation, and thus both progenitor cells and postmitotic (young) neurons are transfected. Then, the introduced plasmid DNAs are diluted in the neural progenitor cells that undergo cell division. Therefore, to examine the function of Dlgap4 on the neural progenitors and their possible involvement of the heterotopia formation, another loss-of-function strategy is needed. CRISPR/Cas9-mediated KO by in utero electroporation or transposase-mediated genome integration of miR-Dlgap4 or mutated-Dlgap4 will be suitable for this kind of experiments. The authors can try them to examine whether continuous loss-of-dlgap4 or expression of mutated-Dlgap4 induce heterotopias at the postnatal stages.

(2) The reviewer thinks the Dlgap4-KD phenotype is not clarified enough. Regarding the Dlgap4 function in neural progenitor cells, the authors showed that KD induces the differentiation of the neural progenitors, impairs the polarity of progenitors and integrity of the ventricular surface, and induces the oblique division. However, the relationships of these phenotypes are not clearly shown by their experimental results.

Considering the time-lag from the electroporation of ShmiR vectors to the effective decrease of Dlgap4 protein, E15 phenotype (one day after IUE) --the 'fate change of the neural progenitors'-- is unlikely derived from the alternation of the spindle orientation of the mitotic cells at the apical surface. Furthermore, it has been shown that the alternation of the spindle orientation itself does not primarily change the fate choice (the decision to either proliferate or differentiate) of the daughter cells in LGN KO.

Instead, it seems that the primary effect of Dlgap4 KD is on the cell polarity and AJs. The KD-neural progenitor cells may tend to retract their apical processes and re-positioned to the SVZ without undergoing mitosis at the apical surface. Therefore the authors should examine the distribution of the neural progenitor cells and the basal progenitor cells in the cerebral walls. Along with this, visualization of the morphology of the KD cells by the sporadic fluorescent labeling is needed. It is also important to examine whether INM of the neural progenitors to the apical surface in G2 phase normally occurs or not.

Other points:

Fig2j: At least peptide competition assay is necessary to confirm the conclusion.

Fig3a and S-Fig2g; The authors only showed the ShmiR-KD efficiency by the transcriptome level by qPCR. To confirm that their IUE procedure reduces the Dlgap4 expression in vivo (at one day after electroporation) and also to verify the reliability of IHC assay with anti-Dlgap4 antibody, anti-Dlgap4 IHC of KD and control brains should be presented.

Fig3b, The image of anti-Nestin IHC shows its VZ immunoreactivity is weaker than that of SVZ/IZ;

this is a different signaling pattern from the previous reports. Other antibodies can be used to visualize the RG fibers (RC2, Vimentin, etc.).

Fig3c-e: although the differences are described as significant by the statistical tests, the effects look small, which may be easily altered by the judgment of VZ vs. SVZ region or dorso-ventral difference of the electroporated area. Therefore, to describe the distribution of the electroporated cells, it is better to use the bin data, and show the distribution by the histograms (for example, separate cerebral walls to 10 bins from the apical and basal side). This is also the case of the rescue experiment (Fig6 c.d).

Fig3 f-g: To examine the apical surface and the AJs, the authors should perform the en-face analysis of the apical surface (immunostaining of the tissues and examination from the ventricular surface). Such study of the additional molecules related to the AJs and Par complex (ZO1, Par3, aPKC) will provide further information. Then, the en-face view of the double-staining with GFP and the above markers will further provide the information about whether the KD-phenotype is cell-autonomous or including non-cell autonomous effect.

Fig4a: The anti-Pax6 immunostaining pattern looks different between control and KD samples. It seems curious that only an apical half of VZ is Pax6+ region in KD case, and the reviewer wonders whether these samples are compared under the same conditions.

Because it is very important to evaluate the differentiation status of the electroporated cells, the additional data/examinations are needed:

- (1) The IHC images should be accompanied by the low magnified view of the cerebral walls to indicate the electroporated region and staining pattern of the marker genes (to show the GFP+ and GFP- area in the same cerebral wall)
- (2) In addition to Pax6, other markers (Sox2, Ki67, PH3) should be examined in the electroporated cells. BrdU-pulse labeling (Fig5a) data may be presented here.
- (3) to determine whether the KD increases the basal progenitor cells, examine the frequency of %SVZ Tbr2+ Ki67+ GFP+ cells/ total GFP+ and %SVZ Tbr2+PH3+GFP+ cells/total GFP+.
(Tbr2 is a marker for neuronally differentiating cells, and expressed both in the young postmitotic neurons and the basal progenitors.)

Results from the rescue experiments are also needed.

In Fig4f and d, and measurement of spindle orientation of the EGFP+ cells: the authors should label the centrosomes (for example, by gamma-tubulin antibody staining) with the apical surface (anti-N-cadherin or ZO1, etc.) to measure the spindle orientation precisely.

In Fig 4g, for comparison of spindle orientation, presentation of the mean value with t-test is not suitable. Use the cumulative plot (or display the raw data as dot plot) and Kolmogorov-Smirnov test (or another statistical test, such as comparison of the median value).

Fig5b, d, e: Because %control is less informative, present the raw% data, which will support the validity of the experiment procedure.

Fig5: Through the Fig5 data, the authors argue the KD-cells tend to be less proliferative, and by the KD, more cells exit cell cycle to become the postmitotic daughter cells. However, the number of electroporated neurons at P0 is reduced by the KD. The reviewer is somehow confusing. Also, it is difficult to understand the meaning of the graph of Fig5e.

Fig5c: Is the ventricular-surface disorganization by KD weakened at E16 (2 days after IUE)?

Fig5d: Did the authors examine %Ki67+/BrdU in the EGFP+ cells?

Otherwise, they should examine %Ki67+/ BrdU+EGFP+ cells as the mitotic index. This value is important to understand the phenotype.

Fig5e-g: The authors labeled 'GFP' in the figures, but did they use Blbp-GFP to label the electroporated cells?

Since Blbp-promoter basically works in the progenitors, it is not suitable for the long-term labeling

of the postmitotic neurons. Other ubiquitous promoter-driven GFP (such as CAG- or CMV-EGFP) should be used to evaluate the position and the number of the neurons. As heterotopia is characterized by the ectopic neurons, the authors should examine the *Dlgap4* function on neuronal migration, accompanied by the rescue experiments.

Fig5e, f, g: Low magnified view of the cerebral walls are also needed to show the overall positioning of the electroporated cells. This is also the case of Fig5g: because the UL depth differs at the anatomically different region. The low-magnified image showing only the *ShmiRNADlgap4* electroporated area showing thin-UL (but the next, neighboring non-electroporated CP are not thin) is also needed.

Fig6e, Because the IUE method cannot precisely control the number of the electroporated cells, comparison of total GFP+ is not so informative. Additional examinations related to cell proliferation such as %BrdU-labeling and %PH3 in total GFP+ cells are needed. The authors also should examine whether KD or OE (or OE of *Dlgap4*-MUT) effect on cell death, for example, by the IHC of anti-cleaved-Caspase 3 antibody.

Fig.7 a,b: How do the authors explain the molecular mechanisms that *DLGAP4* increases or decreases these proteins' expression levels? (*FLNA*, *p38*, *Raptor*, and *Rictor*).

Reviewer #2 (Remarks to the Author):

In this manuscript, Romero et al., describe the identification of novel variations in *DLGAP4* in patients with subcortical heterotopia. *DLGAP4* is a membrane-associated guanylate kinase family with known function in synapses. *DLGAP4* interacts with another subcortical heterotopia associated, microtubule-binding protein *EML1*. Elegant gain and loss of function analysis of *DLGAP4* indicates a role in cell fate, proliferation and migration. *DLGAP4* function involves actin cytoskeleton and mTOR signaling pathways. These studies not only identify a new heterotopia linked genetic mechanism, but also illustrates its importance in progenitor development in the embryonic cortex. Addressing the following issues will help provide a clear link between *DLGAP4* and progenitor development and how disruptions in these functions can lead to cortical heterotopia.

- 1) What is the effect of increased *Tbr2*⁺ cells observed with *Dlgap4* knockdown? Does it lead to increased generation of cortical neurons or ectopic collection of them away from the cortical plate?
- 2) *DLGAP4* is widely expressed in embryonic cortex, including in migrating neurons? Disrupted migration after *DLGAP4* knockdown an indirect effect resulting from apical/basal progenitor malfunction in VZ? Or a direct effect on migrating neurons.
- 3) One general missing link in this paper is a clear outlining of how *DLGAP4* malfunction in the VZ can lead to heterotopia. Additional mapping of the nature of changes in progenitor differentiation or neuronal migration will help in this regard. In addition, addressing why in spite of almost ubiquitous expression of *DLGAP4* in the embryonic cortex, heterotopia are noticeable only in some areas of cortex, will be helpful.

Reviewer #3 (Remarks to the Author):

Summary and major concerns

This manuscript examines the role of *Dlgap4* in the pathogenesis of cortical malformations seen in human patients. The authors describe a range of interesting disruptions to cortical development following knockdown of protein expression in the mouse, at the levels of both progenitor and postmitotic cells, and further describe a range of DLGAP4 interactors using biochemical approaches.

Below I describe my main criticisms of the paper in general terms, followed by more specific comments on the text.

1) Meandering experiments

The manuscript describes effects of *Dlgap4* manipulation on cell proliferation, cell cycle exit, apical surface integrity, the ratios of apical versus basal progenitor cell types, the localization or abundance of apical markers, lamellipodia morphology, cell localization, and cortical upper layer thickness. Several distinct interacting partners for DLGAP4 are described in cell culture and immunoprecipitation experiments.

However, these many disparate ideas do not form a sufficiently satisfying narrative. In particular, in no case has any particular function of DLGAP4, or one of its interaction partners, been convincingly linked to any specific phenotypic alteration to cortical development. Many potential leads are identified, but then not explored further and validated in vivo during mouse development. Several observations, such as the very interesting disruptions to the apical surface, could serve as launch points for more focused lines of experiments, while omitting other things.

2) Lack of distinction between effects in progenitors and effects in neurons

I suggest that, from the beginning, the manuscript makes systematically a very clear distinction between which functions of DLGAP4 might be relevant in which cell types and for which phenotypic aberrations. The protein is expressed widely throughout the cortical plate and neuroepithelium. The cortical phenotype is thus likely a mosaic of problems due to loss of function in both progenitors and neurons, but the manuscript fails to achieve many mechanistic insights because this idea is largely ignored. This is especially clear at the end when the model is presented: the cell type is entirely omitted.

The meandering nature of the experiments described above, as well as the discussion, would be greatly improved with a renewed focus on cell-type-specific effects. For example, by knocking down expression in the mouse neocortex using a neuron-specific promoter, you could begin to tease apart different aspects of phenotype. Are the radial fibers still disrupted? Are neurons still mislocalized along the neuroepithelium, or the apical surface intact? It would be especially interesting to ask whether neurons are found ectopically below the ependyma due to loss of apical adhesion, in progenitors, or rather mismigration due to disruptions in postmitotic cells is instead sufficient. You could further ask which interaction partners are expressed in progenitors vs neurons, and you could search for readouts of related signaling pathways in vivo as well.

I am afraid that the suggested changes—to describe all effects in the language of cell autonomous effects—would require substantial additional experiments and re-writing, but should result in a greatly improved manuscript and help to organize the many ideas presented.

Other comments

Abstract

“Mouse models of this disorder are rare...”

You should clarify here whether you mean 1) researchers rarely use mice to model subcortical heterotopia, or 2) disease-causing mutations known from humans rarely produce comparable effects in mice.

“we identified a key patient with similar phenotype...”

The wording here is confusing- I think you mean a “patient with an EML1-like phenotype, but with

a novel variation in DLGAP4.”

Introduction

You can remove the parenthetical [in gyrencephalic brains], as bRGs can be found in lissencephalic mammals including mice, just in more modest numbers. See for example Vaid et al 2018 doi: 10.1242/dev.169276

Results

“the abnormal C-terminus was also predicted to affect the overall structure of the protein.”

A little bit more description here would be helpful- affect it how? Which domains are lost or preserved in your model?

“In parallel, trio-based exome sequencing revealed a second family...”

You describe the first patient P616-3 as having subependymal heterotopia and occipital lobe polymicrogyria. However, you describe the new patients from family P477 as having instead “pachygyria.” You could clarify here how the cortical phenotype of this patient is similar or different to the first patient. You might also point out where the DLGAP4 mutation in family P477 lies with respect to the protein domains.

Regarding Figures 2 and 3, the expression data for Dlgap4 in the neocortex seems a more natural starting point. I would move the expression data to before the interaction and immunoprecipitation data.

You might consider moving a piece of the Dlgap4 ISH data from the supplement to the main text- it would be nice to see the ISH and IHC data adjacent to one another.

“Dlgap4 labeling appeared present in the somal regions of neuronal progenitors and neurons in the mouse brain (Fig. 3a)”

You could be more specific about the locations of these cell types- you are inferring the identity of the cell types based on their location, but you are not presenting costaining with cell-type-specific markers. E.g., the cells in the ventricular zone are probably apical radial glia, whereas staining in the cortical plate is probably neuronal.

“A punctate perinuclear pattern of endogenous Dlgap4 was observed in dividing and interphase Neuro2A cells, and in primary cultures of Pax6+ Blbp+ cells”

Since you do not show that the cells in culture express Pax6, you should remove that detail from the text.

“As Dlgap4 was highly expressed at the ventricular surface, we compared its expression pattern with polarity and ventricular surface proteins”

I did not find anything here particularly informative, especially in the relatively low power images presented. You could probably remove all of Supp Fig 2a-f, as well as this paragraph, without detracting at all from the manuscript.

Figure 3B: Blbp-GFP should be a reporter for RG cells, correct? Are the GFP+ cells outside of the VZ massively displaced RG-like cells, or rather basal progenitors/neurons that retain the cellular GFP of their progenitors?

Figure 3C: How do you have percentages over 100%? I.e. how can more than 100% of total cells in the ROI be GFP+?

“Nestin+ RG fibers were also reduced and disorganized”

The different fiber morphologies are clear here, but how do you judge that there are fewer fibers in the experimental condition? You should quantify this or remove “reduced.”

“This may indicate ventricular surface weakening, contributing to a periventricular phenotype”

What do you mean by “weakening?” In particular, you haven’t mentioned that the apical markers you employ participate in adherens junctions between apical progenitors. You could tie these ideas together a little more.

"To further explore these results, coimmunoprecipitation experiments were also performed for the spindle orientation protein LGN, an interactor of DLG1 36, and for which the gene (GSPM2) shows mutations in patients with Chudley-McCullough syndrome, featuring heterotopia and polymicrogyria."

This sentence is quite a slog, I suggest breaking it up. Also, the gene name is *GSPM2*.

"Thus, Dlgap4 knockdown leads to altered spindle orientation and cell fate in the VZ. DLGAP4 is found in the same complex with both DLG1 and LGN proteins"

You have shown pairwise interactions using co-IP, but is there evidence that all three are ever found in *the same complex*?

Fig 5d: For these quantifications, did you discriminate between GFP+ and GFP- cells? This could tell you something about possible cell autonomous or non-autonomous effects.

Fig 5e: You should walk through the logic of this experiment more explicitly. When was BrdU administered with respect to the IUE? Are you testing the laminar position of neurons born at E15.5? If so, you need to compare the distributions of BrdU+ neurons between knockdown and control, i.e. the percentages of BrdU+ neurons in CP/IZ. Instead, you show a quantification for the percentage of GFP+ cells which are BrdU+, and its not clear what this demonstrates.

"Fewer cells in the knockdown condition were labeled for Cux1"

If you are going to make this claim, you must quantify the total numbers of Cux1+ nuclei in a column. If true, it raises some questions which you should address: are Cux1+ neurons mislocalized, or stuck in the intermediate zone? Are there elevated levels of apoptosis that removed neurons that would have gone to the upper layers? Did the neurons born after E14.5 adopt a deep layer (e.g. Ctip2+) fate instead?

Supp Fig 3a/b: Where are the empty vector controls for these experiments demonstrating ectopic cells and rosettes? Can electroporation cause these defects?

"Perhaps related to this, increased numbers of GFP+ cells were also observed in the VZ with decreased numbers in the more superficial zones (Supp. Fig. 3c, d)."

Quantification for this? Also, I don't think this is supposed to refer to SFig 3d.

"...was able to rescue the integrity of the F-actin labelled ventricular surface (Fig. 6a)."

Please clarify how exactly you judged that the integrity of the surface was rescued.

"Thus, the mutant protein incorporating an altered C-terminus shows differences from WT DLGAP4, and is notably unable to correctly rescue certain cortical phenotypes."

It would be helpful for the reader here if you briefly summarized which things were rescued by the mutant version and which things were not.

"In the Genepaint database, an expression of Shank2 was observed in the VZ as well as the CP at E14.5 by ISH (Supp. Fig. 3f)."

This ISH image is of uninterpretable background staining. You should definitely remove this, and consider redoing the ISH.

"These results, together with the diminished expression of F-actin in Dlgap4 downregulated mouse brains (Fig. 3f-h), suggest a role of Dlgap4-Shank2 in actin cytoskeleton organization and / or dynamics in progenitor cells."

This bit about interaction between Dlgap4 and Shank2 doesn't seem to lead anywhere. In particular, you don't provide any evidence through experiments or through the literature that Shank2 is at all related to the actin cytoskeleton in apical radial glia. You should follow up on this speculation with manipulation of Shank2, or consider dropping this section from the manuscript.

"...showed a decreased proportion of cells presenting lamellipodia morphology compared to control (Fig 6i,j)."

This should just refer to Fig 6j. Also, how did you define lamellipodia? The ideas in this paragraph have not quite congealed- you are presumably looking at lamellipodia because of some

relationship with actin dynamics, but you should spell this out for the reader.

"There was also a tendency for decreased F-actin stress fiber organization with OE of the WT protein (Fig. 6l)."

You should show us images of stress fiber organization, and explain what this is, why you are looking at it, and how you detect it.

"Altogether, these results further suggest that DLGAP4 influences the actin cytoskeleton."

This is too vague- can you relate your cell culture data to what you see in the neocortex following DLGAP4 knockdown? Is the regulation in some sense positive or negative? What kinds of actin structures are influenced, and which are not, by DLGAP4?

"The DLGAP4 partner, DLG1, has other functions as a second messenger..."

I don't think you can use the term second messenger like this- it refers to small molecules.

"We hence further analyzed these proteins in RPE1 cells overexpressing WT and mutant DLGAP4 by performing Western blot analyses."

In this section, you need to provide more context about the potential molecular targets you are looking at. How exactly are they related to the signaling pathways you describe? For instance, you drop in Raptor and Rictor without telling the reader anything about them.

"De-regulation of this pathway, predicted via Rictor to impact the actin cytoskeleton, is likely to contribute to the DLGAP4 phenotypes presented here."

It is too bold here to use the word "likely." There is no evidence for which mechanisms downstream of DLGAP4 manipulation cause which of the many described phenotypes- you should follow up on these mTOR data by asking how this signaling pathway is affected in vivo in the neocortex.

Discussion

"Thus DLGAP4, known for its role in synapses, appears when associated with particular gene variations, to be a cortical malformation gene."

Even given the qualification, I suggest avoiding the term "cortical malformation gene." Really, it's a cortical formation gene. Maybe just specify that DLGAP4 is now shown to be required for x, y, and z aspects of cerebral cortex development.

"convergent with our Dlgap4 results"

I would reserve "convergent" as a term for convergent evolution. Try "similar" or something of that nature.

"Furthermore, changes in cell fate may be brought about by changed spindle orientations"

This could probably be moved to a new paragraph where you describe more specifically the known mechanisms relating spindle orientation to cell fate, e.g. concepts like unequal inheritance of apical determinants, delamination and differentiation, etc.

"Clearly there is overlapping molecular machinery between synapse formation and maintenance, and ventricular surface homeostasis."

This is not clear to me. If you are going to say this, you need to spell out exactly which molecular machineries and mechanisms you are talking about and which processes are shared between synapse formation and apical surface regulation.

Reviewer #4 (Remarks to the Author):

Romero and colleagues report a new gene, DLGAP4, associated with cortical malformations, suggesting a role for DLGAP4 during cortical development. They identified a de novo frameshift DLGAP4 mutation in a single proband with subcortical heterotopia and a missense trans-compound heterozygous mutation in DLGAP4 and DLGAP1 in a patient with a distinct cortical malformation characterized by parieto-occipital pachygyria. They report that Dlgap4 is expressed in the developing cortex in mice, interacts with EML1, a known subcortical heterotopia gene, and both knockdown and overexpression of Dlgap4 lead to ventricular surface phenotypes in mice. Additionally, they showed the identified frameshift DLGAP4 mutation affects the interaction of Dlgap4 with Shank2 which impacts the actin cytoskeleton. While there are interesting genetic findings in this work, the manuscript lacks evidence to support some of the major hypotheses proposed, specifically the gain of function/gene-gene interaction model in family P477. Statistical analysis or functional data elucidating the functional impact of the missense DLGAP4 mutation and the potential interaction between DLGAP4 and DLGAP1 is necessary to support this hypothesis.

Major comments

1. Can the authors comment on the expression level of DLGAP4 and DLGAP1 in the human brain during development?
2. The authors should document their analysis of the exome sequencing data in Materials and Methods, including data processing and downstream filtering criteria used.
3. Can the authors explain why other predicted damaging variants identified in the two families were not considered as pathogenic? There is no comment on MYO1C and ZFX identified in P477 and ABCG10 identified in P616.
4. It would be helpful if the authors can add the pLI score, missense Z score, and minor allele frequency in public databases for each identified variant in the variant table.
5. Please specify the transcript ID for isoform 1 in Figure 1d as well as in the main text.
6. DLGAP4 has multiple transcripts. Despite its high overall expression level in adult human brain, the canonical transcript (ENST00000373913.3, 989aa) which is the specific one described in the manuscript, has low or no expression in the adult brain depending on the region (based on the GTEx database). Do the two identified mutations (chr20:35155357:T:G and chr20:35154373:-:CAGCTGG) have similar functional impact in all transcripts or at least those highly expressed in the brain?
7. The authors suggested in Discussion that the disease-causing mechanism of the missense DLGAP4 mutation is gain of function (over expression) and might be mediated through interaction with DLGAP1. Please provide statistical and/or functional data to support this hypothesis.
8. Continuing on the point discussed above, the authors should comment on the functional impact of the missense DLGAP1 mutation. Does it lead to decreased expression level, protein conformational change, or others?

Minor comments

1. Supplementary Table 2 should be Supplementary Table 3 and Supplementary Table 3 should be Supplementary Table 1. All main and supplementary tables and figures should be labeled and put in the proper order.
2. For all acronyms used in the manuscript, please spell out the full name when they are used for the first time (e.g. 'IUE' for 'in utero electroporation' in the main text).
3. It is worth mentioning that DLGAP4 has a pLI score of 0.99 in gnomAD, suggesting this gene is highly intolerant to loss of function mutations.
4. Throughout the manuscript the clarity of the writing could be improved; for example, the header of Supplementary Table 3 (the variant table) and penultimate sentence on page 14 are both unclear.

AUTHOR'S RESPONSE TO REVIEWERS

Reviewers' comments:

Reviewer #1

In this manuscript, the authors identified the DLGAP4 mutations associated with cortical malformation including heterotopia and showed that Dlgap4 interacts with EML1-- the microtubule-binding protein which has been shown related to human heterotopia-- and other VZ proteins. The authors further performed knock-down (KD) and overexpression studies in mouse brain to show DLGAP4 function in neural progenitor cells and their daughter cells. The reviewer thinks that this manuscript includes the potentially interesting points, including involvement in the human heterotopia, but also feels that Dlgap4 function on the neural progenitor cells is not clarified enough to understand the mechanism of heterotopia generation because of their experimental design and interpretation of the results.

Major points:

(1) The in utero electroporation induces the transgenes into the cells near the apical surface at the electroporation, and thus both progenitor cells and postmitotic (young) neurons are transfected. Then, the introduced plasmid DNAs are diluted in the neural progenitor cells that undergo cell division. Therefore, to examine the function of Dlgap4 on the neural progenitors and their possible involvement of the heterotopia formation, another loss-of-function strategy is needed. CRISPR/Cas9-mediated KO by in utero electroporation or transposase-mediated genome integration of miR-Dlgap4 or mutated-Dlgap4 will be suitable for this kind of experiments. The authors can try them to examine whether continuous loss-of-dlgap4 or expression of mutated-Dlgap4 induce heterotopias at the postnatal stages.

We acknowledge the reviewer's comments and regarding this point we collaborated with Dr Binnaz Yalcin's lab (University of Burgundy, Dijon, France, INSERM U1231) who characterized a knock-out (KO) mouse model of *Dlgap4*, now included as Supp. Fig. 10 and in Suppl. information. The new findings demonstrate a clear role of *Dlgap4* during cortical development. Heterozygous KO of *Dlgap4* mice show clear neuroanatomical defects including microcephaly, with a reduced cortex and hippocampus. Homozygous KOs are not compatible with life. Despite the absence of heterotopia-like phenotypes in this constitutive knockout model, these results show a contribution of Dlgap4 in the control of the size of several telencephalic regions, further pointing to a role of Dlgap4 in neural progenitors.

Page 18: "Dlgap4 heterozygous knockout (KO) mice show neuroanatomical defects in the dorsal telencephalon

*To further study the role of Dlgap4 during brain development, we obtained mouse mutants from the International Mouse Phenotyping Consortium produced using the knockout-first allele method⁵³ (see Supplementary information, **Supplementary Fig. 10a**). The expected number of WT and heterozygous mice were observed, but no homozygous animals, suggesting that the double dosage of the mutant allele of Dlgap4 is not viable. Using a*

recently developed robust approach for the assessment of 40 brain parameters across 22 distinct brain regions⁵⁴, we analyzed neuroanatomical defects in adult *Dlgap4* heterozygous mice (**Supplementary Fig. 10b-c**, Supplementary information). A number of brain structures showed significantly decreased size in heterozygotes when compared to WTs (**Supplementary Fig. 10b**). The total brain area was reduced by 22% ($p = 0.0004$), with especially decreased size of the hippocampus (29%, $p = 0.00005$) (**Supplementary Fig. 10d**), the corpus callosum (24%, $p = 0.0009$), the anterior commissure (29%, $p = 0.00006$), the thalamus (15%, $p = 0.018$) and the neocortex (10%, $p = 0.0004$) compared to WTs. In order to determine which cortical layers were most affected, the cortex was divided into four bins corresponding to cortical layers I, II-IV, V and VI (**Supplementary Fig. 10e**). We found that Bins 2-4 were reduced in size when compared to WT, including layers II-IV (Bin 2) 28%, $p = 0.03$ and layer VI (Bin 4), 16%, $p = 0.05$ (**Supplementary Fig. 10f**). Finally, cell counts across the entire brain were decreased by 20% ($p = 0.03$) (**Supplementary Fig. 10g**). Despite the absence of heterotopia-like phenotypes in this chronic heterozygous KO model, at least at the Lateral +0.60 mm level studied in unilateral sagittal brain sections, these results strongly suggest that *Dlgap4* plays a role in determining the size of several telencephalic structures. "...

(2) The reviewer thinks the *Dlgap4*-KD phenotype is not clarified enough. Regarding the *Dlgap1* function in neural progenitor cells, the authors showed that KD induces the differentiation of the neural progenitors, impairs the polarity of progenitors and integrity of the ventricular surface, and induces the oblique division. However, the relationships of these phenotypes are not clearly shown by their experimental results.

Considering the time-lag from the electroporation of ShmiR vectors to the effective decrease of *Dlgap4* protein, E15 phenotype (one day after IUE) --the 'fate change of the neural progenitors'-- is unlikely derived from the alternation of the spindle orientation of the mitotic cells at the apical surface.

Furthermore, it has been shown that the alternation of the spindle orientation itself does not primarily change the fate choice (the decision to either proliferate or differentiate) of the daughter cells in LGN KO.

Instead, it seems that the primary effect of *Dlgap4* KD is on the cell polarity and AJs.

We do agree with the reviewer's comments and to improve the manuscript we have now transferred the spindle orientation results to supplementary data (new Suppl. Fig.6) together with the LGN co-IP results. We agree that these results could be the consequence of the ventricular surface weakening observed by *Dlgap4* knockdown (see below). We have included complementary new experiments that support this idea. We also discuss this topic (page 20-22).

Please see the section:

Page 10-11: ..."*Dlgap4* KD modifies the ratios, position and morphology of cortical progenitors", previously entitled "A change in cell fate and RG spindle orientations is observed upon *Dlgap4* knockdown"...

Our new experiments are consistent with a scaffold role of Dlgap4 at the ventricular surface. We demonstrated that DLGAP4 is able to interact by Co-IP with the adhesion molecule β -catenin. Furthermore, we have reinforced the potential connection with actin cytoskeleton by new results showing that DLGAP4 is able to co-IP with the actin-nucleation-promoting factor cortactin (Fig.7e-g). Cortactin is important in promoting among other functions, lamellipodia persistence, actin polymerization and cytoskeletal remodeling during the epithelial–mesenchymal transition (EMT) (Schnoor et al. Trends Cell Biol. 2018; Bryce et al. Curr Biol. 2005; Uruno et al. Nat Cell Biol. 2001). With these results, combined with the results of *Dlgap4* KD at the ventricular surface (shown in Fig. 3; Fig.6a-b; Suppl. Fig. 5a-c), we demonstrate that the disruption of the ventricular surface could be due to loss of key protein-protein interactions, suggesting a potential role of the Dlgap4 scaffold protein in the organization of radial glial cell (RG) AJ components at the apical domain and actin cytoskeleton dynamics. It is worth mentioning that only the OE of the DLGAP4 WT construct was able to rescue the ventricular surface phenotype (Fig.6a, also observed in Fig.6e-g and Suppl. Fig.8c), suggesting that proper WT protein levels during development are essential to avoid structural brain anomalies, such as subependymal heterotopia.

From page 10: ...“Apical RGs are integrated into the AJ belt mediating cell–cell adhesion, in which cadherin, cytoplasmic catenins and F-actin are critical components²⁰. Even in the absence of ventricular extruding ectopic GFP⁺ cells, a decrease in ventricular surface expression of F-actin was observed in the electroporated regions ($p < 0.01$) (**Fig. 3e-f**). Aberrant β -catenin and N-cadherin expression and localization were also evident (**Fig. 3g**, white arrows). This may indicate a ventricular surface weakening, potentially due to cell-to-cell adhesion defects contributing to a ventricular ectopic cell phenotype. We also found that Flag-DLGAP4 co-immunoprecipitated with GFP- β -catenin (**Fig 3h-j**), and when DLGAP4 is mutated with the P616-3 variant, co-IP of β -catenin is reduced by 47% ($p = 0.045$) (**Fig 3h-j**). These experiments further suggest a role for Dlgap4 at the ventricular surface. Hence, Dlgap4 KD results in disruption of cortical development, impacting ventricular surface integrity and cell positioning.”...

From page 15: ...“**DLGAP4 impacts actin cytoskeleton dynamics in vitro...**

... The changed lamellipodia and filopodia however further suggest that DLGAP4 may influence actin cytoskeleton dynamics, with potentially subtle differences between WT and mutant proteins. To further explore these findings, we performed co-IP experiments with the actin-nucleation-promoting factor cortactin, known to interact with the Dlgap4 interactor, Shank2⁴⁵. Cortactin is an actin binding protein promoting among other functions, lamellipodia persistence, actin polymerization and cytoskeletal remodeling during the epithelial–mesenchymal transition (EMT). It interacts with AJ components allowing F-actin accumulation, and is an important factor during the breakdown and formation of AJs⁴⁶⁻⁴⁷. Cortactin is expressed in progenitors and neurons in the developing neocortex and hippocampus^{32,33,48}. As expected, GFP-DLGAP4 was detected in anti-Flag cortactin IPs and vice versa (**Fig. 7e-g**). We found that the co-IP of Flag-cortactin with the GFP-DLGAP4 mutant was reduced by 54% ($p = 0.0018$) (**Fig. 7f**). These results together with the abnormalities of F-actin expression and AJ integrity at the ventricular surface in the Dlgap4 KD condition (**Fig. 3e-g, Fig. 6a**), suggest a role of Dlgap4 in actin cytoskeleton organization and/or dynamics, most probably involving cortactin.”...

We also now focus our discussion on pages 21-22 on adhesion and the actin cytoskeleton.

The KD-neural progenitor cells may tend to retract their apical processes and re-positioned to the SVZ without undergoing mitosis at the apical surface. Therefore the authors should examine the distribution of the neural progenitor cells and the basal progenitor cells in the cerebral walls.

We agree with the reviewer. We show that overall numbers of Pax6⁺Blbp-GFP⁺ cells are decreased and Tbr2⁺Blbp-GFP⁺ cells are increased (Fig 4). As suggested by the reviewer, and in all the pertinent results presented in this manuscript, we have performed a binning analysis and the new graphs are shown now in Fig.3b, Fig.4c, Fig.6d, Fig.8f,j, Suppl. Fig. 5c-f, i-l, Suppl. Fig.7e and Suppl. Fig. 8a-b,e, Suppl. Fig. 10f. Regarding progenitor cells, this analysis revealed an altered distribution along the cortical wall. Interestingly, the distribution of Pax6⁺Blbp-GFP⁺ cells in the most apical bins is altered with more cells in bin 1 and less in bin 2 (upper VZ, Fig 4c) in fitting with the images presented in Fig 4a. On the other hand, Tbr2⁺Blbp-GFP⁺ cells showed a significantly increased proportion in Bin3-4 and a decrease of these cells in Bin1-2 in the mutant condition compared to control (Supplementary Fig. 5d). In addition, we found significant differences for total Tbr2⁺ cells (includes GFP⁻) with a decrease of Tbr2⁺ cells in bin 2 and an increase in bin 3 in the *ShmiDlgap4* compared to *Ctl* brains (Supplementary Fig. 5j). We hence observe a higher proportion of Pax6⁺ cells from KD brains in the most apical bin (interestingly, including increased PH3⁺ cells, see below) and a shift of Tbr2⁺ cells to a more basal bin. We have also included complementary experiments for Tbr2 cells that are discussed later.

Page 10: ...“ *In addition, 48 h after IUE, Dlgap4 KD Blbp-GFP⁺ cells still showed an abnormal distribution compared to control (see arrows, **Supplementary Fig. 5b-c**).*”...

Page 11: ...“ *Additionally, we observed an altered distribution of Pax6⁺ Blbp-GFP⁺ cells (in bins 1-2, **Fig. 4c**), Tbr2⁺ Blbp-GFP⁺ cells (in bins 1-4) and total Tbr2⁺ cells (in bins 2-3) within the cortical wall, with proportionally more Tbr2⁺ cells present in superficial bins (**Supplementary Fig. 5d,j**).*”...

Along with this, visualization of the morphology of the KD cells by the sporadic fluorescent labeling is needed.

According to the reviewer’s comment, we have now included complementary new images and analysis of Pax6⁺Blbp-GFP⁺ and Tbr2⁺Blbp-GFP⁺ cells morphology (See Fig. 3a, Fig. 4g-j, Supplementary Fig. 4). We have included new information related to the abnormal soma surface and cell processes detected for Tbr2⁺ Blbp-GFP⁺ cells and radial morphology in the VZ for RG Blbp-GFP⁺/ Blbp-GFP⁻ cells) comparing *ShmiRNActl* and *Dlgap4* KD experiments.

Page 9: ...“*Nestin⁺ RG fibers appeared occasionally disorganized in basal regions after Dlgap4 KD (yellow arrows, **Fig. 3a**), and also close to the ventricular surface, with some horizontal and rearranged fibers (**Supplementary Fig. 4e**).*”...

Page 11: ...“ Furthermore, the morphological analysis of progenitor cells in electroporated brains showed that $Tbr2^+$ $Blbp-GFP^+$ cells from the *Dlgap4* KD condition present a decreased number of cell processes (25%, $p = 0.034$), with a longer extension (41%, $p = 0.041$), accompanied by an enlarged somatic surface (27%, $p = 0.02$) (Fig. 4g-j)... ”

It is also important to examine whether INM of the neural progenitors to the apical surface in G2 phase normally occurs or not.

The analysis of 30 min $BrdU^+$ $Blbp-GFP^+$ cell distribution between Bin 1 and 2 (which comprises ~80 μm from the ventricular surface), indicates that the distribution of these cells in the *Dlgap4* KD condition is altered. The 30 min $BrdU$ pulse labels cells in S phase, and the results show that there is a decrease in the proportion of $BrdU^+$ cells in bin 1 and no changes in upper bins compared to *Ctl.*, except for an increased trend from bin3 (Suppl. Fig. 5e,k). Bin 2 contains $Pax6^+$ cells and a proportion of $Tbr2^+$ cells (see Fig.4c, Suppl. Fig. 5d,j and Suppl. Fig.9b). As $Pax6^+$ $Blbp-GFP^+$ cells are fewer in bin 2 for *Dlgap4* KD condition, these will contribute less to the $BrdU^+$ labeling, and it is thus likely that $Pax6^+$, $Tbr2^+$ cells in this region (GFP^+ and GFP^-) largely account for the S-phase labeling. We think it is therefore unlikely that there is a problem during G2, and indeed $Pax6^+$ cells accumulating at the ventricular surface may present lengthened mitosis (see below ‘Other points’ for new complementary experiments showing increased counts of ventricular surface $PH3^+$ nuclei at E16.5 in Fig.5f-i). We cannot exclude as well a problem of RGs reaching basal positions of the VZ during G1. Indeed, G1 lengthening has been associated with the transition from stem cell-like apical progenitors to fate-restricted intermediate progenitors (Arai et al. Nat Commun 2011). In addition, an increased proportion of $Tuj1^+$ cells were found in bin 1 in *Dlgap4* KD brains (Suppl. Fig. 5a). Because of the actual international situation, we could not proceed with more specific INM studies for this point, although the combined data sheds light on the reviewer’s question:

Page 12: ...“ *Dlgap4* KD brains showed a decreased proportion of proliferating $BrdU^+$ $Blbp-GFP^+$ cells after a 30 min period (35%, $p = 0.0084$), accompanied by abnormal $BrdU^+$ $Blbp-GFP^+$ and total $BrdU^+$ cell distribution, less in bin 1 and tendencies for more in basal bins (Fig. 5a-b; Supplementary Fig. 5e,k). ”...

Other points:

Fig2j: At least peptide competition assay is necessary to confirm the conclusion.

We decided to delete the panel corresponding to Fig. 2j from the revised version of the manuscript, as it did not seem to be essential data, and was pinpointed by the reviewer as preliminary. Peptide competition results, reducing the anti-*Dlgap4* signal, remain in the manuscript in Supplementary Fig. 2c.

Fig3a and S-Fig2g; The authors only showed the ShmiR-KD efficiency by the transcriptome level by qPCR. To confirm that their IUE procedure reduces the *Dlgap4* expression in vivo (at one day after electroporation) and also to verify the reliability of IHC assay with anti-*Dlgap4* antibody, anti-*Dlgap4* IHC of KD and control brains should be presented.

As suggested by the reviewer, we have now included the new complementary experiments and quantifications to show the reduction in Dlgap4 protein expression by immunohistochemistry. These experiments were performed in E16.5 embryonic brain slices. Quantifications show a clear reduction in Dlgap4 protein after KD, also supporting the specificity of the antibody, coherent with the RT-qPCR results of ShmiRNADlgap4 KD (now presented altogether in Supplementary Fig.4).

Page 9: ... “ *In addition, DLGAP4 protein immunoreactivity was reduced by 62.6% in Blbp-GFP⁺ cells in the VZ, 48h after IUE ($p < 0.0001$) (Supplementary Fig. 4b-d).*”...

Fig3b, The image of anti-Nestin IHC shows its VZ immunoreactivity is weaker than that of SVZ/IZ; this is a different signaling pattern from the previous reports. Other antibodies can be used to visualize the RG fibers (RC2, Vimentin, etc.).

According to the reviewer’s comment, we have re-checked all our Nestin images for both conditions. In order to show both the migration and the ventricular surface defects in one picture, the images we had selected for Nestin staining were not necessarily the most representative (and corresponding to ‘standard’ Nestin labelings). We have now changed the panels in Fig 3. Disorganization of RG fibers is still observed in the ShmiRNADlgap4 condition as pointed out above, including in apical regions of the developing neocortex (also see Suppl. Fig. 4e).

Fig3c-e: although the differences are described as significant by the statistical tests, the effects look small, which may be easily altered by the judgment of VZ vs. SVZ region or dorso-ventral difference of the electroporated area. Therefore, to describe the distribution of the electroporated cells, it is better to use the bin data, and show the distribution by the histograms (for example, separate cerebral walls to 10 bins from the apical and basal side). This is also the case of the rescue experiment (Fig6 c.d).

As the reviewer suggests, we have changed the analysis and graphs in order to observe better the distribution of electroporated cells along 10 bins throughout the cortical wall. Significant reductions/ increases of the proportion of cells are observed along the bins. We have added the binning graphs in all pertinent figures (e.g. Fig.3b, Fig.4c, Fig.6d, Fig.8f,j, Suppl. Fig. 5c-f, i-l, Suppl. Fig.7e and Suppl. Fig. 8a-b,e, Suppl. Fig. 10f).

As well as the examples already described above:

Page 14: ... “ *Assessing Blbp-GFP⁺ cell distribution, we found that only WT DLGAP4 appeared to normalize this aspect of the phenotype, largely restoring Blbp-GFP⁺ cells to control levels (Fig. 6b,d). Thus, in bins 1 and 2 the decrease in Blbp-GFP⁺ cells in the KD condition was restored to control levels.*”...

Page 14: ... “ *Concerning distribution, Pax6⁺ Blbp-GFP⁺ and Tbr2⁺ Blbp-GFP⁺ cells appeared restored in bin 1, partially restored in bin 2, although the mutant lead to a noticeable trend for increased cells in basal regions (e.g. from bin 4, Supplementary Fig. 8a-b).*”...

Fig3 f-g: To examine the apical surface and the AJs, the authors should perform the en-face analysis of the apical surface (immunostaining of the tissues and examination from the ventricular surface). Such study of the additional molecules related to the AJs and Par complex (ZO1, Par3, aPKC) will provide further information. Then, the en-face view of the double-staining with GFP and the above markers will further provide the information about whether the KD-phenotype is cell-autonomous or including non-cell autonomous effect.

As the reviewer suggested, we performed new experiments showing the *en face* view of E16.5 brains, 48 h after IUE (E14.5) of control and KD *Dlgap4* constructs (results are shown in Fig. 5f-i). The experiments suggested by the reviewer, including additional markers such as AJs and Par complex, would be very interesting and informative, but as a matter of time and current international conditions, they could not be performed with *en face* imaging (but see below for AJ marker experiments). However, we provide further information about the KD phenotype in terms of cell-autonomous versus non-cell autonomous effects, associated with cell cycle/proliferation. We analyzed the mitotic marker PH3 co-labeled or not with Blbp-GFP by *en face* confocal imaging. The analysis of PH3⁺Blbp-GFP⁺/Blbp-GFP⁺ cells aimed to evaluate the cell autonomous effects. In addition, the analysis of PH3⁺Blbp-GFP⁻/Blbp-GFP⁺ cells located at a distance <10 μm from the Blbp-GFP⁺ cells reveals non-cell autonomous effects. Finally, total PH3⁺Blbp-GFP⁺ cells were counted per ROI. These new data showed an increased amount of mitotic cells (PH3⁺) in ShmiRNADlgap4 brains, consistent with the increase of Ki67⁺Blbp-GFP⁺ cell proportions in Bin 1 (Supplementary Fig. 5f-h). In addition, overall Ki67⁺Blbp-GFP⁺ cells are reduced in ShmiRNADlgap4 brains. The *en face* results might therefore suggest defects in mitotic progression elicited by *Dlgap4* KD. These results may be in agreement with a higher cell differentiation (Kalebic et al. Cell Stem Cell 2019, Kalebic and Huttner, TINS 2020).

Page 12-13: ... “ Since *Dlgap4* KD brains showed increased Ki67⁺ Blbp-GFP⁺ and Pax6⁺ Blbp-GFP⁺ cells in bin 1 (Fig. 4a,c, Supplementary Fig. 5f,g), we performed enface confocal imaging and PH3 labeling to characterize mitotic cells in this region. An increase in PH3⁺ Blbp-GFP⁺ mitotic cells by 76.5% ($p = 0.046$) was observed in ShmiRNADlgap4 compared to control brains (Fig. 5f-g). In addition, non-cell-autonomous effects were suspected since many PH3⁺ cells appeared GFP⁻. Counting PH3⁺ Blbp-GFP⁻ cells (located within less than 10 μm from PH3⁺ Blbp-GFP⁺ cells) showed an increased proportion in the ShmiRNADlgap4 condition compared to the control (86.3%, $p = 0.02$) (Fig. 5f,h). Finally, *Dlgap4* KD brains showed an increased trend for total PH3⁺ cells (GFP⁺ or GFP⁻, increased by 55.3% compared to control) (Fig. 5f,i). Since cell cycle exit is increased, these results may suggest defects in mitotic progression elicited by *Dlgap4* KD, or alternative perturbations of the VZ, altering progenitor behavior.”...

In addition, we have also identified non-cell autonomous effects linked to the *Dlgap4* KD phenotype(s) with other markers (e.g. Tbr2, see Fig. 6e,g and data not shown in the manuscript regarding for GFP⁻ cells, but included here below, Tuj1, Nestin), as discussed at relevant sections of the manuscript.

Unpublished Tbr2 data:

Unpaired t test	
P value	0,0245
P value summary	*
Significantly different? (P < 0.05)	Yes
One- or two-tailed P value?	Two-tailed
t, df	t=2,984 df=6
How big is the difference?	
Mean \pm SEM of column A	184,6 \pm 14,26 N=4
Mean \pm SEM of column B	290,4 \pm 32,44 N=4
Difference between means	105,8 \pm 35,44

As mentioned above, we also provide further evidence pointing to DLGAP4 function in AJ organization. Our results are consistent with its scaffold role at AJs. We demonstrated that DLGAP4 is found in protein complexes with the adhesion molecule β -catenin. Furthermore, DLGAP4 is able to co-IP with the actin-nucleation-promoting factor cortactin, important in promoting among other functions, lamellipodia persistence, actin polymerization and cytoskeletal remodeling during the epithelial–mesenchymal transition (EMT) (Schnoor et al. 2018; Bryce et al. 2005; Uruno et al. 2001) (Fig.3g-j and Fig.7e-g). *Dlgap4* KD or OE induces disruption of the ventricular surface (Fig. 3a,d,f,g), rosette formation (Suppl. Fig.7b) and disorganization of AJ proteins N-cadherin and β -catenin, as well as reduced F-actin at the ventricular surface (Fig. 3a,d,f,g, Fig. 6a). Importantly, by evaluating whether the *Dlgap4* KD phenotype could be mitigated by concomitant DLGAP4 OE with WT or mutant protein, we showed that only WT DLGAP4 rescues the ventricular surface phenotype, shown by F-actin labeling, and not the mutant protein (Fig.6a). Altogether, and as mentioned above, the abnormalities of F-actin expression (see also Fig.7a-c) and AJ integrity at the ventricular surface in *Dlgap4* KD experiments, suggest a role of *Dlgap4* in AJ scaffolds and actin cytoskeleton organization and/or dynamics.

Fig4a: The anti-Pax6 immunostaining pattern looks different between control and KD samples. It seems curious that only an apical half of VZ is Pax6+ region in KD case, and the reviewer wonders whether these samples are compared under the same conditions. Because it is very important to evaluate the differentiation status of the electroporated cells, the additional data/examinations are needed: (1) The IHC images should be accompanied by the low magnified view of the cerebral walls

to indicate the electroporated region and staining pattern of the marker genes (to show the GFP+ and GFP- area in the same cerebral wall)

We carefully analyzed ShmiR*Ctl* and ShmiR*Dlgap4* brains, always in electroporated areas corresponding to the lateral region of the somatosensory cortex (example now shown in Supp. Fig 4b). We have improved the brightness levels of the image to illustrate better the Pax6 patterns (Fig 4a). However, as mentioned above and shown in the quantifications, there are more Pax6⁺ cells in bin 1 and less in bin 2 in the *Dlgap4* KD condition, which is representative in the image selected for publication. Thus the anti-Pax6 immunostaining pattern indeed appears different in the KD brains. This information is hence now more convincing in the new version of the manuscript.

Page 11: ... “ *Additionally, we observed an altered distribution of Pax6⁺ Blbp-GFP⁺ cells (in bins 1-2, Fig. 4c), Tbr2⁺ Blbp-GFP⁺ cells (in bins 1-4) and total Tbr2⁺ cells (in bins 2-3) within the cortical wall, with proportionally more Tbr2⁺ cells present in superficial bins (Supplementary Fig. 5d,j).*” ...

In agreement with the reviewer comments, we have included complementary low magnification images related to the data presented in Fig. 4 (Pax6) in an extra supplementary file with some other examples for the reviewer. After having confirmed the similarity of the *Ctl* and ShmiRN*Dlgap4* regions, we show, along with the manuscript images, the low magnification images we have which were not too bleached by the previous photography (Fig.2b; Supp. Fig.4b or supplementary file included for the reviewers), as well as schemas of the cortical area electroporated (Fig. 6a; Fig. 8a,e).

(2) In addition to Pax6, other markers (Sox2, Ki67, PH3) should be examined in the electroporated cells. BrdU-pulse labeling (Fig5a) data may be presented here.

As mentioned before, as well as data presented in Fig 5, we have now included complementary analyses of BrdU⁺Blbp-GFP⁺ cell distribution along the cortical wall (see Supplementary Fig. 5e,k). Furthermore, we have performed new analyses and included new experiments that were also suggested such as Ki67⁺Blbp-GFP⁺, the analysis of cell cycle exit and proliferation index, and PH3⁺ Blbp-GFP⁺ from an apical *enface* view, and quantifications are shown in new Fig.5, Suppl. Fig. 5 and Suppl. Fig. 8. Ki67 and PH3 results consistently show increases of cells in bin 1. We have kept the BrdU-pulse data in Fig. 5 to facilitate the flow to readers.

Page 12: ... “ *We further analyzed proliferating cells after Dlgap4 KD, by performing injections of BrdU (incorporated in dividing cells) at E15.5, 24 h after IUE, with mice sacrificed 30 min and 24 h later. Dlgap4 KD brains showed a decreased proportion of proliferating BrdU⁺ Blbp-GFP⁺ cells after a 30 min period (35%, p = 0.0084), accompanied by abnormal BrdU⁺ Blbp-GFP⁺ and total BrdU⁺ cell distribution, less in bin 1 and tendencies for more in basal bins (Fig. 5a-b; Supplementary Fig. 5e,k).*” ...

Page 12: ... “ *Since Dlgap4 KD brains showed increased Ki67⁺ Blbp-GFP⁺ and Pax6⁺ Blbp-GFP⁺ cells in bin 1 (Fig. 4a,c, Supplementary Fig. 5f,g), we performed en face confocal imaging and PH3 labeling to characterize mitotic cells in this region. An increase in PH3⁺*

Blbp-GFP⁺ mitotic cells by 76.5% (p = 0.046) was observed in ShmiRNADlgap4 compared to control brains (Fig. 5f-g).”...

(3) to determine whether the KD increases the basal progenitor cells, examine the frequency of %SVZ Tbr2⁺ Ki67⁺ GFP⁺ cells/ total GFP⁺ and %SVZ Tbr2⁺PH3⁺GFP⁺ cells/total GFP⁺.

(Tbr2 is a marker for neuronally differentiating cells, and expressed both in the young postmitotic neurons and the basal progenitors.)

Results from the rescue experiments are also needed.

We have included some complementary experiments and analyses (including binning) in order to answer the reviewer’s concerns about basal progenitors and the differentiation status of electroporated cells. This did not include the specific double labelings mentioned (due to the current pandemic difficulties), however, we did perform other pertinent analyses.

As mentioned previously, in Bin 2/3 are located most of Tbr2⁺ cells in control brains (≈70% of total at E15.5). Overall, we found consistently increased counts of Tbr2⁺Blbp-GFP cells and reduced counts for Pax6⁺ Blbp-GFP cells (Fig. 4). Also, during S-phase of the cell cycle, RG cells are expected to be located in Bin 2, the most basal region of the VZ. By performing the binning analysis, in *Dlgap4* KD brains we found reduced Tbr2⁺Blbp-GFP cells in Bin1-2 and increased proportions from Bin 3-4 towards more basal bins (see Fig. 4d and Supplementary Fig. 5d,j).

In order to determine whether increased Tbr2⁺ cells are consistent with more basal progenitor cells, we further analyzed BrdU-pulse results along the cortical wall, 24 h after IUE (E15.5). Results showed no significant differences in Bin2 in KD brains vs Ctl, however an increased trend was observed in this bin and importantly towards more basal layers of the developing cortex (Suppl. Fig.5e). Thus, since we found a reduction of Pax6⁺ in Bin2 (and overall counts, Figs. 4 and Supp. Fig.5) with *Dlgap4* KD, therefore the BrdU label in this region is likely to be explained by the Tbr2⁺ cells (overall increased in *Dlgap4* KD at E15.5).

In addition, we analyzed proliferation and cell cycle exit 48 h after IUE at E16.5 along the cortical wall. We did not observe a higher proportion of Ki67⁺ cells in the *ShRNADlgap4* brains in Bin2/3, and BrdU overall is less (Suppl.Fig.5f-h). Indeed, we observed more cells that exit the cell cycle (Supplementary Fig. 5i). With all these data combined, we believe that perhaps more Tbr2⁺ cells are produced from Pax6⁺ RG cells (at the expense of self-renewal) as shown at 24 h after IUE (Matzusaki and Shitamukai, Cold Spring Harb Perspect Biol 2015), and 48 h after IUE they exit the cell cycle to produce post-mitotic neurons. This would fit with the findings showing that there are fewer Ki67⁺ cells in bin 2 (E16.5). In saying this, we are aware that only ≈10% of multipolar basal progenitors in the mouse are proliferative during mid corticogenesis (Kalebic et al. 2019).

Moreover, we have shown that at E16.5, Tbr2⁺Blbp-GFP⁺ cells present an abnormal morphology, showing for example an increased soma size and a decreased number of primary processes, which has been associated with a decreased proliferative capacity (Kalebic and Huttner, 2020).

Finally, as mentioned above, from *en face* experiments, a higher proportion of mitotic cells (PH3⁺Blbp-GFP⁺ and PH3⁺Blbp-GFP⁻) were found in ShmiDlgap4 KD brains at E16.5. Since cell cycle exit is increased, these results may suggest defects in mitotic progression elicited by Dlgap4 KD, or alternative perturbations of the VZ, altering progenitor behavior.

“Rescue” experiments, evaluating whether the Dlgap4 KD phenotype could be mitigated by concomitant DLGAP4 OE with WT or C-terminal mutant protein (resistant to the ShmiRNA), were performed and included in Fig. 6 and Suppl. Fig. 8 (results page 13-14). ShmiDlgap4 induced an increase in Tbr2⁺Blbp-GFP⁺ cells, and the OE of either the WT or mutant constructs was able to restore these to control values. These results may suggest that the N-terminal domain of the protein (WT and mutant) is involved in the molecular mechanisms leading to cell differentiation. Complementary to these results, 30 min BrdU-pulse showed a decrease in proliferating cells in the ShmiDlgap4 condition. In this case, concomitant DLGAP4 OE experiments only with the WT construct were able to restore control levels in BrdU uptake. From these experiments we conclude that proper Dlgap4 levels and specific protein-protein interactions with different DLGAP4 domains are necessary to allow normal cortical development. These points were included in the new discussion section (pages 23-24). Thus our results are consistent with changes in the basal progenitor population.

In Fig4f and d, and measurement of spindle orientation of the EGFP⁺ cells: the authors should label the centrosomes (for example, by gamma-tubulin antibody staining) with the apical surface (anti-N-cadherin or ZO1, etc.) to measure the spindle orientation precisely.

We acknowledge the reviewer’s suggestion regarding the use of a centrosome marker in combination with an apical surface marker to improve spindle orientation measurements. However, as mentioned above, we have now moved these results to supplementary data (new Suppl. Fig.6), hence lowering their emphasis, we were also faced with the impossibility of conducting further experiments at this stage. In addition, we have discussed the change in progenitor dynamics more deeply and pinpoint this as more probably a consequence of the weakening of the ventricular surface related to other complementary experiments performed. We have nevertheless improved the section regarding the measurement of spindle orientation, referring to preliminary results.

Page 11: ... “ *The change in the ratio of progenitor cells prompted us to assess RG spindle orientations in the VZ. Preliminary quantitative analyses of Blbp-GFP⁺ cell anaphase angle at the ventricular surface showed a decrease of the median value (73.09° Ctl, 48.03° ShmiDlgap4, p = 0.012), consistent with an increased proportion of horizontal and oblique divisions at the expense of vertical ones (Supplementary Fig. 6a-c). To further explore these findings, we studied the possible interaction of DLGAP4 with a protein involved in spindle orientation, LGN (coded by the GPSM2 gene), which has been implicated in Chudley-McCullough syndrome, involving heterotopia and polymicrogyria³⁷. We show that GFP-LGN was present in Flag-DLGAP4 IPs and vice versa (Supplementary Fig. 6d-f). Mutant DLGAP4 showed non significant reduction of immunoprecipitated GFP-LGN (36%, p = 0.21) (Supplementary Fig. 6d,e,g).*” ...

In Fig 4g, for comparison of spindle orientation, presentation of the mean value with t-test is

not suitable. Use the cumulative plot (or display the raw data as dot plot) and Kolmogorov-Smirnov test (or another statistical test, such as comparison of the median value).

We thank the reviewer for mentioning this important point. We have changed the graph for a dot plot showing the median values and performed a Mann Whitney test (still significant at $*p=0.011747$). This data is included in the new Supplementary Fig. 6.

Fig5b, d, e: Because % control is less informative, present the raw % data, which will support the validity of the experiment procedure.

As this reviewer and two other reviewers pointed to the concern about analyzing cell autonomous effects on proliferation, we have performed complementary new experiments and included them in the main figures, replacing the previous ones. Please see below our explanation and how we have addressed and discussed this point (Fig 5c-e and in the manuscript text). We have expressed the results as suggested by the reviewer (raw %). In addition, we have included in the methods section “Cell counting and quantifications” the total range of Blbp-GFP⁺ electroporated cells counted in all the experiments. The embryos with low numbers of electroporated cells were excluded from the analysis.

Page 12: ... “ Moreover, with BrdU injection at E15.5, *Dlgap4* KD brains showed a decrease in the proliferation index (21.5%, $Ki67^+ BrdU^+ Blbp-GFP^+ / BrdU^+ Blbp-GFP^+$, $p < 0.0005$) and a corresponding increase in the proportion of cells exiting the cell cycle (23.3%, $Ki67^- BrdU^+ Blbp-GFP^+ / BrdU^+ Blbp-GFP^+$, $p < 0.0001$) (Fig. 5c-e). Exiting cells are increased in proliferative bins (significant in bin 2, **Supplementary Fig. 5I**). These results may suggest a premature differentiation of cortical progenitors into neurons.”...

Fig5: Through the Fig5 data, the authors argue the KD-cells tend to be less proliferative, and by the KD, more cells exit cell cycle to become the postmitotic daughter cells. However, the number of electroporated neurons at P0 is reduced by the KD. The reviewer is somehow confusing.

Also, it is difficult to understand the meaning of the graph of Fig5e.

We apologize for the confusion. We have improved experimental data presentation in Fig.5, taking into account this reviewer’s and other reviewers’ comments. New experiments and analyses were performed and results are now included in Fig. 8 and Supplementary Fig. 9.

As it was pointed out, E15.5 BrdU injections (24 h after IUE) showed an increased cell cycle exit 24 h later (E16.5) and reduced proliferation index (Fig 5c-e), as mentioned above (and see below). We also found a reduced number of BrdU⁺ Blbp-GFP⁺/ Blbp-GFP cells in the ShmiRNADlgap4 condition at P0 (previous Fig. 5e). This is consistent with the reduced number of proliferating cells already observed at E15.5 (BrdU 30 min, previous Fig.5a-b) and confirmed by the analysis of the proliferation marker Ki67 at E16.5, by analyzing Ki67⁺ Blbp-GFP⁺/ Blbp-GFP cells (now included in Supplementary Fig. 5f-h). Although P0 images also showed a potential migration phenotype by BrdU⁺ labeling, and due to the fact that Fig.5e may be redundant and lead to confusions, we have eliminated this panel from the manuscript. We have in addition now focused our data on characterizing perturbed migration, also likely to contribute to the reduced thickness of Cux1⁺ cells at P0 (see new improved P0 data analyses in Fig.8a-d).

Fig5c: Is the ventricular-surface disorganization by KD weakened at E16 (2 days after IUE)?

Yes it is. We have now added new complementary figures that show this (e.g: Fig.3g; Fig. 5c; Suppl. Fig. 5b). We observed this phenotype even at P8 (see Suppl. Fig. 9).

Fig5d: Did the authors examine %Ki67+/BrdU in the EGFP+ cells?

Otherwise, they should examine %Ki67+/ BrdU+EGFP+ cells as the mitotic index. This value is important to understand the phenotype.

As mentioned before, we have improved this part of the manuscript by re-analyzing and including new complementary experiments to have a better understanding of proliferation and differentiation with *Dlgap4* KD. We have hence improved the representative images of the IHCs and the quantification graphs showing cell autonomous effects on proliferation index and cell cycle exit (%Ki67+/BrdU+GFP+ cells and %Ki67-/BrdU+GFP+, respectively). Results are now shown in Fig. 5c-e and expressed as raw %. These results are in agreement with further quantifications shown in Supplementary Fig. 5f-h,l.

Page 12: ... “ *Moreover, with BrdU injection at E15.5, Dlgap4 KD brains showed a decrease in the proliferation index (21.5%, Ki67+ BrdU+ Blbp-GFP+ / BrdU+ Blbp-GFP+, p < 0.0005) and a corresponding increase in the proportion of cells exiting the cell cycle (23.3%, Ki67- BrdU+ Blbp-GFP+ / BrdU+ Blbp-GFP+, p < 0.0001) (Fig. 5c-e). Exiting cells are increased in proliferative bins (significant in bin 2, Supplementary Fig. 5I). These results may suggest a premature differentiation of cortical progenitors into neurons.* ”...

Fig5e-g: The authors labeled ‘GFP’ in the figures, but did they use Blbp-GFP to label the electroporated cells?

Since Blbp-promoter basically works in the progenitors, it is not suitable for the long-term labeling of the postmitotic neurons. Other ubiquitous promoter-driven GFP (such as CAG- or CMV-EGFP) should be used to evaluate the position and the number of the neurons. As heterotopia is characterized by the ectopic neurons, the authors should examine the *Dlgap4* function on neuronal migration, accompanied by the rescue experiments.

The reviewer is correct to mention this and we have changed the nomenclature in the manuscript to Blbp-GFP. The intention of our work was to evaluate the effects of *Dlgap4*, a synaptic protein with a well-known scaffold function in mature neurons, in progenitor cells. Blbp-GFP is useful for this and the GFP is then inherited in neurons. In our revised neuronal migration experiments, we included three experimental groups, control, *Dlgap4* KD and OE of mutant DLGAP4. These newly added data showed that GFP under the Blbp promoter is still expressed in the cortical plate at P8, indicating that GFP is preserved in the RG-derived lineage. We actually used a long version of the Blbp promoter (1.6 Kb) obtained from the N. Heintz laboratory (Rockefeller University, New York). Schmid et al (Glia. 2006) suggested that length of the Blbp-promoter by containing critical regulatory CIS-regions could be associated with the level of GFP expression, and very likely its stability. In their work, fluorescence intensity dwindles in embryonic brains potentially due to a short 250 bp promoter (see also Feng and Heintz, Development. 1995). In addition, an original paper from

Heintz lab (Feng et al. Neuron 1994) demonstrated that Blbp expression in the mouse cerebral cortex VZ is present from mid-corticogenesis until the second postnatal week.

Neurons born after E14.5 are destined for the superficial layers of the mouse cortex, which express Cux1. In our improved experiments presented here, better evaluating effects on neurons, we identified the neuronal populations at P8 derived from IUE in RG cells at E14.5, by co-labeling GFP with Cux1 and Ctip2 markers in three experimental conditions (control, *Dlgap4* KD and OE of mutant *Dlgap4*). Control brains showed GFP⁺ co-labeled with Cux1 in superficial layers of the cortex as expected. This was also observed in *Dlgap4* KD and OE mutant *Dlgap4* conditions, although a decreased Cux1 labeling was observed in *Dlgap4* KD brains in the CP. These experiments also show that the delayed migrating/arrested neurons (see also *Dcx* promoter experiments described below) already observed at P0 are still present at P8. Moreover, in the *Dlgap4* KD, we show that some of these ectopic cells can have a late born neuronal identity (Cux1⁺GFP⁺), however most of them present an early born neuronal identity (Ctip2⁺GFP⁺) at this age. The disruption in neuronal migration observed later during development (from E16.5 Blbp-GFP cells by binning analysis) could derive from an altered RG morphology (c.f. Nestin stainings, Fig 3), failing as substrate for radial neuronal migration, as well as autonomous neuronal effects.

Page 17: ... “ *Neurons born after E14.5 are destined for the superficial layers of the cortex, we hence stained for Cux1, a marker of cortical superficial layers II/III. Fewer cells in the Dlgap4 KD condition were labeled for Cux1, and the thickness of the Cux1 staining in the CP was also significantly reduced (20%, p = 0.044) as well the density (32%, p= 0.0066) (Fig. 8a-c). Blbp-GFP⁺ labeling also indicated a possible migration defect since many mutant cells were still observed distributed along the cortical wall at P0 (Fig. 8d). In addition, proportions of GFP⁺ cells were also ectopically located at P8 forming abnormal clusters of cells in both Dlgap4 KD and DLGAP4 mutant OE (Supplementary Fig. 9). The identity of these cells was confirmed by co-immunolabelings either with Cux1 or Ctip2 (marker for cortical deep layers), showing the presence of many Ctip2⁺ GFP⁺ cells (arrowheads) in the heterotopia-like region (see Supplementary Fig. 9). The Cux1⁺ cell layer also appeared to be reduced in thickness at P8 in mutant conditions, likely due to a combination of several factors such as proliferation, apoptosis and migration defects (Supplementary Fig. 9).” ...*

In addition, to confirm specific effects in neurons (responding to this and other reviewers comments) we also used *Dcx* promoter-specific constructs to overexpress DLGAP4 (Fig 8, E14.5 – E18.5).

Page 18: ... “ *To study more specifically neuronal defects, we used a Dcx promoter-GFP reporter plasmid co-electroporated with a similarly neuron-specific overexpressed WT or mutant DLGAP4 (pDcx-DLGAP4-WT or -MUT -Flag-ires-GFP) constructs (Fig. 8e). Four days after electroporation, a delay in neuronal migration was found in both DLGAP4 WT and MUT OE conditions along the cortical wall (p < 0.0001) (Fig. 8f-g). An increased proportion of Dcx-GFP⁺ cells were found in bins 1-2 and less cells reached superficial bins 8-9 in both conditions compared to control (Fig. 8f-g). Cux1⁺-GFP⁺ cells were identified showing that these neurons could adopt an appropriate identity. Total Dcx-GFP⁺ Cux1⁺/GFP⁺ cells did not change among conditions (Fig. 8h, p > 0.05). However, similar results of*

retarded migration were found by analyzing *Dcx-GFP⁺/Cux1⁺* cells along the cortical wall ($p < 0.0001$) (Fig. 8i-j). Overall, specific OE of DLGAP4 constructs in neurons using the *Dcx* promoter revealed perturbed migration, while the number of *Cux1⁺* cells seems to be reduced only when *Dlgap4* expression is also impaired (mutated, KO or KD *Dlgap4*) in progenitors.”...

Fig5e, f, g: Low magnified view of the cerebral walls are also needed to show the overall positioning of the electroporated cells. This is also the case of Fig5g: because the UL depth differs at the anatomically different region. The low-magnified image showing only the ShmiRNADlgap4 electroporated area showing thin-UL (but the next, neighboring non-electroporated CP are is not thin) is also needed.

We have now greatly improved the migration / *Cux1* results and verified we are analyzing equivalent regions in each condition. New images were included in the manuscript from the beginning (Fig.2a-c), to clarify and define the cortical region analyzed throughout the manuscript. We have included complementary low magnification panels specific to this point to show the non-electroporated regions analyzed. New experiments performed where the animals were euthanized at P8 (Supplementary Fig. 9), shows similarly the non-electroporated and the electroporated regions, where the *Cux1* phenotype can be observed in the *Dlgap4* KD condition as well. Please, find all this information in Fig.2a-c, Suppl. Fig.4b, Suppl. Fig 9 and the supplementary file with low magnification images.

Fig6e, Because the IUE method cannot precisely control the number of the electroporated cells, comparison of total GFP+ is not so informative. Additional examinations related to cell proliferation such as %BrdU-labeling and %PH3 in total GFP+ cells are needed.

In the experiments showing % *Blbp-GFP⁺* cells among conditions (Fig6e), we aimed to show that the presence of both ShmiRNADlgap4 and the human DLGAP4 mutant construct specifically had a cytotoxic effect on RGs by IUE (reduced total counts). This was not observed in Rescue experiments with the WT DLGAP4 construct. In order to address the reviewer’s comments on proliferation, we have performed new complementary rescue experiments including WT and mutant DLGAP4 constructs. A 30 min BrdU pulse allowed us to study active cell cycle cells in S-phase and results were included in Supplementary Fig. 8c-e. Results showed that only the WT construct was able to rescue correctly the proliferation defects induced by *Dlgap4* KD. As mentioned before, proliferation index and cell cycle exit were also evaluated (new experiments and images are shown in Fig. 5c-e and Supplementary Fig. 5f-h).

Additionally, as mentioned before, we evaluated the mitotic marker PH3 co-labeled or not with *Blbp-GFP*, in experiments performed 48h after IUE by *en face* view (see. Fig. 5f-i). We analyzed the mitotic marker PH3 co-labeled or not with *Blbp-GFP* by enface confocal imaging. We analyzed PH3⁺*Blbp-GFP⁺*/Total *Blbp-GFP⁺* cells aimed to evaluate the cell autonomous effects. In addition, we analyzed PH3⁺*Blbp-GFP⁻*/Total *Blbp-GFP⁺* cells located at a distance <10 μm from the *Blbp-GFP⁺* ones in order to analyze non-cell autonomous effects. Finally, total PH3⁺*Blbp-GFP⁺*/Total *Blbp-GFP⁺* cells were counted per ROI. These new data showed an increased amount of mitotic cells (PH3⁺) in ShmiRNADlgap4 brains, consistent with the increase of Ki67⁺*Blbp-GFP⁺* cell proportions in Bin 1 (Supplementary

Fig. 5f-h). In addition, overall Ki67⁺Blbp-GFP⁺ cells are reduced in ShmiRNADlgap4 brains. Thus we have now improved the analyses as the reviewer suggests presenting results in our figures as % of total GFP⁺ cells.

Page 12: ... “ Since Dlgap4 KD brains showed increased Ki67⁺ Blbp-GFP⁺ and Pax6⁺ Blbp-GFP⁺ cells in bin 1 (Fig. 4a,c, Supplementary Fig. 5f,g), we performed en-face confocal imaging and PH3 labeling to characterize mitotic cells in this region. An increase in PH3⁺ Blbp-GFP⁺ mitotic cells by 76.5% ($p = 0.046$) was observed in ShmiRNADlgap4 compared to control brains (Fig. 5f-g).”...

Page 14: ... “ Finally, a 30 min BrdU pulse analyzed 24 h after IUE, showed that only the OE of the WT construct was able to correct the decreased proportion of BrdU⁺ Blbp-GFP⁺ cells (Ctl vs. OE DLGAP4 MUT, $p < 0.001$) (Supplementary Fig. 8c,d), with also altered distribution of BrdU cells in the mutant (Supplementary Fig. 8e).” ...

In addition,

The authors also should examine whether KD or OE (or OE of Dlgap4-MUT) effect on cell death, for example, by the IHC of anti-cleaved-Caspase 3 antibody.

We have included new immunohistochemistry experiments evaluating total activated caspase-3 (n=2-4 embryos analyzed by duplicate, using the anti-caspase-3 antibody #559565, BD, 1:250). Results showed an increase in the immunoreactivity of total activated caspase-3 marker in both overexpression conditions. These results are now included in Supplementary Fig. 8f-g. We also observed significantly increased caspase 3+ cells in Dlgap4 KD brains Fig. 8h-i.

Page 13: ...“ Moreover, active caspase-3 was increased in DLGAP4 OE conditions, also detected in GFP⁻ cells ($p = 0.045$) (Supplementary Fig. 7f-g). Indeed, active caspase-3 was also increased in Dlgap4 KD brains ($p = 0.039$) (Supplementary Fig.7h-i). Overall, these findings indicate that Dlgap4 expression levels seem to be critical to maintain VZ integrity.”

Fig.7 a,b: How do the authors explain the molecular mechanisms that DLGAP4 increases or decreases these proteins' expression levels? (FLNA, p38, Raptor, and Rictor).

This is an interesting question for which we do not yet have the answer. We performed *in vitro* experiments in RPE1 cells, in order to shed light on potential molecular pathways for Dlgap4 function, focusing on those already known to participate in other cortical malformations, which involve cytoskeleton dynamics (Jacinto et al. Nat Cell Biol. 2004; Crino, Cold Spring Harb Perspect Med. 2015; Guerrini and Dobyns, Lancet Neurol. 2014). Since all proteins may interact in similar pathways, we can imagine feedback changes regulating protein levels (e.g. changed phosphorylation altering degradation). If the Dlgap4 interactor Raptor levels become increased, then Rictor levels might correspondingly change if the mTORC1 pathway is favored over mTORC2. Indeed, we demonstrated that DLGAP4 co-immunoprecipitates with DLG1 and Raptor and the OE of WT and MUT (P616 mutation) DLGAP4 involves changes in the levels of these proteins and other related ones (e.g. Rictor,

p38, FLNA). These proteins' transcripts are also expressed in progenitor cells (Telley et al. Science 2016), although the link between changes in the expression levels and the phenotype observed in the mouse developing cortex remains to be clarified. We also do not discard other molecular pathways being implicated in *Dlgap4* function in progenitors. Therefore, the molecular mechanism by which *Dlgap4* participates (or not) in these pathways remains to be clarified but will most probably be pursued by our group or others in the future. We have included more discussion points regarding the reviewer's question aiming to open some potential lines for future investigations (see Fig. 7 and pages 22-23 in the discussion section).

Reviewer #2 (Remarks to the Author):

In this manuscript, Romero et al., describe the identification of novel variations in *DLGAP4* in patients with subcortical heterotopia. *DLGAP4* is a membrane-associated guanylate kinase family with known function in synapses. *DLGAP4* interacts with another subcortical heterotopia associated, microtubule-binding protein *EML1*. Elegant gain and loss of function analysis of *DLGAP4* indicates a role in cell fate, proliferation and migration. *DLGAP4* function involves actin cytoskeleton and mTOR signaling pathways. These studies not only identify a new heterotopia linked genetic mechanism, but also illustrates its importance in progenitor development in the embryonic cortex. Addressing the following issues will help provide a clear link between *DLGAP4* and progenitor development and how disruptions in these functions can lead to cortical heterotopia.

1) What is the effect of increased *Tbr2*⁺ cells observed with *Dlgap4* knockdown? Does it lead to increased generation of cortical neurons or ectopic collection of them away from the cortical plate?

As mentioned above in response to reviewer 1, our new data reveals that *Tbr2*⁺ cells have abnormal morphologies during development. Also, we have performed additional IUE experiments at E14.5, using *Ctl.*, *Dlgap4* KD and *Dlgap4* mutant OE conditions and analyzed the brains at P8. We observe in both conditions delayed migration and cells ectopically located close to the lateral ventricle. The identity of these cells was confirmed by co-immunolabelings either with *Cux1* (layers II-IV) or *Ctip2* (layers V-VI), showing the presence of many *Ctip2*⁺ *GFP*⁺ cells in the heterotopia-like region (see Supplementary Fig. 9). A decreased proportion of *Cux1*⁺*GFP*⁺ cells was observed at P0 (Figure 8a-b) and this is also evident at P8 above the heterotopia (Supplementary Fig. 9). This may be explained by abnormal proliferation, increased cell cycle exit and increased cell death (included in responses to reviewer 1). Overall, we see no evidence of an increased generation of cortical neurons. We have modified the manuscript and improved figures to better present this outcome (presented below).

Also related to this comment, we characterized adult *Dlgap4* knock-out (KO) mice, now included as Suppl. Fig. 10 and Suppl. material. Heterozygous *Dlgap4* mice show clear neuroanatomical defects including microcephaly, with a small neocortex and hippocampus. Homozygous KOs are not compatible with life. Despite the absence of heterotopia-like phenotypes in heterozygous KOs, these results strongly suggest a contribution of *Dlgap4* in the control of size of several telencephalic regions, further pointing to a role of *Dlgap4* in neuronal progenitors, with mutation not leading to an overall increase in neuron number.

Page 17: ... “ **Dlgap4 KD and overexpression leads to anomalies in neuronal migration**

Neurons born after E14.5 are destined for the superficial layers of the cortex, we hence stained for *Cux1*, a marker of cortical superficial layers II/III. Fewer cells in the *Dlgap4* KD condition were labeled for *Cux1*, and the thickness of the *Cux1* staining in the CP was also significantly reduced (20%, $p = 0.044$) as well the density (32%, $p = 0.0066$) (**Fig. 8a-c**). *Blbp-GFP*⁺ labeling also indicated a possible migration defect since many mutant cells were still observed distributed along the cortical wall at P0 (**Fig. 8d**). In addition, proportions of *GFP*⁺ cells were also ectopically located at P8 forming abnormal clusters of cells in both *Dlgap4* KD and *DLGAP4* mutant OE (**Supplementary Fig. 9**). The identity of these cells was confirmed by co-immunolabelings either with *Cux1* or *Ctip2* (marker for cortical deep layers), showing the presence of many *Ctip2*⁺ *GFP*⁺ cells (arrowheads) in the heterotopia-like region (see **Supplementary Fig. 9**). The *Cux1*⁺ cell layer also appeared to be reduced in thickness at P8 in mutant conditions, likely due to a combination of several factors such as proliferation, apoptosis and migration defects (**Supplementary Fig. 9**). ” ...

Page 18: ... “ **Dlgap4 heterozygous knockout (KO) mice show neuroanatomical defects in the dorsal telencephalon**

To further study the role of *Dlgap4* during brain development, we obtained mouse mutants from the International Mouse Phenotyping Consortium produced using the knockout-first allele method⁵³ (see Supplementary information, **Supplementary Fig. 10a**). The expected number of WT and heterozygous mice were observed, but no homozygous animals, suggesting that the double dosage of the mutant allele of *Dlgap4* is not viable. Using a recently developed robust approach for the assessment of 40 brain parameters across 22 distinct brain regions⁵⁴, we analyzed neuroanatomical defects in adult *Dlgap4* heterozygous mice (**Supplementary Fig. 10b-c**, Supplementary information). A number of brain structures showed significantly decreased size in heterozygotes when compared to WTs (**Supplementary Fig. 10b**). The total brain area was reduced by 22% ($p = 0.0004$), with especially decreased size of the hippocampus (29%, $p = 0.00005$) (**Supplementary Fig. 10d**), the corpus callosum (24%, $p = 0.0009$), the anterior commissure (29%, $p = 0.00006$), the thalamus (15%, $p = 0.018$) and the neocortex (10%, $p = 0.0004$) compared to WTs. In order to determine which cortical layers were most affected, the cortex was divided into four bins corresponding to cortical layers I, II-IV, V and VI (**Supplementary Fig. 10e**). We found that Bins 2-4 were reduced in size when compared to WT, including layers II-IV (Bin 2) 28%, $p = 0.03$ and layer VI (Bin 4), 16%, $p = 0.05$ (**Supplementary Fig. 10f**). Finally, cell counts across the entire brain were decreased by 20% ($p = 0.03$) (**Supplementary Fig. 10g**). Despite the absence of heterotopia-like phenotypes in this chronic heterozygous KO model, at least at the level of the unilateral sagittal brain region studied at Lateral +0.60 mm, these results strongly suggest that *Dlgap4* plays a role in determining the size of several telencephalic structures.” ...

2) *DLGAP4* is widely expressed in embryonic cortex, including in migrating neurons?

Yes, please see the expression pattern in the mouse developing cortex by *in situ* hybridization in Fig. 2a and Supplementary Fig. 2a, RNAseq data performed in developmental mouse brain related to early born neurons (EN) and late born neurons (LN) in Suppl. Fig. 2b (Telley et al. 2016), and also by immunofluorescence (IF) new Fig.2b-f.

Indeed, we improved the IF panel of *Dlgap4* and β III-tubulin (Tuj1) at E16.5, including a higher magnification showing protein co-expression. In addition, we include human embryonic brain RNAseq data (see Suppl. Fig. 1b), where the reviewer can also see that *DLGAP4* expression is present in multiple cell types in developing human brain samples.

2) cont. Disrupted migration after *DLGAP4* knockdown an indirect effect resulting from apical/basal progenitor malfunction in VZ? Or a direct effect on migrating neurons.

Firstly, we indeed show evidence for affected RG fibers after *Dlgap4* KD. Nestin⁺ RG fibers were disorganized in basal regions (yellow arrows, Fig. 3a) and also close to the ventricular surface, sometimes appearing as horizontal fibers and round cells with no evidence of basal processes (new Supplementary Fig. 4e). The disruption in neuronal migration observed will hence most probably partly result from altered RG morphology, failing as substrate for radial neuronal migration, but also from autonomous migration defects.

Indeed, secondly, as mentioned in response to reviewer 1, when we specifically changed *Dlgap4* dosage in immature neurons using a *Dcx* promoter, we observed retarded migration, showing that as well as in progenitors, *Dlgap4* also plays a role in migrating neurons. Perturbed neuronal migration was clearly observed in *Dcx-Dlgap4* overexpression experiments (see new Fig. 8e-j), however our attempts to control the knockdown using the *Dcx* promoter were unfortunately unsuccessful. We tested several conditions in different IUE experiments, but unfortunately, for some reason, the co-electroporation of the Shmi vectors with any of the vectors expressing *DLGAP4* (WT or Mutant, with or without ires-GFP, with or without Flag) under the *Dcx* promoter did not work.

Thus, even though the intention of our work was to evaluate the effects of *Dlgap4*, a synaptic protein with a well-known scaffold function in mature neurons at the postsynaptic density, in progenitor cells, the new migration data are highly relevant to help explain the phenotype. We have now clarified this in the text (page 17).

Thirdly, as mentioned in the manuscript, neurons born after E14.5 are destined for the superficial layers of the mouse cortex, which express *Cux1*, a marker of cortical superficial layers. We identified the neuronal populations derived from E14.5 *in utero* electroporated radial glial (RG) cells at P0 and P8, by co-labeling GFP with neuronal markers in three experimental conditions (control, *Dlgap4* KD and overexpression of mutant *Dlgap4*). Control brains showed GFP⁺ cells co-labeled with *Cux1* in the superficial layers of the cortex as expected. This was also observed in *Dlgap4* KD and overexpression of mutant *Dlgap4* conditions, although a decreased *Cux1* labeling was observed. These experiments are important because they show that the delayed migrating/arrested neurons already observed by P0 are also found at P8. Indeed, ectopic GFP cells with *Cux1* or *Ctip2* neuronal identities were also located close to the ventricle. We have included these data in Supplementary Fig. 9, please see page 17.

Page 18: ...“ *To study more specifically neuronal defects, we used a Dcx promoter-GFP reporter plasmid co-electroporated with a similarly neuron-specific overexpressed WT or mutant DLGAP4 (pDcx-DLGAP4-WT or -MUT -Flag-ires-GFP) constructs (Fig. 8e). Four days after electroporation, a delay in neuronal migration was found in both DLGAP4 WT*

and *MUT* OE conditions along the cortical wall ($p < 0.0001$) (**Fig. 8f-g**). An increased proportion of *Dcx-GFP*⁺ cells were found in bins 1-2 and less cells reached superficial bins 8-9 in both conditions compared to control (**Fig. 8f-g**). *Cux1*⁺-*GFP*⁺ cells were identified showing that these neurons could adopt an appropriate identity. Total *Dcx-GFP*⁺ *Cux1*⁺/*GFP*⁺ cells did not change among conditions (**Fig. 8h**, $p > 0.05$). However, similar results of retarded migration were found by analyzing *Dcx-GFP*⁺/*Cux1*⁺ cells along the cortical wall ($p < 0.0001$) (**Fig. 8i-j**). Overall, specific OE of *DLGAP4* constructs in neurons using the *Dcx* promoter revealed perturbed migration, while the number of *Cux1*⁺ cells seems to be reduced only when *Dlgap4* expression is also impaired (mutated, KO or KD *Dlgap4*) in progenitors.” ...

Thus, from these experiments, *DLGAP4* overexpression autonomously affects neuronal migration at E18.5, and this point is now taken into account when interpreting our results. The latter could be due to multipolar to bipolar transition defects or slowed neuronal migration, however, these were not further examined here, as the exact features of the RG phenotype, affecting proliferation and the substrate for radial neuronal migration, were the major focuses of our study.

3) One general missing link in this paper is a clear outlining of how *DLGAP4* malfunction in the VZ can lead to heterotopia. Additional mapping of the nature of changes in progenitor differentiation or neuronal migration will help in this regard. In addition, addressing why in spite of almost ubiquitous expression of *DLGAP4* in the embryonic cortex, heterotopia are noticeable only in some areas of cortex, will be helpful.

In agreement with the reviewer’s concern, we have modified significantly our manuscript incorporating new experiments, and taking into account comments from the other reviewers.

Linking how *DLGAP4* malfunction in the VZ can lead to heterotopia, we have included new complementary data consistent with a potential role of *Dlgap4* at the AJs. Loss of these specific *DLGAP4* protein-protein interactions in RGs is likely to be a plausible explanation for the trapping of neurons in these areas contributing to the heterotopia phenotype.

Although it is also known that some proteins present gradient expression patterns in the neocortex (Samson and Livesey, 2009), our data suggest that *Dlgap4* is largely ubiquitous. We speculate that different areas of the developing cortex may have different susceptibilities to perturbation, related to variability of mechanical tension and proliferation /migration characteristics. Of the mouse heterotopia models known, most seem to give rise to heterotopia in caudo-medial regions above the hippocampus (e.g. Cappello et al. Neuron 2012; Croquelois et al. Cerebral Cortex 2009; Lee et al. J Neurosci. 1997, Stouffer et al. Neurobiol Dis. 2016), although exact reasons for this are to our knowledge currently unknown. Our electroporations were performed mainly in the somatosensory cortex, and ventricular surface anomalies and heterotopia were hence observed in this region.

As also replied to reviewer 1, concerning AJs, we demonstrated that *DLGAP4* is able to interact by Co-IP with β -catenin (Fig. 3h-j). Furthermore, we also include new results showing that *DLGAP4* is able to co-IP with the actin-nucleation-promoting factor cortactin (Fig. 7e-g). We found that *Dlgap4* KD induces disruption of the ventricular surface and the

disorganization of AJ proteins N-cadherin and β -catenin, as well as F-actin, the latter also shown in *in vitro* experiments (Fig. 3a-g, Fig. 6a, Fig. 7). OE experiments also lead to rosette formation (Suppl. Fig. 7b). Importantly as well, DLGAP4 WT, but not mutant, rescued the ventricular surface phenotype induced by *Dlgap4* KD, shown amongst other assays, by F-actin labeling (Fig.6a). Altogether, we now better focus our manuscript, strongly suggesting that the disruption of the ventricular surface could be due to a role of the *Dlgap4* scaffold protein in the organization of RG AJ components at the apical domain and cytoskeleton dynamics.

All these findings were included in the new version of the manuscript (see below) and in the discussion section (pages 21-22).

From page 10: ... “ Apical RGs are integrated into the AJ belt mediating cell–cell adhesion, in which cadherin, cytoplasmic catenins and F-actin are critical components ²⁰. Even in the absence of ventricular extruding ectopic GFP⁺ cells, a decrease in ventricular surface expression of F-actin was observed in the electroporated regions ($p < 0.01$) (**Fig. 3e-f**). Aberrant β -catenin and N-cadherin expression and localization were also evident (**Fig. 3g**, white arrows). This may indicate a ventricular surface weakening, potentially due to cell-to-cell adhesion defects contributing to a ventricular ectopic cell phenotype. We also found that Flag-DLGAP4 co-immunoprecipitated with GFP- β -catenin (**Fig 3h-j**), and when DLGAP4 is mutated with the P616-3 variant, co-IP of β -catenin is reduced by 47% ($p = 0.045$) (**Fig 3h-j**). These experiments further suggest a role for *Dlgap4* at the ventricular surface. Hence, *Dlgap4* KD results in disruption of cortical development, impacting ventricular surface integrity and cell positioning.”...

Page 13: ...“ In addition, rosette-like structures were present in the VZ potentially indicating changed adhesion between cells (**Supplementary Fig. 7b**).”...

...“ Next, we sought to evaluate whether the *Dlgap4* KD phenotype could be mitigated by concomitant DLGAP4 OE with the WT or mutant construct (resistant to the ShmiRNA). WT (75%, 9/12 brains) but not mutant (0%, 0/9 brains) DLGAP4 was able to rescue the continuity of the ventricular surface and the subependymal heterotopia-like phenotype (**Fig 6a**). F-actin labeling showed a disrupted pattern in mutant conditions compared to control or WT OE (**Fig. 6a**).”...

From page 15: ...“ **DLGAP4 impacts actin cytoskeleton dynamics in vitro**

Since the actin cytoskeleton has been shown previously to be perturbed in different types of heterotopia ⁴⁰⁻⁴³, we further assessed the potential role of DLGAP4 on actin cytoskeleton dynamics, using the human Retina Epithelial Pigmented (RPE1) cell line, transfected with WT or mutant DLGAP4. RPE1 cells form extensive lamellipodia, a highly compact meshwork of actin filaments at the leading edge of the cell, together with a variable number of filopodia (**Fig. 7a**, GFP-empty control). Of note, no significant differences were found in the proportion of GFP⁺ transfected cells in DLGAP4 WT vs mutant conditions, nor in the expression levels of DLGAP4 protein (data not shown). Transfections with WT and mutant DLGAP4 showed a decreased proportion of cells presenting lamellipodia morphology compared to control (by 26%, $p = 0.0029$ and 52%, $p < 0.0001$, respectively) (**Fig. 7b**). Increased filopodia elongation was observed similarly with both protein conditions (control:

3.4 $\mu\text{m} \pm 0.23 \mu\text{m}$; WT DLGAP4: 6.71 $\mu\text{m} \pm 1.08 \mu\text{m}$, $p = 0.036$; mutant DLGAP4: 6.61 $\mu\text{m} \pm 0.5 \mu\text{m}$, $p = 0.041$) (**Fig. 7c**). Actin stress fibers are long bundles of filaments extending across the cell, making links to the extracellular matrix via integrins and focal adhesion complexes⁴⁴. Organized stress fibers were not significantly affected (**Fig. 7d**). The changed lamellipodia and filopodia however further suggest that DLGAP4 may influence actin cytoskeleton dynamics, with potentially subtle differences between WT and mutant proteins. To further explore these findings, we performed co-IP experiments with the actin-nucleation-promoting factor cortactin, known to interact with the Dlgap4 interactor, Shank2⁴⁵. Cortactin is an actin binding protein promoting among other functions, lamellipodia persistence, actin polymerization and cytoskeletal remodeling during the epithelial–mesenchymal transition (EMT). It interacts with AJ components allowing F-actin accumulation, and is an important factor during the breakdown and formation of AJs⁴⁶⁻⁴⁷. Cortactin is expressed in progenitors and neurons in the developing neocortex and hippocampus^{32,33,48}. As expected, GFP-DLGAP4 was detected in anti-Flag cortactin IPs and vice versa (**Fig. 7e-g**). We found that the co-IP of Flag-cortactin with the GFP-DLGAP4 mutant was reduced by 54% ($p = 0.0018$) (**Fig. 7f**). These results together with the abnormalities of F-actin expression and AJ integrity at the ventricular surface in the Dlgap4 KD condition (**Fig. 3e-g, Fig. 6a**), suggest a role of Dlgap4 in actin cytoskeleton organization and/or dynamics, most probably involving cortactin.”...

Reviewer #3:

Summary and major concerns

This manuscript examines the role of Dlgap4 in the pathogenesis of cortical malformations seen in human patients. The authors describe a range of interesting disruptions to cortical development following knockdown of protein expression in the mouse, at the levels of both progenitor and postmitotic cells, and further describe a range of DLGAP4 interactors using biochemical approaches.

Below I describe my main criticisms of the paper in general terms, followed by more specific comments on the text.

1) Meandering experiments

The manuscript describes effects of Dlgap4 manipulation on cell proliferation, cell cycle exit, apical surface integrity, the ratios of apical versus basal progenitor cell types, the localization or abundance of apical markers, lamellipodia morphology, cell localization, and cortical upper layer thickness. Several distinct interacting partners for DLGAP4 are described in cell culture and immunoprecipitation experiments.

However, these many disparate ideas do not form a sufficiently satisfying narrative. In particular, in no case has any particular function of DLGAP4, or one of its interaction partners, been convincingly linked to any specific phenotypic alteration to cortical development. Many potential leads are identified, but then not explored further and validated in vivo during mouse development. Several observations, such as the very interesting disruptions to the apical surface, could serve as launch points for more focused lines of experiments, while omitting other things.

We thank the reviewer for this comment, in line with the other reviewers, and we have now reorganized our data, included new complementary experimental information, some figures/panels were removed from the main manuscript to supplementary figures (or even deleted from the manuscript) and *vice versa*. We believe that the manuscript is now significantly improved not only regarding the experimental information but also in the narrative of the discoveries presented.

We have now focused more deeply on the ventricular surface phenotype in the *Dlgap4* KD condition. To this end, we have performed Co-IP experiments of DLGAP4 with key proteins such as β -catenin, N-cadherin and cortactin. Particularly, cortactin is an actin-nucleation-promoting factor, influencing among other functions, lamellipodia persistence, actin polymerization and cytoskeletal remodeling during the epithelial–mesenchymal transition (EMT) (Schnoor et al. Trends Cell Biol. 2018; Bryce et al. Curr Biol. 2005; Uruno et al. Nat Cell Biol. 2001). We hence make a more solid connection to F-actin organization and AJs, important for ventricular surface integrity, and notably subependymal heterotopia can arise due to FLNA mutations, a further actin and adhesion molecule binding protein (as discussed on pages 20-22). DLGAP4 also interacts with EML1, a subcortical heterotopia protein influencing both the actin and microtubule cytoskeleton (Uzquiano et al. Cell Reports. 2019). As mentioned in response to reviewers 1 and 2, loss of specific DLGAP4 protein-protein interactions, in RGs as well as migrating neurons, provide important plausible explanations for the generation of the subependymal heterotopia phenotype in the cortex.

Specifically, we have demonstrated that DLGAP4 is able to interact by Co-IP with the adhesion molecule β -catenin (Fig. 3h-j). Furthermore, we also included new results showing that DLGAP4 is able to co-IP with cortactin (Fig. 7e-g). We demonstrated that *Dlgap4* KD and OE induces the disruption of the ventricular surface (Fig. 3a-f), as well as rosette formation (Suppl. Fig.7b) and the disorganization of AJ proteins N-cadherin and β -catenin, as well as F-actin at the ventricular surface and in cultured cells (Fig. 3a-g, Fig. 6a, Fig.7). WT but not mutant DLGAP4 rescued the ventricular phenotype induced by *Dlgap4* KD, shown by F-actin labeling (Fig.6a). Disruption of the ventricular surface is likely to be due to a role of the *Dlgap4* scaffold protein in the organization of RG AJ components. All these findings were included in an improved version of the manuscript, as presented in the replies above to reviewers 1 and 2.

2) Lack of distinction between effects in progenitors and effects in neurons
I suggest that, from the beginning, the manuscript makes systematically a very clear distinction between which functions of DLGAP4 might be relevant in which cell types and for which phenotypic aberrations. The protein is expressed widely throughout the cortical plate and neuroepithelium. The cortical phenotype is thus likely a mosaic of problems due to loss of function in both progenitors and neurons, but the manuscript fails to achieve many mechanistic insights because this idea is largely ignored. This is especially clear at the end when the model is presented: the cell type is entirely omitted.

We completely agree with the reviewer. Although we focused our work on *Dlgap4* function in progenitor populations (performing IUE at E14.5 with analyses at E15.5 to E16.5) we

have also now confirmed a role for Dlgap4 in migrating neurons (as described in response to reviewers 1 and 2, e.g. please see the response to reviewer 2's point 2, see also page 17-18 of the results section and the next answer to this reviewer's comments) and hence the phenotype can indeed be explained by several contributing cell types. We have now clarified this in the manuscript (e.g. see page 19-21 in the discussion).

Related to the comment on the model, this was a representation of what we postulated might occur in RGs under normal and mutant conditions, as previously mentioned in the figure legend. To avoid confusion or over-interpretations we have decided to remove these models from the manuscript, especially since we show that migration is also autonomously affected.

3) The meandering nature of the experiments described above, as well as the discussion, would be greatly improved with a renewed focus on cell-type-specific effects. For example, by knocking down expression in the mouse neocortex using a neuron-specific promoter, you could begin to tease apart different aspects of phenotype. Are the radial fibers still disrupted? Are neurons still mislocalized along the neuroepithelium, or the apical surface intact? It would be especially interesting to ask whether neurons are found ectopically below the ependyma due to loss of apical adhesion, in progenitors, or rather mismigration due to disruptions in postmitotic cells is instead sufficient. You could further ask which interaction partners are expressed in progenitors vs neurons, and you could search for readouts of related signaling pathways in vivo as well.

I am afraid that the suggested changes—to describe all effects in the language of cell autonomous effects—would require substantial additional experiments and re-writing, but should result in a greatly improved manuscript and help to organize the many ideas presented.

We acknowledge the reviewer's comments and suggestions and have taken them into account as best we could, we now feel our manuscript is greatly improved. As mentioned previously, we performed complementary experiments and analyses to improve the manuscript and used a cell type-specific promoter (Dcx) in order to test Dlgap4 function in neurons specifically. Use of the Dcx promoter with our KD constructs did not work in our hands, however we successfully overexpressed Dlgap4 in a neuron-specific manner. The new data is now shown in Figure 8, describing retarded neuronal migration under these conditions. Importantly, there was no visible perturbation of the apical surface. Interestingly though, there was an abnormal accumulation of neurons in the VZ. Thus mis-migration almost certainly contributes to the phenotype, although ventricular surface breakages due to effects in progenitors specifically will also contribute to causing cell accumulation in the ventricles. We can conclude that the phenotype we observe with KD is likely to be due to a mosaic of anomalies in RG, IPs and neurons collectively. As mentioned in reply to reviewers 1 and 2, our data are also consistent with both cell autonomous as well as non-cell autonomous effects (e.g. Tbr2 (as mentioned in reply to reviewer 1), Tuj1, Nestin, PH3 results).

Page 17: ...“ *To study more specifically neuronal defects, we used a Dcx promoter-GFP reporter plasmid co-electroporated with a similarly neuron-specific overexpressed WT or mutant DLGAP4 (pDcx-DLGAP4-WT or -MUT -Flag-ires-GFP) constructs (Fig. 8e). Four*

days after electroporation, a delay in neuronal migration was found in both DLGAP4 WT and MUT OE conditions along the cortical wall ($p < 0.0001$) (Fig. 8f-g). An increased proportion of Dcx-GFP⁺ cells were found in bins 1-2 and less cells reached superficial bins 8-9 in both conditions compared to control (Fig. 8f-g). Cux1⁺-GFP⁺ cells were identified showing that these neurons could adopt an appropriate identity. Total Dcx-GFP⁺ Cux1⁺/GFP⁺ cells did not change among conditions (Fig. 8h, $p > 0.05$). However, similar results of retarded migration were found by analyzing Dcx-GFP⁺/Cux1⁺ cells along the cortical wall ($p < 0.0001$) (Fig. 8i-j). Overall, specific OE of DLGAP4 constructs in neurons using the Dcx promoter reveals perturbed migration, while the number of Cux1⁺ cells seems to be reduced only when Dlgap4 expression is also impaired (mutated, KO or KD Dlgap4) in progenitors.”...

As mentioned before, our main scope was to study a potential non-synaptic role of Dlgap4 in cortical progenitors, and the consequences of a loss of function variant found in a patient with cortical malformations (subependymal heterotopia). We focused our protein partner analyses on the apical surface interactions, since AJs and the actin cytoskeleton are likely to be regulated by scaffold proteins. We do not rule out that some of these interactions may be important also in IPs and migrating neurons. Interactors studied were related to the cytoskeleton, adhesion and polarity structures and functions in progenitor cells. We show here a comparison of the expression levels in the developing mouse cortex for some of these partners (Dlgap4, Dlg1, cortactin, β -catenin, F-actin and LGN) based on single cell RNA seq data presented in Telley et al. Science 2016. As shown, the partners studied are relatively ubiquitous and expressed in all cell types. We feel that further analyses on the pathophysiological mechanisms of Dlgap4 in IPs and neurons exceed the aims of this paper, showing a role for the synaptic protein Dlgap4 in progenitors. We do agree though it would be interesting to pursue this in the future.

Other comments

Abstract

“Mouse models of this disorder are rare...”

You should clarify here whether you mean 1) researchers rarely use mice to model subcortical heterotopia, or 2) disease-causing mutations known from humans rarely produce comparable effects in mice.

According to the reviewer’s comment, we have changed the sentence to be clearer, and indeed we were referring to 2).

... “ *Mouse and human heterotopia mutations were identified in the microtubule-binding protein Echinoderm microtubule-associated protein-like 1, EML1.* ”...

“we identified a key patient with similar phenotype...”

The wording here is confusing- I think you mean a “patient with an EML1-like phenotype, but with a novel variation in DLGAP4.”

We now refer to an EML1-like phenotype throughout the manuscript, e.g:

... “ *Further exploring pathological mechanisms, we identified a patient with an EML1-like phenotype and a novel genetic variation in DLGAP4.* ”...

Introduction

You can remove the parenthetical [in gyrencephalic brains], as bRGs can be found in lissencephalic mammals including mice, just in more modest numbers. See for example Vaid et al 2018 doi: 10.1242/dev.169276

We agree with the reviewer and we have re-phrased as follows:

...“ As well as generating post-mitotic neurons, RGs give rise to other progenitor cells including intermediate progenitors localized in the subventricular zone (SVZ), and basal RGs (bRGs) mainly accumulating in an outer SVZ in gyrencephalic brains ¹³⁻¹⁸.”...

Results

“the abnormal C-terminus was also predicted to affect the overall structure of the protein.”

A little bit more description here would be helpful- affect it how? Which domains are lost or preserved in your model?

We have now described more precisely this part in the results section corresponding to Fig.1. Also, we have moved the I-TASSER predicted models to Supplementary Fig.1a.

Page 5-6: ...“ P616-3 presented a *de novo* variant consisting of an insertion of a 7-nucleotide repeat (exon 11, c.2714_2715insCAGCTGG), affecting the last part of the conserved GH1 domain, a Guanylate Kinase Associated Protein (GKAP) homology domain 1 (**Fig. 1b-d**). In DLGAP4, the GH1 domain comprises the amino acids (aa) between the glutamic acid at aa 804 (Glu804) and the tryptophan Trp907 ²⁵. The first of three consecutive CAGCTGG sequences starts at the Asn905 in P616-3, whereas this sequence is only duplicated in the normal human population. The insertion leads to a frameshift mutation changing the open reading frame and resulting in a different C-terminus after the Leu909 (**Fig. 1c, e**), including the loss of a poly proline rich domain. The last 80 aa of the wild type (WT) protein are hence lost and replaced by 95 alternative aa. This mutation has a disease-causing predicted value of 1 in a Mutation Taster analysis ²⁹. In addition, DLGAP4 has a pLI score of 0.992 in the Genome Aggregation Database (gnomAD), suggesting that the gene is highly intolerant to loss of function mutations.

In order to assess if the P616-3 mutation affects the overall predicted 3D structure and function, we took advantage of the I-TASSER iterative method ³⁰. As well as the abnormal C-terminus, this method predicts changes in the overall structure of the protein (**Supplementary Fig. 1a**, DLGAP4 mutant). Thus, the 7 nucleotide insertion affecting the C-terminus of DLGAP4 in patient P616-3 is predicted to lead to multiple changes and to impact overall protein function ³¹.”...

“In parallel, trio-based exome sequencing revealed a second family...” You describe the first patient P616-3 as having subependymal heterotopia and occipital lobe polymicrogyria. However, you describe the new patients from family P477 as having instead “pachygyria.” You could clarify here how the cortical phenotype of this patient is similar or different to the first patient. You might also point out where the DLGAP4 mutation in family P477 lies with respect to the protein domains.

We now added on page 6: “Other cortical development gene mutations e.g. in DCX, have also been shown to give rise to subcortical heterotopia or reduced cortical folding problems such as pachygyria, depending on the patient ⁶. In parallel, trio-based exome sequencing revealed a second family, P477, with affected monozygous twins having a distinct cortical malformation characterized by parieto-occipital pachygyria.”...

We also define polymicrogyria in the introduction:

Page 4: ...“ *Patients have a mixed form of subependymal and subcortical heterotopia, as well as macrocephaly, polymicrogyria (too many folds on the surface of the brain) and/or agenesis of the corpus callosum* ^{22,23}.”...

Concerning the second family, as well as information described in “***DLGAP4 mutations are associated with human cortical malformations***”, also in Suppl. Information we now say ... “*The P477 family shows a variation in DLGAP4 (described in the text and in Suppl. Table 1), inherited from the father, which is located downstream of the GH1 domain within the C-terminal region (the GH1 domain comprises the amino acids between Glu804 and Trp907 of DLGAP4 protein) (Fig. 1b-c)* ²⁵”...

Regarding Figures 2 and 3, the expression data for Dlgap4 in the neocortex seems a more natural starting point. I would move the expression data to before the interaction and immunoprecipitation data.

In agreement with the reviewer’s suggestions, we have changed the order of the figures (see new Fig. 2, Fig. 3 and Suppl. Fig. 1 and Suppl. Fig. 2). We agree this is a better order.

You might consider moving a piece of the Dlgap4 ISH data from the supplement to the main text- it would be nice to see the ISH and IHC data adjacent to one another.

This is indeed a good idea. We have now moved the ISH data from supplementary to the main figure (please see new Fig. 2 and new Suppl. Fig. 2).

“Dlgap4 labeling appeared present in the somal regions of neuronal progenitors and neurons in the mouse brain (Fig. 3a)”

You could be more specific about the locations of these cell types- you are inferring the identity of the cell types based on their location, but you are not presenting costaining with cell-type-specific markers. E.g., the cells in the ventricular zone are probably apical radial glia, whereas staining in the cortical plate is probably neuronal.

According to the reviewer’s suggestions, we have explained more clearly the presumed neuronal progenitors that we are talking about in each zone of the developing neuroepithelium. Also, we have included complementary new experiments co-labeling Dlgap4 with different markers (e.g. Dlgap4/TuJ1, and Dlgap4/Tbr2) and included this in the new Fig. 2. Thus, we have improved the images shown in the previous version of the manuscript, including a new panel with low and high magnification images showing the Dlgap4 co-labeling with TuJ1 in the VZ/SVZ at E16.5 of developing mouse neocortex (Fig. 2d-e). We have now included new images showing the immunolabeling of Dlgap4 and Tbr2, which is represented in a single confocal acquisition plane (Fig. 2f). Finally, because of antibody incompatibilities, we could not show the co-labeling with Pax6 or Sox2 markers, although we clearly observe Dlgap4 labeling in Tbr2 negative cells in the VZ (the majority) in Fig. 2f.

Page 7-8: ...“ *Dlgap4 is expressed in the mouse developing cortex*

We assessed *Dlgap4* expression by *in situ* hybridization (ISH) at different stages of mouse cortical development (from E13.5 to E18.5, **Fig. 2a**; **Supplementary Fig. 2a**, and data not shown). At every stage, *Dlgap4* was expressed in the VZ, where RGs are located, as well as in the cortical plate (CP), populated by neurons. *Dlgap4* was also detected in the SVZ, in the intermediate zone (IZ) populated by migrating neurons, as well as the ganglionic eminence (GE). Consistent with ISH results, the single-cell transcriptomic atlas of the developing mouse neocortex at E14.5, also showed the expression pattern of *Dlgap4* transcripts in the same cell types³³ (**Supplementary Fig. 2b**).

Immunohistochemistry (IHC) at E14.5 using a specific anti-*Dlgap4* antibody (**Fig. 2b-f**, **Supplementary Fig. 2c**) also showed *Dlgap4* protein expression with variable intensities in all regions of the cortical wall, including where presumed progenitor cells are located, in the VZ (apical RG) and SVZ (intermediate progenitors). The fluorescence was noticeably intense at the ventricular surface, and a strong expression was also observed in neurons in the CP. There was an apparent difference in expression of *Dlgap4* and class III beta tubulin (TuJ1) the latter notably strongly expressed in axons in the IZ. TuJ1 and *Dlgap4* however both showed expression in the cell bodies of migrating neurons in the SVZ³⁴ (**Fig. 2d-e**). Indeed, *Dlgap4* labeling appeared present in the soma regions of each cell type, including VZ RG (*Tbr2*⁻) and SVZ *Tbr2*⁺ intermediate progenitors, as well as neurons in the mouse brain (**Fig. 2c-f**, **Supplementary Fig. 2b**).

“A punctate perinuclear pattern of endogenous *Dlgap4* was observed in dividing and interphase Neuro2A cells, and in primary cultures of Pax6+ Blbp+ cells”

Since you do not show that the cells in culture express Pax6, you should remove that detail from the text.

In agreement with the reviewer’s suggestions, we have changed this part of the text.

Page 8: ... “ A punctate peri-nuclear pattern of *Dlgap4* was observed in dividing and interphase Neuro2A cells, and in primary cultures of RGs, the latter identified by GFP expression driven by a *Blbp* RG-specific promoter (**Supplementary Fig. 2d-e**).” ...

“As *Dlgap4* was highly expressed at the ventricular surface, we compared its expression pattern with polarity and ventricular surface proteins”

I did not find anything here particularly informative, especially in the relatively low power images presented. You could probably remove all of Supp Fig 2a-f, as well as this paragraph, without detracting at all from the manuscript.

In agreement with the reviewer’s suggestions, we have simplified the previous figure and reduced the results section by removing the Pearson’s correlation data from the manuscript. We decided to maintain only the results showing the co-localization of *Dlgap4* with DLG1 as supplementary data, complementing the results shown there (see new Suppl. Fig.3).

Figure 3B: *Blbp*-GFP should be a reporter for RG cells, correct? Are the GFP+ cells outside of the VZ massively displaced RG-like cells, or rather basal progenitors/neurons that retain the cellular GFP of their progenitors?

Taking into account the reviewer’s question (similar to that of reviewer 1), we have now clarified this point and included it in the manuscript text. *Blbp* is a transcription factor

expressed by RGs. At E14.5, RGs that incorporate this vector (pBlbp-GFP from Addgene) will express GFP and this fluorescent protein is stably maintained after cell differentiation to basal progenitors or neurons. As mentioned before in response to other reviewers, confirming this, we still are able to see the expression of GFP at P8, in cells which co-express Cux1, a neuronal marker of late born neurons (layers II-IV) (see new Suppl. Fig. 9).

Indeed, we used a long version of the Blbp promoter (1.6 Kb) obtained from the N. Heintz laboratory (Rockefeller University, New York). Schmid et al (2006) suggested that length of the Blbp-promoter by containing critical regulatory CIS-regions could be associated with the level of GFP expression, and very likely its stability. In their work, fluorescence intensity dwindles in embryonic brains when using a short 250 bp promoter (see also Feng and Heintz, 1995).

Please now find in the text from the “*Dlgap4 knockdown reveals a cortical progenitor phenotype*” section on page 9, the clarification of this point:

...“ *In these latter experiments, ShmiDlgap4 and ShmiCtl constructs were individually co-electroporated with a Blbp-GFP reporter plasmid at E14.5, with sacrifices 1-2 days later, giving rise to an expression of GFP in RGs, with GFP remaining after RG differentiation into intermediate progenitors and neurons* ³⁶ (Fig. 3a).”...

Figure 3C: How do you have percentages over 100%? I.e. how can more than 100% of total cells in the ROI be GFP+?

We estimated the mean of all raw values and then transformed each raw data value into a percentage, taking into account the error in each group. For example, to estimate each value (%) for a *Ctl.* group containing n=6 values of Blbp-GFP⁺ cells/ROI (each value is originally the mean of two replicates per embryo), we generated the mean of these 6 values and defined this as 100%. Then, to consider the error for these 6 embryos, we divided each raw data value by the mean of all the 6 values and expressed them as %. Another way could have been not considering the error and setting all control groups to 100% or to leave them as raw % values. Importantly, and to avoid future confusion, we now make our adopted method clearer by adding this information to the methods.

Page 35: ...“**Cell counting and quantifications...**

... “*In % analyses, average control (ShmiRNACtl or in rescue experiments, ShmiRNACtl + Flag-empty) values were defined as 100% and the percentage of each individual value was established as relative to the average number. Each experimental condition was relativized to the control.*”...

“Nestin+ RG fibers were also reduced and disorganized”

The different fiber morphologies are clear here, but how do you judge that there are fewer fibers in the experimental condition? You should quantify this or remove “reduced.”

We have modified the text as the reviewer suggested, removing the word reduced since we did not quantify, and also including new data in the same sentence.

Page 9: ... “ *Nestin⁺ RG fibers appeared occasionally disorganized in basal regions after *Dlgap4* KD (yellow arrows, **Fig. 3a**), and also close to the ventricular surface, with some horizontal and rearranged fibers (**Supplementary Fig. 4e**).” ...*

“This may indicate ventricular surface weakening, contributing to a periventricular phenotype”

What do you mean by “weakening?” In particular, you haven’t mentioned that the apical markers you employ participate in adherens junctions between apical progenitors. You could tie these ideas together a little more.

We agree and have added more information regarding AJs and apical RGs and improved this part of the text. We also discussed this point in the corresponding discussion section (pages 20-22).

Page 10: ... “ *Apical RGs are integrated into the AJ belt mediating cell–cell adhesion, in which cadherin, cytoplasmic catenins and F-actin are critical components²⁰. Even in the absence of ventricular extruding ectopic GFP⁺ cells, a decrease in ventricular surface expression of F-actin was observed in the electroporated regions ($p < 0.01$) (**Fig. 3e-f**). Aberrant β -catenin and N-cadherin expression and localization were also evident (**Fig. 3g**, white arrows). This may indicate a ventricular surface weakening, potentially due to cell-to-cell adhesion defects contributing to a ventricular ectopic cell phenotype. We also found that Flag-DLGAP4 co-immunoprecipitated with GFP- β -catenin (**Fig 3h-j**), and when DLGAP4 is mutated with the P616-3 variant, co-IP of β -catenin is reduced by 47% ($p = 0.045$) (**Fig 3h-j**). These experiments further suggest a role for *Dlgap4* at the ventricular surface. Hence, *Dlgap4* KD results in disruption of cortical development, impacting ventricular surface integrity and cell positioning.” ...*

We have furthermore included other complementary results that reinforce these ideas, as also mentioned in replies to reviewers 1 & 2 (e.g. point 3 of reviewer 2).

“To further explore these results, coimmunoprecipitation experiments were also performed for the spindle orientation protein LGN, an interactor of DLG1 36, and for which the gene (GSPM2) shows mutations in patients with Chudley-McCullough syndrome, featuring heterotopia and polymicrogyria.”

This sentence is quite a slog, I suggest breaking it up. Also, the gene name is *GSPM2*.

We apologize for the typo error and in agreement to the reviewer’s comments; we have modified and improved this part of the manuscript to be clearer, also taking into account reviewer 1’s comments.

Page 11: ... “ *The change in the ratio of progenitor cells prompted us to assess RG spindle orientations in the VZ. Preliminary quantitative analyses of *Blbp*-GFP⁺ cell anaphase angle at the ventricular surface showed a decrease of the median value (73.09° Ctl, 48.03° *ShmiDlgap4*, $p = 0.012$), consistent with an increased proportion of horizontal and oblique divisions at the expense of vertical ones (**Supplementary Fig. 6a-c**). To further explore these findings, we studied the possible interaction of DLGAP4 with a protein involved in spindle orientation, LGN (coded by the GSPM2 gene), which has been implicated in Chudley-*

McCullough syndrome, involving heterotopia and polymicrogyria ³⁷. We show that GFP-LGN was present in Flag-DLGAP4 IPs and vice versa (**Supplementary Fig. 6d-f**). Mutant DLGAP4 showed non significant reduction of immunoprecipitated GFP-LGN (36%, $p = 0.21$) (**Supplementary Fig. 6d,e,g**).”...

“Thus, Dlgap4 knockdown leads to altered spindle orientation and cell fate in the VZ. DLGAP4 is found in the same complex with both DLG1 and LGN proteins”
You have shown pairwise interactions using co-IP, but is there evidence that all three are ever found in *the same complex*?

We have clarified this point in order to avoid misunderstandings on the findings.

Page 11-12: ...“ *These results and our previous experiments support that DLGAP4 is present in the same protein complexes with either LGN or DLG1, both known for regulating spindle orientation and planar cell division* ^{19, 38, 39}.”...

According to other reviewers’ comments, and in order to focus the manuscript, we have now placed less emphasis on the impact of the altered spindle orientations and we have transferred the spindle orientation results to supplementary data (new Suppl. Fig. 6) together with the LGN co-IP results. Indeed, these results may be downstream of ventricular surface defects observed after Dlgap4 KD (see below). We also discuss this topic (page 20-22).

Fig 5d: For these quantifications, did you discriminate between GFP+ and GFP- cells? This could tell you something about possible cell autonomous or non-autonomous effects.

In order to understand better the effects of *Dlgap4* KD, and taking into account that other reviewers also mentioned this point, we performed new complementary experiments and quantifications. These were included in the main figures, replacing the previous ones. Please see below our explanation and how we have addressed and discussed this point (Fig 5c-e and in the manuscript text). We expressed the results as suggested by reviewer 1 (raw %). In addition, we included in the methods section “Cell counting and quantifications” the total range of Blbp-GFP⁺ electroporated cells counted in all the experiments. Embryos with low number of electroporated cells were excluded from the analyses. We have made it clearer for these experiments that we were talking about GFP⁺ cells.

Page 12/13: ... “ ***Dlgap4* KD leads to reduced proliferation and increased cell cycle exit** ... *Cell proliferation and cell cycle exit were also assessed at E16.5. The overall proportion of proliferating Ki67⁺ Blbp-GFP⁺ cells was reduced in Dlgap4 KD brains (27%, $p = 0.0073$) (Supplementary Fig. 5f-h). Moreover, with BrdU injection at E15.5, Dlgap4 KD brains showed a decrease in the proliferation index (21.5%, Ki67⁺ BrdU⁺ Blbp-GFP⁺ / BrdU⁺ Blbp-GFP⁺, $p < 0.0005$) and a corresponding increase in the proportion of cells exiting the cell cycle (23.3%, Ki67⁻ BrdU⁺ Blbp-GFP⁺ / BrdU⁺ Blbp-GFP⁺, $p < 0.0001$) (Fig. 5c-e). Exiting cells are increased in proliferative bins (significant in bin 2, Supplementary Fig. 5l). These results may suggest a premature differentiation of cortical progenitors into neurons.”....*

Fig 5e: You should walk through the logic of this experiment more explicitly. When was BrdU administered with respect to the IUE? Are you testing the laminar position of neurons

born at E15.5? If so, you need to compare the distributions of BrdU+ neurons between knockdown and control, i.e. the percentages of BrdU+ neurons in CP/IZ. Instead, you show a quantification for the percentage of GFP+ cells which are BrdU+, and its not clear what this demonstrates.

We apologize for the confusion. We have improved experimental data presentation in Fig.5, taking into account this and other reviewers' comments. New experiments and analyses were performed and results are now included in Fig. 5 and Supplementary Fig. 9.

We now state: E15.5 BrdU injections (24 h after IUE) showed an increased cell cycle exit 24 h later (E16.5) and reduced proliferation index (Fig. 5c-e). We also found a reduced number of BrdU+ Blbp-GFP+/ Blbp-GFP cells in the ShmiRNADlgap4 condition at P0 (previous Fig. 5e). This is also consistent with the reduced number of proliferating cells already observed at E15.5 (BrdU 30 min, Fig.5a-b) and confirmed by the analysis of the proliferation marker Ki67 at E16.5, by analyzing Ki67+ Blbp-GFP+/ Blbp-GFP cells (now included in Supplementary Fig. 5f-h). Although P0 images also showed the migration phenotype by BrdU+ labeling, and due to the fact that Fig.5e may be redundant and lead to confusion, we have eliminated this panel from the manuscript. Perturbed migration could also contribute to the reduced thickness of Cux1+ cells at P0 (see new improved neuronal migration data analyses in Fig.8a-d). The migration phenotype is also observed at P8, in new experiments performed by IUE at E14.5 and included in the new Suppl. Fig. 9.

We hope thus we have now included the relevant temporal information allowing the reader to correctly follow our experiments.

In addition, we have also included new complementary neuronal migration analyses from a novel set of experiments using the DCX promoter (Fig. 8e-j), as mentioned above.

“Fewer cells in the knockdown condition were labeled for Cux1”

If you are going to make this claim, you must quantify the total numbers of Cux1+ nuclei in a column. If true, it raises some questions which you should address: are Cux1+ neurons mislocalized, or stuck in the intermediate zone? Are there elevated levels of apoptosis that removed neurons that would have gone to the upper layers?.

As mentioned before, and to address this reviewer's point, new quantifications were performed by counting the total number of Cux1+ cells / ROI (please see Fig.8a-d).

Also, as well as retarded migration, it seems likely that increased apoptosis contributes to the cortical malformation phenotype (please see response to reviewer 2). Similarly, studying a knock-out (KO) mouse model of *Dlgap4*, now included as Suppl. Fig. 10, our new findings show that heterozygous *Dlgap4* mice show clear neuroanatomical defects consistent with microcephaly (most severely affecting upper layers). Homozygous KOs are not compatible with life. These results strongly suggest a contribution of *Dlgap4* in the control of the size of several telencephalic regions. This is likely to involve abnormal proliferation as well as apoptosis.

From Page 18: ...“ ***Dlgap4 heterozygous knockout (KO) mice show neuroanatomical defects in the dorsal telencephalon***

To further study the role of *Dlgap4* during brain development, we obtained mouse mutants from the International Mouse Phenotyping Consortium produced using the knockout-first allele method⁵³ (see Supplementary information, **Supplementary Fig. 10a**). The expected number of WT and heterozygous mice were observed, but no homozygous animals, suggesting that the double dosage of the mutant allele of *Dlgap4* is not viable. Using a recently developed robust approach for the assessment of 40 brain parameters across 22 distinct brain regions⁵⁴, we analyzed neuroanatomical defects in adult *Dlgap4* heterozygous mice (**Supplementary Fig. 10b-c**, Supplementary information). A number of brain structures showed significantly decreased size in heterozygotes when compared to WTs (**Supplementary Fig. 10b**). The total brain area was reduced by 22% ($p = 0.0004$), with especially decreased size of the hippocampus (29%, $p = 0.00005$) (**Supplementary Fig. 10d**), the corpus callosum (24%, $p = 0.0009$), the anterior commissure (29%, $p = 0.00006$), the thalamus (15%, $p = 0.018$) and the neocortex (10%, $p = 0.0004$) compared to WTs. In order to determine which cortical layers were most affected, the cortex was divided into four bins corresponding to cortical layers I, II-IV, V and VI (**Supplementary Fig. 10e**). We found that Bins 2-4 were reduced in size when compared to WT, including layers II-IV (Bin 2) 28%, $p = 0.03$ and layer VI (Bin 4), 16%, $p = 0.05$ (**Supplementary Fig. 10f**). Finally, cell counts across the entire brain were decreased by 20% ($p = 0.03$) (**Supplementary Fig. 10g**). Despite the absence of heterotopia-like phenotypes in this chronic heterozygous KO model, at least at the Lateral +0.60 mm level studied in unilateral sagittal brain sections, these results strongly suggest that *Dlgap4* plays a role in determining the size of several telencephalic structures. ”...

Did the neurons born after E14.5 adopt a deep layer (e.g. Ctip2+) fate instead?

We have included new information addressing this point. To further analyze the fate of electroporated progenitors, we performed immunostainings at P8 with Ctip2 and Cux1 markers. As mentioned above, we analyzed the identity of ectopic cells in the *Dlgap4* KD cells by co-immunolabelings either with Cux1 (layers II-IV) or Ctip2 (layers V-VI). Although most of the cells born after IUE at E14.5 adopt a Cux1 neuronal identity while reaching the cortical plate, the Cux1⁺ cell layer appeared to be reduced in thickness at P8 in mutant conditions (see Suppl. Fig. 9). Moreover, in the *Dlgap4* KD, we show that whilst some of these ectopic cells can have a late born neuronal identity (Cux1⁺GFP⁺), most of them present an early born neuronal identity (Ctip2⁺GFP⁺) (see Suppl. Fig. 9). So, in answer to the reviewer's question, in the heterotopia-like region in the *Dlgap4* KD condition, the neurons born after E14.5 can also adopt deep layer identity.

Supp Fig 3a/b: Where are the empty vector controls for these experiments demonstrating ectopic cells and rosettes? Can electroporation cause these defects?

To address the reviewer's comment, we have included an empty-vector control in this figure panel. In our experience, we have never observed defects at the ventricular surface (VS), nor rosette formation after the empty vector was electroporated *in utero* (IUE). We always performed each experiment several times, in a number of pregnant mice to be sure that the

results were accurate (~n>20 embryos per condition) and not due to artifacts. In addition, two external researchers carried out subsets of the experiments blind to the vectors electroporated. The VS phenotype was always obvious when the researcher used ShmiRNADlgap4.

We are aware of a recent article that discusses artificial heterotopias generated by physical trauma caused by the capillary injecting tube during the IUE procedure, which are rarely reported (Wang et al. Anim Cells Syst (Seoul) 2020). This is however not the case in our study which has been particularly rigorously performed in this respect. Periventricular ectopias are described in the Wang et al article, but not rosette formation, the latter phenomenon previously reported in different mutant models (including cortical malformations), only to our knowledge in the context of loss of apical-junctional complexes and subsequent loss of cell polarity, where cell-cell adhesion is affected (Buchsbaum et al., EMBO Rep. 2020; Schmid et al. Front Cell Neurosci. 2014; Cappello et al. Neuron. 2012; Kadowaki et al. Dev Biol. 2007; Rasin et al. Nat Neurosci. 2007; Lien et al. Science. 2006; Klezovitch et al. Genes Dev. 2004). We are convinced that the rosette formation we observed is related to Dlgap4 KD/OE.

“Perhaps related to this, increased numbers of GFP+ cells were also observed in the VZ with decreased numbers in the more superficial zones (Supp. Fig. 3c, d).”

As this reviewer and other reviewers suggested, we have now included new complementary data, analyses and quantifications, performing the binning analyses of the total % Blbp-GFP+ cells and other markers (see responses to reviewer 1) along the manuscript to better characterize the convincing phenotypes induced by Dlgap4 KD. Please find here some examples from the results section:

...“Upon binning analysis (**Fig. 3b**), E15.5 Blbp-GFP+ cells were found reduced in the apical-most bins (1-2) in the Dlgap4 KD condition ($p < 0.01$). Conversely, the proportion of cells in bin 3 was increased ($p = 0.026$) and a similar tendency was also observed in the more superficial bins (**Fig. 3b**).” ...

...“In addition, 48 h after IUE, Dlgap4 KD Blbp-GFP+ cells still showed an abnormal distribution compared to control (see arrows, **Supplementary Fig. 5b-c**).” ...

...“Additionally, we observed an altered distribution of Pax6+ Blbp-GFP+ cells (in bins 1-2, **Fig. 4c**), Tbr2+ Blbp-GFP+ cells (in bins 1-4) and total Tbr2+ cells (in bins 2-3) within the cortical wall, with proportionally more Tbr2+ cells present in superficial bins (**Supplementary Fig. 5d,j**).” ...

...“Dlgap4 KD brains showed a decreased proportion of proliferating BrdU+ Blbp-GFP+ cells after a 30 min period (35%, $p = 0.0084$), accompanied by abnormal BrdU+ Blbp-GFP+ and total BrdU+ cell distribution, less in bin 1 and tendencies for more in basal bins (**Fig. 5a-b; Supplementary Fig. 5e,k**). Cell proliferation and cell cycle exit were also assessed at E16.5. The overall proportion of proliferating Ki67+ Blbp-GFP+ cells was reduced in Dlgap4 KD brains (27%, $p = 0.0073$) (**Supplementary Fig. 5f-h**). Moreover, with BrdU injection at E15.5, Dlgap4 KD brains showed a decrease in the proliferation index (21.5%,

*Ki67⁺ BrdU⁺ Blbp-GFP⁺ / BrdU⁺ Blbp-GFP⁺, $p < 0.0005$) and a corresponding increase in the proportion of cells exiting the cell cycle (23.3%, *Ki67⁻ BrdU⁺ Blbp-GFP⁺ / BrdU⁺ Blbp-GFP⁺, $p < 0.0001$) (Fig. 5c-e). Exiting cells are increased in proliferative bins (significant in bin 2, Supplementary Fig. 5l).”...**

...“ Assessing *Blbp-GFP⁺* cell distribution, we found that only WT *DLGAP4* appeared to normalize this aspect of the phenotype, largely restoring *Blbp-GFP⁺* cells to control levels (Fig. 6b,d). Thus, in bins 1 and 2 the decrease in *Blbp-GFP⁺* cells in the KD condition was restored to control levels.”...

...“ Concerning distribution, *Pax6⁺ Blbp-GFP⁺* and *Tbr2⁺ Blbp-GFP⁺* cells appeared restored in bin 1, partially restored in bin 2, although the mutant lead to a noticeable trend for increased cells in basal regions (e.g. from bin 4, Supplementary Fig. 8a-b).”...

“...was able to rescue the integrity of the F-actin labelled ventricular surface (Fig. 6a).”
Please clarify how exactly you judged that the integrity of the surface was rescued.

We have now explained more clearly what we meant with the rescue of F-actin (leading to a continuous or non-disrupted ventricular surface):

Page 13: ...“ Next, we sought to evaluate whether the *Dlgap4* KD phenotype could be mitigated by concomitant *DLGAP4* OE with the WT or mutant construct (resistant to the *ShmiRNA*). WT (75%, 9/12 brains) but not mutant (0%, 0/9 brains) *DLGAP4* was able to rescue the continuity of the ventricular surface and the subependymal heterotopia-like phenotype (Fig 6a). F-actin labeling showed a disrupted pattern in mutant conditions compared to control or WT OE (Fig. 6a). Overall numbers of *Blbp-GFP⁺* cells were found to be systematically lower when the KD was combined with the *DLGAP4* mutant OE condition (36% lower, $p = 0.033$) (Fig. 6b,c).”...

“Thus, the mutant protein incorporating an altered C-terminus shows differences from WT *DLGAP4*, and is notably unable to correctly rescue certain cortical phenotypes.”
It would be helpful for the reader here if you briefly summarized which things were rescued by the mutant version and which things were not.

We have included new complementary experiments, co-electroporating concomitant *DLGAP4* OE with WT or mutant constructs (resistant to the *ShmiRNADlgap4*) in RGs. In addition, new quantification graphs and stats were performed, improving the information provided in the previous version of the manuscript (see new Fig.6 and Suppl. Fig. 8). As suggested by the reviewer, we have briefly extended this sentence to explain clearer and summarize the findings of this section.

Page 14 ...“ Thus, mutant *DLGAP4* incorporating an altered C-terminus showed differences from the WT protein, and was unable to completely prevent the observed KD cortical phenotypes, including the ventricular surface disruption, *Blbp-GFP⁺* cell distribution and *Pax6⁺ Blbp-GFP⁺* and *BrdU⁺ Blbp-GFP⁺* cell counts. *Tbr2⁺ Blbp-GFP⁺* cell numbers and distribution were partially restored with the mutant.”...

“In the Genepaint database, an expression of Shank2 was observed in the VZ as well as the CP at E14.5 by ISH (Supp. Fig. 3f).”

This ISH image is of uninterpretable background staining. You should definitely remove this, and consider redoing the ISH.

Taking into account the reviewer’s suggestions, we have removed this image, results and sentence from the manuscript.

“These results, together with the diminished expression of F-actin in Dlgap4 downregulated mouse brains (Fig. 3f-h), suggest a role of Dlgap4-Shank2 in actin cytoskeleton organization and / or dynamics in progenitor cells.” This bit about interaction between Dlgap4 and Shank2 doesn’t seem to lead anywhere. In particular, you don’t provide any evidence through experiments or through the literature that Shank2 is at all related to the actin cytoskeleton in apical radial glia. You should follow up on this speculation with manipulation of Shank2, or consider dropping this section from the manuscript.

Taking into account the reviewer’s comments and suggestions, we have removed Shank2 results from the manuscript, considering it less relevant for the present work. Of note though, Shank2 does appear to interact with cortactin, making the link between Dlgap4 and the actin cytoskeleton.

We focus now in the manuscript on new complementary experiments investigating the potential participation of the actin-nucleation-promoting factor cortactin. Cortactin promotes among other functions, lamellipodia persistence and actin polymerization through the Arp2/3 complex, cytoskeletal remodeling during the epithelial–mesenchymal transition (EMT), it interacts with AJ components allowing F-actin accumulation at AJs, and is important for the dynamic regulation of protein complexes during breakdown and formation of AJs (Schnoor et al. 2018; Sroka et al. 2016). Interestingly, cortactin is expressed in mouse progenitors and neurons in the developing neocortex and hippocampus, and in the human developing neocortex (Kanton et al. 2019; Polioudakis et al. 2019; Telley et al. 2016). Please see the expression graphs below. We therefore feel it is quite relevant to mention cortactin results in our manuscript including its co-IP with Dlgap4 (see Page 15 and see below).

Source: <http://solo.bmap.ucla.edu/shiny/webapp/#>

Clusters:

- stem cells 1
- stem cells 2
- neuroectodermal-like
- stem cells 3
- cortical neurons 1
- mesenchymal-like ce
- MCC
- G2M5 ventral proge
- G2M5 vPC
- NSCradial glia
- radial glia 2
- midbrain/hindbrain
- G2M5 dorsal progen
- ventral progenitors 1
- ventral progenitors 2
- radial glia 1
- IP and early cortical
- gliogenicouter radi
- G2M5 dorsal progen
- chondrocytes
- cortical neurons 2

Plot output:

Source: <https://bioinf.eva.mpg.de/shiny/sample-apps/scApeX/>

Source: <http://genebrowser.unige.ch/science2016/>

“...showed a decreased proportion of cells presenting lamellipodia morphology compared to

control (Fig 6i,j).”

This should just refer to Fig 6j. Also, how did you define lamellipodia? The ideas in this paragraph have not quite congealed- you are presumably looking at lamellipodia because of some relationship with actin dynamics, but you should spell this out for the reader.

Agreeing with the reviewer, we have modified this part and included the additional information requested. The Fig panels have now been updated.

From page 14: ... “ *Since the actin cytoskeleton has been shown previously to be perturbed in different types of heterotopia⁴⁰⁻⁴³, we further assessed the potential role of DLGAP4 on actin cytoskeleton dynamics, using the human Retina Epithelial Pigmented (RPE1) cell line, transfected with WT or mutant DLGAP4. RPE1 cells form extensive lamellipodia, a highly compact meshwork of actin filaments at the leading edge of the cell, together with a variable number of filopodia (Fig. 7a, GFP-empty control). Of note, no significant differences were found in the proportion of GFP⁺ transfected cells in DLGAP4 WT vs mutant conditions, nor in the expression levels of DLGAP4 protein (data not shown).*”...

“There was also a tendency for decreased F-actin stress fiber organization with OE of the WT protein (Fig. 6l).”

You should show us images of stress fiber organization, and explain what this is, why you are looking at it, and how you detect it.

We have included a new figure panel to show the differences in what we term organized stress fibers. In addition, we have provided more details to better understand the interpretation of the results (see Fig. 7a-d). In addition, and taking into account the previous comment of the reviewer, we have extended the explanation of this part:

Page 15: ... “ *Actin stress fibers are long bundles of filaments extending across the cell, making links to the extracellular matrix via integrins and focal adhesion complexes⁴⁴. Organized stress fibers were not significantly affected (Fig. 7d). The changed lamellipodia and filopodia however further suggest that DLGAP4 may influence actin cytoskeleton dynamics, with potentially subtle differences between WT and mutant proteins.*”...

“Altogether, these results further suggest that DLGAP4 influences the actin cytoskeleton.”

This is too vague- can you relate your cell culture data to what you see in the neocortex following DLGAP4 knockdown? Is the regulation in some sense positive or negative? What kinds of actin structures are influenced, and which are not, by DLGAP4?

Regarding this point, we have discussed better this topic in the manuscript, as mentioned above, in addition with further biochemical experiments included.

Page 15: ... “ *To further explore these findings, we performed co-IP experiments with the actin-nucleation-promoting factor cortactin, known to interact with the Dlgap4 interactor, Shank2⁴⁵. Cortactin is an actin binding protein promoting among other functions, lamellipodia persistence, actin polymerization and cytoskeletal remodeling during the epithelial–mesenchymal transition (EMT). It interacts with AJ components allowing F-actin accumulation, and is an important factor during the breakdown and formation of AJs⁴⁶⁻⁴⁷.*

Cortactin is expressed in progenitors and neurons in the developing neocortex and hippocampus^{32,33,48}. As expected, GFP-DLGAP4 was detected in anti-Flag cortactin IPs and vice versa (**Fig. 7e-g**). We found that the co-IP of Flag-cortactin with the GFP-DLGAP4 mutant was reduced by 54% ($p = 0.0018$) (**Fig. 7f**). These results together with the abnormalities of F-actin expression and AJ integrity at the ventricular surface in the *Dlgap4* KD condition (**Fig. 3e-g, Fig. 6a**), suggest a role of *Dlgap4* in actin cytoskeleton organization and/or dynamics, most probably involving cortactin.”...

“The DLGAP4 partner, DLG1, has other functions as a second messenger...”

I don't think you can use the term second messenger like this- it refers to small molecules.

We agree with the reviewer and therefore, we have modified this sentence.

...“*DLG1, is targeted by the p38 family of kinases, which are known regulators of the actin cytoskeleton*^{51,52}.”...

“We hence further analyzed these proteins in RPE1 cells overexpressing WT and mutant DLGAP4 by performing Western blot analyses.”

In this section, you need to provide more context about the potential molecular targets you are looking at. How exactly are they related to the signaling pathways you describe? For instance, you drop in Raptor and Rictor without telling the reader anything about them.

In agreement with the reviewer, we have briefly explained the context (please also see our response to the point below) and the rationale of this set of experiments. We also considered as well another reviewer's comments and suggestions regarding this part of the manuscript. Overall, we tried to avoid over-simplification, and we have tried to be clearer that we present observations notably related to several key cytoskeletal signaling pathways. In addition, we have discussed this topic on pages 21-23.

Page 16...“*Finally, we investigated how DLGAP4 may further influence cytoskeletal signaling. We assessed whether DLGAP4 may have a role in the downstream mTOR pathway, since it has been described in patients presenting cortical malformations*^{4,49}. *Phosphorylated DLG1 impacts PI3K-AKT-mTOR signaling in both the PNS and CNS*^{50,51}.”...

“De-regulation of this pathway, predicted via Rictor to impact the actin cytoskeleton, is likely to contribute to the DLGAP4 phenotypes presented here.”

It is too bold here to use the word “likely.” There is no evidence for which mechanisms downstream of DLGAP4 manipulation cause which of the many described phenotypes- you should follow up on these mTOR data by asking how this signaling pathway is affected in vivo in the neocortex.

We agree with the reviewer's comments and have removed this sentence. As mentioned above, we now more precisely present all mTOR biochemical data in new Fig. 7 and the preliminary data concerning molecular mechanisms, trying to avoid over simplification. In addition, we wish to underline these results should be considered preliminary, and we agree

with the reviewer that further *in vivo* experiments and analyses would be needed to confirm our findings. Unfortunately, we could not yet perform these suggested experiments because of the pandemic context. We have hence made sure to add a sentence in the discussion stating that the results are preliminary and require *in vivo* confirmation.

In the modified Results section:

Page 16... “ Furthermore, OE of either WT or mutant DLGAP4 induced a marked increase in Raptor expression (374%, $p = 0.0028$ and 405%, $p = 0.0016$, respectively, $F_{2, 6} = 25.18$), involved in mTOR signaling, known to influence the actin cytoskeleton⁵². Interestingly, only mutant DLGAP4 induced a significant decrease in Rictor expression (47%, $p = 0.0099$, $F_{2, 9} = 8.103$) (Fig. 7h,i). We further tested for the presence of these latter proteins in the same complexes as DLGAP4 by co-transfection of tagged constructs. We found that HA-Raptor was present in the bound fraction with immunoprecipitated Flag-WT DLGAP4 (and vice versa), and bound HA-Raptor was specifically reduced (32%) with mutant Flag-DLGAP4 (Fig. 7j-l). Rictor however, did not co-IP with either WT or mutant DLGAP4 (data not shown). Altogether, OE of DLGAP4 constructs induces changes in p38 and mTOR pathway expression *in vitro*, potentially related to actin cytoskeleton modifications engaged in the DLGAP4-dependent phenotypes observed *in vivo*.”...

Discussion:

Page 23... “We hence suggest that DLGAP4 is involved in similar mechanisms, although these results are still preliminary and further experiments, including *in vivo* confirmation, would help corroborate this hypothesis.”...

Discussion

“Thus DLGAP4, known for its role in synapses, appears when associated with particular gene variations, to be a cortical malformation gene.”

Even given the qualification, I suggest avoiding the term “cortical malformation gene.”

Really, it’s a cortical formation gene. Maybe just specify that DLGAP4 is now shown to be required for x, y, and z aspects of cerebral cortex development.

In agreement with the reviewer, we have made the changes on the gene denomination, avoiding mentioning it as cortical malformation gene. As the discussion is substantially modified/improved, this sentence is no longer in the main text, however below please find an example.

... “ We provide here evidence of a crucial role of DLGAP4 maintaining the progenitor pool, regulating neurogenesis, as well as influencing migration. This work hence enlarges the pathological spectrum for a single gene, and helps to identify possible molecular pathways in RGs involved in this phenotype, adding important mechanistic insights to the understanding of pathogenic mechanisms associated with overlapping phenotypes of subependymal and subcortical heterotopia.”...

“convergent with our Dlgap4 results”

I would reserve “convergent” as a term for convergent evolution. Try “similar” or something

of that nature.

We agree with the reviewer however, this sentence was also removed from the improved version of the manuscript in the discussion section.

“Furthermore, changes in cell fate may be brought about by changed spindle orientations” This could probably be moved to a new paragraph where you describe more specifically the known mechanisms relating spindle orientation to cell fate, e.g. concepts like unequal inheritance of apical determinants, delamination and differentiation, etc.

In agreement with the reviewer, we have modified this section and included new literature to improve the discussion. In addition we took into account other reviewers’ suggestions concerning cell fate experiments, so indeed the section changed substantially. For example:

Page 22: “A number of investigations have demonstrated the essential role of AJ components in RG proliferation, as well as in maintaining their morphology, polarity and localization, important for correct cortical development^{12,19,41,69,70}. Alterations in RG morphology were found in *Dlgap4* KD brains. Moreover, *Tbr2*⁺*Blbp*-GFP⁺ cells present an abnormal morphology, showing an increased soma size and a decreased number of primary processes. We found that the OE of either the WT or mutant constructs was able to restore control *Tbr2*⁺*Blbp*-GFP⁺ cell values. These results may suggest that the N-terminal domain of the protein (WT and mutant) is involved in regulating the molecular mechanisms leading to cell differentiation. In addition, *DLGAP4* interactors *LGN* and *DLG1*, are both involved in cell polarity and spindle orientation in neuronal progenitors^{19,39,77,78} and *EML1* has also been linked to these phenomena²¹.”...

“Clearly there is overlapping molecular machinery between synapse formation and maintenance, and ventricular surface homeostasis.”

This is not clear to me. If you are going to say this, you need to spell out exactly which molecular machineries and mechanisms you are talking about and which processes are shared between synapse formation and apical surface regulation.

We agree with the reviewer’s point. We have toned down this sentence, modifying this part of the discussion (page 23), and included the references showing the role of *DLGAP4* at excitatory synapses, and emphasizing its function also in progenitors maintaining apical surface homeostasis.

Reviewer #4 (Remarks to the Author):

Romero and colleagues report a new gene, *DLGAP4*, associated with cortical malformations, suggesting a role for *DLGAP4* during cortical development. They identified a de novo frameshift *DLGAP4* mutation in a single proband with subcortical heterotopia and a missense trans-compound heterozygous mutation in *DLGAP4* and *DLGAP1* in a patient with a distinct cortical malformation characterized by parieto-occipital pachygyria. They report that *Dlgap4* is expressed in the developing cortex in mice, interacts with *EML1*, a known subcortical heterotopia gene, and both knockdown and overexpression of *Dlgap4* lead to

ventricular surface phenotypes in mice. Additionally, they showed the identified frameshift DLGAP4 mutation affects the interaction of Dlgap4 with Shank2 which impacts the actin cytoskeleton. While there are interesting genetic findings in this work, the manuscript lacks evidence to support some of the major hypotheses proposed, specifically the gain of function/gene-gene interaction model in family P477. Statistical analysis or functional data elucidating the functional impact of the missense DLGAP4 mutation and the potential interaction between DLGAP4 and DLGAP1 is necessary to support this hypothesis.

Major comments

1. Can the authors comment on the expression level of DLGAP4 and DLGAP1 in the human brain during development?

We have compared the expression of both genes in the CoDEX (Cortical Development Expression) database and human organoids (see sources below). As shown in Supplementary Fig. 1, both genes are relatively ubiquitous but importantly, there is evidence of expression of DLGAP1 and DLGAP4 in progenitors as well as in immature neurons. DLGAP4 shows evidence of higher levels of expression than DLGAP1. Part of this information was included in Supplementary Fig. 1.

Source: <http://solo.bmap.ucla.edu/shiny/webapp/#>

Damon Polioudakis†, Luis de la Torre-Ubieta†, Justin Langerman, Andrew G. Elkins, Xu Shi, Jason L. Stein, Celine K. Vuong, Susanne Nichterwitz, Melinda Gevorgian, Carli K. Opland, Daning Lu, William Connell, Elizabeth K. Ruzzo, Jennifer K. Lowe, Tarik Hadzic, Flora I. Hinz, Shan Sabri, William E. Lowry, Mark B. Gerstein, Kathrin Plath, Daniel H. Geschwind. A Single-Cell Transcriptomic Atlas of Human Neocortical Development during Mid-gestation. *Neuron*. VOLUME 103, ISSUE 5, P785-801.

Source: <https://bioinf.eva.mpg.de/shiny/sample-apps/scApeX/>

Sabina Kanton, Michael James Boyle, Zhisong He, Malgorzata Santel, Anne Weigert, Fátima Sanchís-Calleja, Patricia Guijarro, Leila Sidow, Jonas Simon Fleck, Dingding Han, Zhengzong Qian, Michael Heide, Wieland B Huttner, Philipp Khaitovich, Svante Pääbo, Barbara Treutlein, J Gray Camp. Organoid Single-Cell Genomic Atlas Uncovers Human-Specific Features of Brain Development. *Nature*. 2019. 574(7778):418-422.

2. The authors should document their analysis of the exome sequencing data in Materials and Methods, including data processing and downstream filtering criteria used.

As requested by the reviewer, we include the supplementary method section “**Analysis of human *DLGAP4* in *EML1*-like cortical malformation patients**” to material and methods (page 36-37).

...“Whole exome sequencing of peripheral blood DNA from the proband and both parents was performed using the Agilent SureSelect Human All Exon Kits v5, and sequence was generated on a HiSeq2500 machine (Illumina). Sequences were aligned to hg19 by using BWA v.0.6.1, and single nucleotide variants (SNVs) and indels were called by using GATK v.1.3. Annotation of variants was performed with GATK Unified Genotyper. All calls with read coverage of $\leq 2 \times$ or a Phred-scaled SNP quality score of ≤ 20 were removed from consideration. The annotation process was based on the latest release of the Ensembl database. Variants were annotated and analysed using the Polyweb software interface

designed by the Bioinformatics platform of University Paris Descartes and Imagine Institute (see also Suppl tables 1-2).

Filters used for variant screening were as following: (i) all variants previously observed (in dbSNP138 and/or in in-house projects) were excluded; (ii) only variants leading to abnormal protein sequence (splicing, non-synonymous, frameshift, stop) were retained; (iii) we considered the PolyPhen-2 and SIFT prediction status as informative but not restrictive. Because all patients were sporadic cases from unrelated parents, the following models of inheritance in the variant screening were considered: autosomal recessive (in particular compound heterozygous but without excluding homozygous variants), X-linked and de novo SNVs.

Genomic DNA amplifications were performed using standard procedures (primers in Supplementary Table 3), and PCR products were analyzed by direct sequencing using an ABI3700 DNA analyzer (Applied Biosystems, Foster City, CA).”...

3. Can the authors explain why other predicted damaging variants identified in the two families were not considered as pathogenic? There is no comment on MYO1C and ZFX identified in P477 and ABCG10 identified in P616.

The reviewer is correct to ask this question and we have now added the missing information. The following details explaining each gene’s initial selection or non-selection criteria as candidate have been included (see P477 and P616 supplementary information).

P477

...“MYO1C and ZFX variants were not considered as primary candidates to explain the P477 brain malformation since dominant MYO1C mutations are known to cause bilateral sensorineural hearing loss in humans (Adamek et al., Cell Mol Life Sci. 2011; Lin et al., Biochemistry. 2011) not present in patients identified here; and the Zfx knockout and conditional targeted mice present abnormalities in the hematopoiesis system (not affected in patients) but not in the nervous system (Jackson Laboratory), making it a less likely candidate.”

P616

“...The splicing variant in ABCB10 gene was not considered as a potential candidate to explain the P616 brain malformation because numerous essential splicing variants exist for this gene in the gnomAD database. TTN and LRRK2 are known disease causing genes. TTN gene was previously implicated in dominant cardiopathy and recessive muscular dystrophy (#MIM613765, 608807, 603689, 611705 and 600334). Mutation in the LRRK2 gene is one of the most common causes of inherited Parkinson disease (#MIM607060).”

4. It would be helpful if the authors can add the pLI score, missense Z score, and minor allele frequency in public databases for each identified variant in the variant table.

We agree that pLI score, missense Z score and minor allele frequency for each variant are important information to add. We have now included these from gnomAD v2.1.1 in the supplementary Table 1.

Also in the main text:

Page 6:...“In addition, *DLGAP4* has a pLI score of 0.992 in the Genome Aggregation Database (gnomAD), suggesting that the gene is highly intolerant to loss of function mutations.”...

5. Please specify the transcript ID for isoform 1 in Figure 1d as well as in the main text.

We have included in the figure and the main text, the transcript ID for *DLGAP4*, NM_014902.4 and Ensembl information: gene ENSG00000080845, ENST00000373913.3, 989 aa protein variant.

Page 5: ...“Trio-based exome sequencing identified a variation in *DLGAP4* (ENST00000373913.3) in the patient P616-3.”...

6. *DLGAP4* has multiple transcripts. Despite its high overall expression level in adult human brain, the canonical transcript (ENST00000373913.3, 989aa) which is the specific one described in the manuscript, has low or no expression in the adult brain depending on the region (based on the GTEx database). Do the two identified mutations (chr20:35155357:T:G and chr20:35154373:-:CAGCTGG) have similar functional impact in all transcripts or at least those highly expressed in the brain?

Using Alamut software, we predicted the consequences of the two variants found in P477 and P616 on the three *DLGAP4* isoforms described by Ensembl and NCBI reference databases. In all transcripts, the P616 variant results in a frameshift mutation and the P477 variant in a Ser-Ala missense mutation. This is now included in Supp. information.

patient	reference	exons	bp	AA	cDNA	protein	sift	IMT	Polyphen2	
P616	NM_014902.5	ENST00000373913.7	13	5056	989	c.2715_2716insCAGCTGG	p.Ser906Glnfs*100	/	/	/
	NM_183006.3	ENST00000340491.8	7	2392	453	c.1107_1108insCAGCTGG	p.Ser370Glnfs*100	/	/	/
	NM_001042486.3	ENST00000475894	7	3110	285	c.603_604insCAGCTGG	p.Ser202Glnfs*100	/	/	/
P477	NM_014902.5	ENST00000373913.7	13	5056	989	c.2893T>G	p.Ser965Ala	Tolerated	disease causing	probably damaging
	NM_183006.3	ENST00000340491.8	7	2392	453	c.1285T>G	p.Ser429Ala	Tolerated	disease causing	probably damaging
	NM_001042486.3	ENST00000475894	7	3110	285	c.781T>G	p.Ser261Ala	Tolerated	disease causing	probably damaging

7. The authors suggested in Discussion that the disease-causing mechanism of the missense *DLGAP4* mutation is gain of function (over expression) and might be mediated through interaction with *DLGAP1*. Please provide statistical and/or functional data to support this hypothesis.

According to the reviewer’s comment, we have performed complementary co-IP experiments in Neuro2A cells in order to study whether there is an interaction between *DLGAP1* and *DLGAP4*. These proteins belong to the *DLGAP* or *SAPAP* family and interactions between its members have never been described before. Our experiments demonstrated that both proteins share the same sub-cellular compartments and are found together in the bound fraction (the IP works in both directions), showing they are present in the same protein complexes. These results are now included in Supplementary Fig. 1c-d, and the discussion section was modified in this context, and we have taken care to avoid over speculation and over simplification of the mutation mechanisms implicated in the phenotypes.

Page 7: ...“ Given the severe cortical phenotypes, we queried *DLGAP1* and *DLGAP4* expression patterns across human cortical development. RNA sequencing expression data from single human cells showed the expression of *DLGAP1* and *DLGAP4* in progenitor cells and neurons, being higher and more ubiquitous for the latter gene (**Supplementary Fig. 1b**)³². *DLGAP1* is the closest in homology to *DLGAP4*, compared to other family members, and immunoprecipitation (IP) experiments showed that these proteins are present in the same protein complex (**Supplementary Fig. 1c-d**). These combined data led us to further assess the potential role of these proteins and notably *DLGAP4* in cortical development.”...

From the discussion where we have attempted to remain factual concerning *DLGAP4* and *DLGAP1*:

Pages 19-20: ...“ Reconstructing molecular mechanisms, we show that *DLGAP4* is found in protein complexes with key RG polarity (e.g. *DLG1*, β -catenin) and spindle orientation (e.g. *LGN*, *EML1*) proteins, other scaffold family members (*DLGAP1*), as well as with actin cytoskeleton-regulating partners (e.g. cortactin). Importantly, *DLGAP4* mutation perturbs these interactions...

...*DLGAP4* belongs to the SAPAP family, which has been described as one of the first components to reach neuronal synapses, involved in the re-localization of PSD-95 from the cytoplasm to the plasma membrane^{25,26,35}. Our study suggests scaffolding roles in cortical progenitors and migrating neurons. As well as identifying a C-terminal heterotopia mutation (family P616), we also identified one further cortical malformation family (P477) showing complex gene variations involving both *DLGAP4* and *DLGAP1*. Concerning *DLGAP4*, the P477 twin siblings notably show a C-terminal (p.Ser965Ala) variation, as well as exhibiting an occipital malformation, similar to patient P616-3. The *DLGAP4* protein sequence is highly conserved in the regions of the C-terminal disruptions identified here²⁵, and no similar variations affecting these residues have been detected in the normal population⁵⁹. It is thus likely that variations in this particular region of *DLGAP4* give rise to the MCDs observed. Increased levels of *DLGAP4* have previously been associated with early-onset cerebellar ataxia⁶⁰. We show here that experimental OE in the developing mouse dorsal cortex can also produce a phenotype. ...All in all though, it is clear that appropriate levels of *DLGAP4* are required for correct brain development.”...

8. Continuing on the point discussed above, the authors should comment on the functional impact of the missense *DLGAP1* mutation. Does it lead to decreased expression level, protein conformational change, or others?

We performed new experiments to test this with GFP-Dlgap1 in which we performed site directed mutagenesis and verified the patient mutation by Sanger sequencing.

Transfections in Neuro2a cells were performed, followed by cell lysates and western blots to compare the WT and mutant protein levels following standard protocols. As the reviewer will see (included in Supp Fig. 1e), we observed no signs of mutant protein stability/degradation nor did we observe decreased levels at the specific band molecular weight compared to WT protein.

In addition, we performed the prediction of DLGAP1 WT and mutant protein structures using the I-TASSER method. These models, as we have presented for DLGAP4, were the best-predictions taking into account the C-scores provided by the method. The C-score is a confidence score for estimating the quality of predicted models by I-TASSER. It is calculated based on the significance of threading template alignments and the convergence parameters of the structure assembly simulations. C-score is typically in the range of [-5,2], where a C-score of higher value signifies a model with a high confidence and *vice versa*. Comparing DLGAP1 models shown below (WT vs MUT DLGAP1), although there are subtle differences, we find that WT and mutant proteins show overall similar predicted structure models, suggesting no dramatic changes in conformation.

This, together with the similar expression levels of WT and mutant proteins obtained by WB, suggest that the mutation found in the DLGAP1 patient would not induce dramatic changes in the structure that would be expected to lead to stability problems. Despite this, the DLGAP1 p.(Asp466Gly) point mutation leads to a change from an acidic to a non-polar and the smallest in size amino acid. We hence speculate that certain protein-protein interactions would be destabilized. We now mention this in the Supp Fig 1 legend of the manuscript, but chose not to include these DLGAP1 models for the sake of space.

DLGAP1 WT: C-score -2.5, estimated RMSD $14.3\pm 3.8\text{\AA}$ and estimated TM-score 0.42 ± 0.14 ; DLGAP1 P477 mutation: C-score -2.49, estimated RMSD $14.2\pm 3.8\text{\AA}$ and estimated TM-score 0.42 ± 0.14 . The asterisk in green shows the point mutation site.

Minor comments

1. Supplementary Table 2 should be Supplementary Table 3 and Supplementary Table 3 should be Supplementary Table 1. All main and supplementary tables and figures should be labeled and put in the proper order.

We apologize for this mistake. We have performed the modifications indicated.

Supplementary Table 1. Summary of whole exome sequencing variants found altered in P477-3 and P616-3 probands.

Supplementary Table 2. Summary of whole exome sequencing information of the two family trios: P477 and P616.

Supplementary Table 3. Nucleotide sequences of the primers used in the different approaches performed in this study.

2. For all acronyms used in the manuscript, please spell out the full name when they are used for the first time (e.g. 'IUE' for 'in utero electroporation' in the main text).

We apologize for these omissions. We have now included the full name in all the acronyms used in the manuscript, the first time they appear in the text. For example,

Page 9: ... “ *Since the role of Dlgap4/DLGAP4 during cortical development was previously unknown, we explored this further by examining the effects of its downregulation using in utero electroporation (IUE) in the developing mouse cortex.*” ...

3. It is worth mentioning that DLGAP4 has a pLI score of 0.99 in gnomAD, suggesting this gene is highly intolerant to loss of function mutations.

We thank the reviewer for this important comment, which we have included in the revised version of the manuscript (page 6).

...“ *The last 80 aa of the wild type (WT) protein are hence lost and replaced by 95 alternative aa. This mutation has a disease-causing predicted value of 1 in a Mutation Taster analysis* ²⁹. *In addition, DLGAP4 has a pLI score of 0.992 in the Genome Aggregation Database (gnomAD), suggesting that the gene is highly intolerant to loss of function mutations.*”...

4. Throughout the manuscript the clarity of the writing could be improved; for example, the header of Supplementary Table 3 (the variant table) and penultimate sentence on page 14 are both unclear.

We apologize for the unclear sentences. We have read carefully and improved the language of the writing throughout the manuscript. For example:

Supplementary Table 3. Nucleotide sequences of the primers used in the different approaches performed in this study.

REVIEWERS' COMMENTS

Reviewer #1 (Remarks to the Author):

The authors have made substantial efforts and added new data to address the comments by this reviewer. Some minor changes are still required as described below:

Line252 and Fig4f

Fig4f may be deleted because it is redundant, and the meaning of this ratio is unclear.

Line264

In general, 'horizontal division' means the division with a vertical cleavage plane. To avoid confusion for readers, please correct this sentence.

Line290-and Fig5f

In this en face view, the F-actin AJ mesh looks intact in the KD case. As this seems to be inconsistent with other results (Fig3e etc.), the interpretation of this result is needed.

Line 411

Please add the reference for Dcx promoter.

Reviewer #2 (Remarks to the Author):

Editor: The reviewer conveyed to us that they felt their concerns, as well as those of Reviewer #3 were addressed.

Reviewer #4 (Remarks to the Author):

In this manuscript, the authors investigate the role of DLGAP4 in cortical malformation. The noteworthy results are the novel findings of Dlgap4, a known synaptic protein, on neural progenitor adhesion and migration, linking this biology to cortical heterotopia. To my knowledge, this work is original and will be of significance to the field. However, efforts to provide additional evidence to support conclusions and claims were hindered due to pandemic difficulties. The methodology is sound and meets expected standards of the field for rigor and reproducibility.

We acknowledge the reviewers for the comments and suggestions that helped us to improve our manuscript.

REVIEWERS' COMMENTS

Reviewer #1 (Remarks to the Author):

The authors have made substantial efforts and added new data to address the comments by this reviewer. Some minor changes are still required as described below:

Line252 and Fig4f

Fig4f may be deleted because it is redundant, and the meaning of this ratio is unclear.

We removed the ratio panel as the reviewer suggested, as well as the text in the results section: “*These effects were also reflected by a reduced ratio of Pax6+ Blbp-GFP+ / Tbr2+ Blbp-GFP+ cells (p = 0.012) (Fig. 4f).*”

In addition, to make the manuscript easier to follow, we included, as it was shown for the Pax6 marker, Tbr2+ Blbp-GFP+ cell distribution as panel f (Fig. 4f), removing it from Supplementary Fig. 5d.

Line264

In general, 'horizontal division' means the division with a vertical cleavage plane. To avoid confusion for readers, please correct this sentence.

We have modified the text and the text in Supplementary Fig.6a accordingly.

Page 11:... “Preliminary quantitative analyses of Blbp-GFP+ cell anaphase angle at the ventricular surface showed a decrease of the median value (73.09° Ctl, 48.03° ShmiDlgap4, p = 0.012), consistent with an increased proportion of vertical and oblique divisions at the expense of horizontal divisions (Supplementary Fig. 6a-c).”...

Line290-and Fig5f

In this en face view, the F-actin AJ mesh looks intact in the KD case. As this seems to be inconsistent with other results (Fig3e etc.), the interpretation of this result is needed.

As the reviewer suggested, we have included a sentence clarifying this point, which was missing in the revised version of the manuscript.

Aiming to answer this point, we also performed similar analyses as presented in Fig3e, measuring the intensity of the F-actin labeling comparing SCR vs *Dlgap4* KD brains, in *en face* experiments. The mean intensity of fluorescence showed a trend for reduction (15% reduction, p = 0.39), although it was not significant (see below). We decided to include these data in the manuscript, as Supplementary Figure 6h. As we mentioned in the manuscript, not all brains showed a disrupted ventricular

surface, our analyses with *en face* imaging allowed us to further examine the state of the cells when the ventricular surface was 'intact', especially to compare GFP+ to GFP- cells to address the question of cell autonomous versus non cell autonomous defects.

Mean grey value:

Unpaired t test with Welch's correction

P value	0,3891
P value summary	ns
Significantly different? (P < 0.05)	No
One- or two-tailed P value?	Two-tailed
Welch-corrected t, df	t=1,267 df=1,275
How big is the difference?	
Mean ± SEM of column A	62,68 ± 7,057 N=2
Mean ± SEM of column B	53,16 ± 2,584 N=3
Difference between means	-9,519 ± 7,516
95% confidence interval	-67,83 to 48,79
R squared	0,5573

We have clarified this in the manuscript:

Page 12: ... “ Since *Dlgap4* KD brains showed increased *Ki67*⁺ *Blbp*-GFP⁺ and *Pax6*⁺ *Blbp*-GFP⁺ cells in bin 1 (Fig. 4a,c, Supplementary Fig. 5e,f), we performed *en face* confocal imaging and phosphohistone 3 (PH3) labeling to characterize mitotic cells in this region.” ...

... “*En face* F-actin labeling revealed a tendency for reduced mean intensity in *Dlgap4* KD brains (15%, $p = 0.39$, Supplementary Fig. 6h). The areas of the three brains analyzed did not show obvious ventricular surface fragmentation, thus further confirming that non-cell-autonomous effects can occur in the absence of damage.” ...

Line 411

Please add the reference for *Dcx* promoter.

We have included the reference in the methods section:

Page 34 :... “In *Dcx*-*DLGAP4* overexpression experiments, *pDcx*-*DLGAP4*WT-Flag-ires-GFP or *pDcx*-*DLGAP4*MUT-Flag-ires-GFP were each co-electroporated with 1:1 *pDcx*-GFP⁸⁷....”

Reference

89. Belvindrah R, Nissant A, Lledo PM. Abnormal neuronal migration changes the fate of developing neurons in the postnatal olfactory bulb. *J Neurosci.* 2011 May 18;31(20):7551-62. doi: 10.1523/JNEUROSCI.6716-10.2011. PMID: 21593340

Reviewer #2 (Remarks to the Author):

Editor: The reviewer conveyed to us that they felt their concerns, as well as those of Reviewer #3 were addressed.

Reviewer #4 (Remarks to the Author):

In this manuscript, the authors investigate the role of DLGAP4 in cortical malformation. The noteworthy results are the novel findings of Dlgap4, a known synaptic protein, on neural progenitor adhesion and migration, linking this biology to cortical heterotopia. To my knowledge, this work is original and will be of significance to the field. However, efforts to provide additional evidence to support conclusions and claims were hindered due to pandemic difficulties. The methodology is sound and meets expected standards of the field for rigor and reproducibility.